# `DES-LOC`: Desynced Low Communication Adaptive Optimizers for Foundation Models

**Alex Iacob**[†,1,2]     **Lorenzo Sani**[1,2]     **Mher Safaryan**[3]     **Paris Giampouras**[*,4]

**Samuel Horváth**[*,5]     **Andrej Jovanović**[*,1]     **Meghdad Kurmanji**[*,1]

**Preslav Aleksandrov**[1,6]     **William F. Shen**[1]     **Xinchi Qiu**[1]     **Nicholas D. Lane**[1,2]

## ABSTRACT

Scaling foundation model training with Distributed Data Parallel (`DDP`) methods is bandwidth-limited. Existing infrequent communication methods like `Local SGD` were designed to synchronize model parameters only and cannot be trivially applied to adaptive optimizers due to additional optimizer states. Heuristic approaches that keep states local or reset them lack guarantees and can be unstable in compute-efficient batch regimes; conversely, `Local Adam` synchronizes all states uniformly and is provably convergent but triples communication costs. We propose Desynced Low Communication Adaptive Optimizers (`DES-LOC`), a family of optimizers assigning independent synchronization periods to parameters and momenta, enabling lower communication costs while preserving convergence. Our theoretical analysis shows that while parameter synchronization dominates the asymptotic rate in-expectation, high-probability convergence guarantees require at least infrequent synchronization of the second momentum. Furthermore, we prove that more frequent momentum sync permits larger stable step sizes. Experiments on language models of up to 1.7B show that `DES-LOC` can communicate $\mathbf{170\times}$ less than `DDP` and $\mathbf{2\times}$ less than the previous state-of-the-art `Local Adam`, enabling $\mathbf{1.3}$–$\mathbf{2.1\times}$ wall-clock speedups over `DDP` for 1-13B models on 100Gb/s links. Furthermore, unlike previous heuristic methods, `DES-LOC` is robust to worker failures offering a scalable, efficient, and fault-tolerant solution for foundation model training.

## 1 INTRODUCTION

Training foundation models requires distributing optimization across workers for improved memory and compute. However, frequent gradient communication in standard Distributed Data Parallelism (`DDP`) (Li et al., 2020a) increases networking costs and limits scalability. Early works like `Local SGD` (Stich, 2019) and `FedAvg` (McMahan et al., 2017) reduced this overhead by synchronizing infrequently, averaging parameters only after $K \gg 1$ local steps, instead of gradients at every step. However, modern foundation model training, e.g., Large Language Models (Dubey et al., 2024), uses **adaptive optimizers** (Kingma & Ba, 2015) which require additional momenta.

Some extensions of `Local SGD` to adaptive optimizers (Sani et al., 2025; Douillard et al., 2023) only average model parameters, which poses challenges. First, they lack convergence guarantees. Second, keeping momenta local (Douillard et al., 2023) accumulates noisy small-batch gradients and provides no means to initialize workers. This makes them unsuitable for failure-prone environments. Third, re-initializing momenta (Sani et al., 2024; 2025) destabilizes training.

`Local Adam` (Cheng & Glasgow, 2025) addresses these challenges, proving periodic synchronization *can* converge faster than standard `Adam` with `DDP`, and remain **robust** to the addition of new workers. However, it requires synchronizing momenta alongside model parameters, tripling communication costs compared to `Local SGD`. Hence, we aim to answer the following question:

[†]`aai30@cam.ac.uk`; [*]Equal contributions; [1]University of Cambridge; [2]Flower Labs; [3]Institute of Science and Technology Austria; [4]University of Warwick; [5]Mohamed bin Zayed University of AI, [6]IMEC

*Can independently syncing parameters and momenta improve communication*
*efficiency for adaptive optimizers while maintaining convergence and robustness?*

As a result of our inquiry, we propose a new optimizer family, Desynced Low Communication Adaptive Optimizers (`DES-LOC`), which sets independent synchronization frequencies for parameters and momenta. This reduces communication overhead by synchronizing momenta less frequently.

---

**Contributions :**

1. **Provable convergence.** We prove convergence (see Section 3) for `DES-LOC` under: non-convex objectives when using SGD with momentum (`SGDM`), and weakly convex objectives when using `Adam`. Our theory indicates a higher momentum sync frequency enables larger step sizes. Furthermore, high-probability bounds demand momenta be synced with finite period for $\beta_2 < 1.0$.

2. **Communication reduction.** We empirically show that parameters require more frequent sync than momenta, and that less frequent momentum sync reduces communication costs ($\mathbf{2}\times$ vs `Local Adam`, $\mathbf{170}\times$ vs `DDP`), leading to $1.3 - 2.1\times$ reductions in training time over `DDP` on our hardware.

3. **Scalability to large models.** We validate `DES-LOC` at billion-scale language model training, demonstrating competitive `ICL` performance against both `Local Adam` and `DDP`.

4. **Hardware robustness.** Unlike previous heuristic methods, `DES-LOC` avoids persistent local states, enabling it to seamlessly integrate new workers to support environments prone to system failures.

---

## 2 DESYNCED LOW COMMUNICATION ADAPTIVE OPTIMIZERS (`DES-LOC`)

We start by characterizing the relation between the rate of change of optimizer states and `Local Adam`, and how these can be leveraged to lower the communication cost. Consider the `Adam` update: $u_t = \beta_1 u_{t-1} + (1 - \beta_1)g_t$ and $v_t = \beta_2 v_{t-1} + (1 - \beta_2)g_t \odot g_t$.

For `Local Adam`, convergence is contingent on $\beta_2$ satisfying $1 - \beta_2 = \widetilde{\mathcal{O}}\big(K^{-3/2}R^{-1/2}\big)$ (Cheng & Glasgow, 2025) where $K$ is the number of local steps and $R$ the total communication rounds. Large $K$ or $R$, implies $\beta_2 \to 1$, and conversely larger $\beta_2$ permits higher $K$ or $R$.

A useful summary measure is the number of steps until a state's weight decays to a fraction $\psi$, $\tau_\psi(\beta) = \frac{\ln \psi}{\ln \beta}$. Following Pagliardini et al. (2025), we use the half-life $\tau_{0.5}$ as our primary measure, omitting $\beta$ when clear. For typical values of $\beta$, we have $\tau_{0.5}(0.95) \approx 13.5$ (Allal et al., 2025), $\tau_{0.5}(0.999) \approx 692.8$ (Kingma & Ba, 2015), and $\tau_{0.5}(0.9999) \approx 6931$ (Taniguchi et al., 2024). Intuitively, larger half-lives imply synchronizing gradients over longer horizons as the optimizer is less sensitive to new gradients; choosing $\beta = 0$ ignores all previous momenta, whereas $\beta \to 1$ progressively attenuates signal from the current gradient.

While the half-life captures the horizon for which an optimizer state remains relevant to model updates, it provides no information on its absolute rate of change. With coordinate-wise clipping, each gradient component satisfies $|(g_t)_i| \le \rho$. Unrolling `Adam`'s recursions over $K$ local steps gives the follow relation: $u_{t+K} = \beta_1^K u_t + (1 - \beta_1)\sum_{k=0}^{K-1} \beta_1^k g_{t+K-1-k}$ and its second moment analogue. Since $|g_{t,i}| \le \rho$ and $|(g_t \odot g_t)_i| \le \rho^2$, the maximal $\ell_\infty$ drift of each moment is (see Section F):

$$\left\|u_{t+K} - u_t\right\|_\infty \le 2\rho\left(1 - \beta_1^K\right), \tag{1}$$

$$\left\|v_{t+K} - v_t\right\|_\infty \le 2\rho^2\left(1 - \beta_2^K\right). \tag{2}$$

From the above, large $\beta$ values and small clip bounds $\rho$, a common practice in foundation model training (Brown et al., 2020; Scao et al., 2022), limit the absolute changes in optimizer states. We can construct similar reasoning for other optimizers (Sutskever et al., 2013; Taniguchi et al., 2024), and norm-based clipping (Pascanu et al., 2013; Brown et al., 2020). From the above, the half-life of an optimizer state should inform its synchronization frequency. For example, if $\tau_{0.5}(0.95) \approx 13.5$ and $K = 256$, synchronization only affects few initial local steps. Over the course of the local training, the impact of the synchronised optimizer state shall decay to $0$ given Equations 1 and 2. Conversely, if $K = 16$, synchronization approximately matches the half-life, strongly influencing local updates.

## 2.1 DES-LOC ALGORITHM

---

**Algorithm 1** DES-LOC

---

**Require: Model tensors, update functions, hyper-parameters**
1:   $x_0 \in \mathbb{R}^d, \{s_{-1}^j\}_{j=1}^N \in (\mathbb{R}^d)^N$ — initial parameter vector, the initial $N$ optimizer states
2:   $\{\text{UPDATE}^j\}_{j=1}^N : (\mathbb{R}^d \times \mathbb{R}^d \to \mathbb{R}^d)^N$ — updates optimizer state $j$ from its previous state and the gradient.
3:   $\text{OPT} : \mathbb{R}^d \times \mathbb{R}^d \times \mathbb{R}_+ \times (\mathbb{R}^d)^N \to \mathbb{R}^d$ — update params from all worker models.
4:   $\text{SERVEROPT} : \mathbb{R}^d \to \mathbb{R}^d$ — update params using an abstract outer optimizer
5:   $\rho \in \mathbb{R}_+, \{\eta_t\}_{t=0}^{T-1} \in (\mathbb{R}_+)^{T-1}$ — clipping radius for $\textbf{clip}(\cdot, \rho)$, learning-rate for each time-step
6:   $T, M \in \mathbb{N}_+$ — total optimization steps and number of workers
7:   $K_x \in \mathbb{N}_+, \{K_j\}_{j=1}^N \in (\mathbb{N}_+)^N$ — communication periods (steps)
**Ensure:** $x_T, \{s_{T-1}^j\}_{j=1}^N$
8:   **for each worker** $m$: $x_0^m \leftarrow x_0, s_{-1}^{j,m} \leftarrow s_{-1}^j$ ⟶ *local init*
9:   **for** $t = 0, \ldots, T-1$ **do** ⟶ *training loop*
10:      **for all** workers $m = 0, \ldots, M-1$ **in parallel do**
11:          $g_t^m \leftarrow \nabla F(x_t^m; \xi_t^m)$ ⟶ *stochastic grad*
12:          $\widehat{g}_t^m \leftarrow \textbf{clip}(g_t^m, \rho)$ ⟶ *per-coordinate clipping*
13:          **for** $j = 1$ **to** $N$ **do**
14:              **if** $t \bmod K_j = 0$ **then** ⟶ *sync $s^j$*
15:                  $s_t^{j,m} \leftarrow \text{UPDATE}^j\big(\mathbb{E}_m[s_{t-1}^{j,m}], \widehat{g}_t^m\big)$
16:              **else**
17:                  $s_t^{j,m} \leftarrow \text{UPDATE}^j\big(s_{t-1}^{j,m}, \widehat{g}_t^m\big)$
18:          **if** $t \bmod K_x = 0$ **then** ⟶ *sync $x$*
19:              $x_{t+1}^m \leftarrow \text{OPT}\big(\text{SERVEROPT}(\mathbb{E}_m[x_t^m]), \widehat{g}_t^m, \eta_t, \{s_t^{j,m}\}_{j=1}^N\big)$
20:          **else**
21:              $x_{t+1}^m \leftarrow \text{OPT}\big(x_t^m, \widehat{g}_t^m, \eta_t, \{s_t^{j,m}\}_{j=1}^N\big)$

---

Motivated by the above insights, we formalize Desynced Low Communication Adaptive Optimizers as a family of optimizers offering the same convergence and robustness as Local Adam but with significantly lower communication costs. Our approach applies generically to adaptive optimizers parameterized by $\text{OPT} : (\mathbb{R}^d, \mathbb{R}^d, \mathbb{R}_{>0}, \{\mathbb{R}^d\}^N) \to \mathbb{R}^d$, with $N$ optimizer states $\{s_{-1}^j\}_{j=1}^N \subset \mathbb{R}^d$, each updated by $\text{UPDATE}^j : (\mathbb{R}^d, \mathbb{R}^d) \to \mathbb{R}^d$. Coordinate-wise clipping is defined as $[\textbf{clip}(X, \rho)]_i = \text{sgn}(X_i) \cdot \min\{|X_i|, \rho\}$. To ensure that our method is provably convergent, SERVEROPT is that of FedAvg (McMahan et al., 2017). However, our algorithm directly extends to the larger FedOpt (Reddi et al., 2021) framework, which we discuss in Section C.1.

We focus our analysis on SGDM and Adam. As shown in Algorithm 1, DES-LOC synchronizes parameters $x \in \mathbb{R}^d$ and optimizer states $\{s^j\}_{j=1}^N$ at state-specific intervals $K_x, \{K_j\}_{j=1}^N \in \mathbb{N}_+$. Setting $N = 2$, $s_t^1 = u_t$, $s_t^2 = v_t$, and using update rules $\text{UPDATE}^1, \text{UPDATE}^2$ based on the Adam update rules above yields DES-LOC-Adam (see Algorithm 2).

> **Toy Example** To highlight DES-LOC's practical benefit, Fig. 1 illustrates a scenario where DES-LOC and Local Adam converge under noisy gradients, while prior heuristic methods (Douillard et al., 2023; Sani et al., 2025; Iacob et al., 2025; Sani et al., 2024) fail.

## 3 CONVERGENCE GUARANTEES FOR DES-LOC

This section provides theoretical support for the proposed DES-LOC approach. We focus on a version of the Adam optimizer that uses only a single momentum state. Extensions to the full Adam optimizer with both momenta are available in Section D.1 with high-probability bounds shown in Section E.

Formally, we consider the following optimization problem:

$$\min_{x \in \mathbb{R}^d} f(x) := \frac{1}{M} \sum_{m=1}^M f_m(x), \quad \text{with} \quad f_m(x) = \mathbb{E}_{\xi \sim \mathcal{D}_m}[F_m(x; \xi)]. \tag{3}$$

In this setup, all $M$ machines collaboratively minimize the objective in (3). Generally, we assume each machine $m$ has access to only dataset $\mathcal{D}_m$, which can differ from device to device. This recovers the homogeneous distribution case when all machines have the same dataset $\mathcal{D}_1 = \mathcal{D}_2 = \cdots = \mathcal{D}_M$

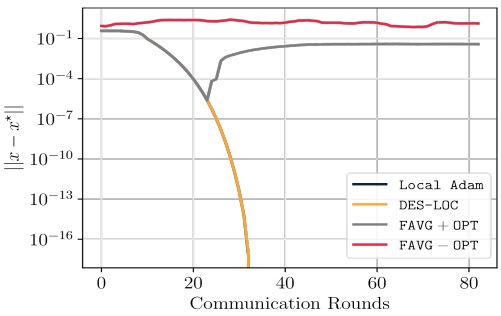 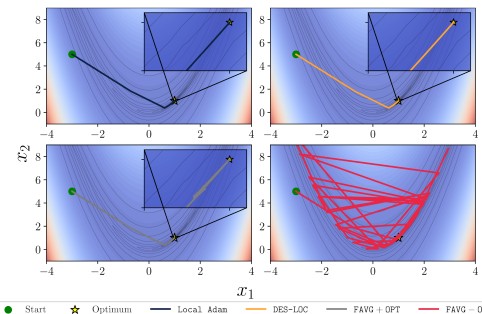

Figure 1: We present: (left) the distance to the optimum and (right) a 2-D contour of a toy problem where DES-LOC ($K_x = 192, K_u = 192, K_v = 692$) and Local Adam ($K = K_x$) both converge to the optimum (overlapping). Methods keeping optimizer states local (■) fail to converge. Periodically resetting states (■) similarly stalls due to repeated oscillations. We optimize the non-convex function $f(x_1, x_2) = (1 - x_1)^2 + 100(x_2 - x_1^2)^2$ with $M = 256$ workers and IID Gaussian noise ($\sigma = 1.5$). We show an example of such a toy problem on Non-IID data in Fig. 8.

and minimize the same loss $f_1(x) = f_2(x) = \cdots = f_m(x) = f(x)$. We assume each machine $m$ computes mini-batch stochastic gradients corresponding to randomly selected samples $\xi \sim \mathcal{D}_m$ from dataset $\mathcal{D}_m$. We further use the following technical assumptions on the problem structure.

**Assumption 1** (Lower bound and smoothness). *The overall loss function $f: \mathbb{R}^d \to \mathbb{R}$ is lower bounded by some $f^* \in \mathbb{R}$ and all local loss functions $f_m$ are L-smooth:*

$$\|\nabla f_m(x) - \nabla f_m(y)\| \leq L\|x - y\|, \quad \text{for any } x, y \in \mathbb{R}^d.$$

**Assumption 2** (Unbiased noise with bounded stochastic variance). *The stochastic gradient $g^m$ of local loss function $f_m$ computed by machine $m$ is unbiased and the noise has bounded variance:*

$$\mathbb{E}[g^m] = \nabla f_m(x), \quad \mathbb{E}[\|g_t^m - \nabla f_m(x)\|^2] \leq \sigma^2, \quad \text{for any } x \in \mathbb{R}^d.$$

**Assumption 3** (Bounded heterogeneity). *For any $x \in \mathbb{R}^d$, the heterogeneity is bounded by*

$$\frac{1}{M} \sum_{m=1}^{M} \|\nabla f_m(x)\|^2 \leq G^2 + B^2 \|\nabla f(x)\|^2.$$

All three assumptions are standard and widely used in the convergence analysis of optimization algorithms Yu et al. (2019); Karimireddy et al. (2020b); Wang et al. (2021); Yuan et al. (2022). Note that the bounded heterogeneity condition recovers the homogeneous case when $G^2 = 0$ and $B^2 = 1$. To facilitate the technical presentation of the analysis, we view model and optimizer state synchronizations through assigning probabilities to each averaging event. Particularly, instead of averaging model parameters every $K_x$ steps (i.e., $t \bmod K_x = 0$), we average with probability $p_x = \frac{1}{K_x}$, which are statistically equivalent. In the following theorem, we provide convergence rate of SGDM optimizer under such probabilistic and decoupled synchronization:

**Theorem 1.** *Let Assumptions 1, 2 and 3 hold. Then, choosing the step size $\eta = \min(\eta_0, \frac{1}{\sqrt{T}})$ with*

$$\eta_0 \stackrel{\text{def}}{=} \frac{1}{4L} \min\left(1 - \beta, \frac{1}{6\sqrt{\psi \max(1, B^2 - 1)}}\right), \quad \text{where} \quad \psi \stackrel{\text{def}}{=} \frac{4(1 - p_x)}{p_x^2} \cdot \frac{(1 - \beta)(1 - p_u)}{1 - (1 - p_u)\beta}, \quad (4)$$

*the average iterates $x_t = \mathbb{E}_m[x_t^m]$ of DES-LOC-SGDM converge with the following rate:*

$$\frac{1}{T} \sum_{t=0}^{T-1} \mathbb{E}\|\nabla f(x_t)\|^2 \leq \frac{4}{\sqrt{T}}\left(f(x_0) - f^* + \frac{L\sigma^2}{2M}\right) + \mathcal{O}\left(\frac{1 + \psi}{T}\right). \quad (5)$$

We now discuss the convergence result and its implications. The obtained rate (5) is asymptotically optimal for this setup (Arjevani et al., 2023). Notably, the leading term $\mathcal{O}(\frac{1}{\sqrt{T}})$ is unaffected by the number of local steps. Interestingly, probabilities $p_x, p_u$, and the momentum parameter $\beta$ appear in the higher-order term $\mathcal{O}(\frac{1}{T})$, and thus have a limited impact on asymptotic convergence speed.

Regarding state synchronization, it is evident from (4) that model synchronization has a greater impact on convergence due to the dependence $\psi = \mathcal{O}(\frac{1}{p_x^2})$. With vanishing $p_x$, the $\psi$ term becomes

unbounded and breaks the rate. For optimizer states, it seems that momentum averaging can be turned off ($p_u = 0$) without affecting the asymptotic behavior of the rate. Setting $p_x = 1$ and $p_u = 0$ recovers standard mini-batch SGDM (Liu et al., 2020). However, the $\psi$ term also appears in the step-size restriction (4). As $p_u \to 0$, $\frac{(1-p_u)}{1-(1-p_u)\beta} \to \frac{1}{1-\beta}$. This imposes the most severe restriction on the learning rate $\eta_0$ since $\eta_0 \propto \frac{1}{\sqrt{\psi}}$, as $\psi$ is maximized. This theory shows that increasing the frequency $p_u$ of momentum averaging—while not changing the asymptotic rate—allows for a larger step size, potentially leading to faster convergence in practice. This theory justifies that momentum states can be synchronized less frequently than parameters and that more averaging improves convergence by supporting larger step sizes. Furthermore, our high probability analysis of `DES-LOC-Adam` in Section E shows that the sync frequency of momenta must be finite for $\beta_2 < 1.0$.

## 4 EXPERIMENTAL DESIGN

Our experimental setup addresses the following research questions:

**RQ1** Do *theoretical* rates of change predict the *empirical* evolution of optimizer states?
**RQ2** How does the synchronization frequency of a model/optimizer state impact performance?
**RQ3** To what extent can `DES-LOC` cut communication w.r.t. `Local Adam` in practical scenarios?
**RQ4** How does `DES-LOC` scale with increasing model size and longer training horizons?
**RQ5** How does `DES-LOC` perform when using a `Nesterov` outer optimizer?
**RQ6** Is `DES-LOC` compatible with non-`Adam` inner optimizers such as `Muon`?

### 4.1 EXPERIMENTAL SETUP

**Models and data.** We train a 135M-parameter `GPT`-style model (arch. in Table 3) with sequence length 2048. We distinguish worker batch size $\mathcal{B}_w$ from global $\mathcal{B} = \sum_{w=0}^{M-1} \mathcal{B}_w$ (Sani et al., 2025). A 2M token global batch is split across $M = 4$ workers sampling `IID` from `SmolLM2` (Allal et al., 2025): 70% `Fineweb-Edu` (Penedo et al., 2024), 10% `Cosmopedia` (Ben Allal et al., 2024), 10% `Python-Edu`, 5% `FineMath 4+`, and 5% `Infi-WebMath 4+`. The 135M model trains for 6.4B tokens (2.4× compute-optimal (Hoffmann et al., 2022)). For **RQ4**, we scale to a 1.7B model for 40B tokens (2× compute-optimal) (Sardana et al., 2024). We show `Non-IID` data results in Fig. 14.

**Optimizers.** We use `Adam` (Kingma & Ba, 2015) (results in Section B) and its variant `ADOPT` (Taniguchi et al., 2024), which modifies the update to guarantee convergence for any $\beta_2$. For the 135M model, we grid-search $(\beta_1, \beta_2, \eta)$ under `DDP`; the 1.7B model uses hyperparameters from Allal et al. (2025); Taniguchi et al. (2024). Learning rates use the `WSD` schedule (Hägele et al., 2024; Allal et al., 2025). We favor `ADOPT` ($\beta_2 = 0.9999$) in high-$\beta$ regimes where `Adam` is unstable. We also ablate the outer optimizer, comparing `FedAvg` with a `Nesterov` optimizer (Reddi et al., 2021; Douillard et al., 2023; Charles et al., 2025) on a 700M model trained on 40B tokens.

**Baselines.** We compare `DES-LOC` with: (i) synchronous `DDP`; (ii) Local `Adam`/`ADOPT`; (iii) `FAVG+OPT` (persistent states (Sani et al., 2025; Douillard et al., 2023)); and (iv) `FAVG−OPT` (reset states (Sani et al., 2024; Iacob et al., 2025)). Persistent-state `FedAvg` is `DES-LOC` with $K_u, K_v = \infty$, an upper bound on comms efficiency. `DDP` is an upper bound on ML performance.

**Metrics.** We evaluate models by (i) perplexity and (ii) per-worker asymptotic communication cost assuming a bandwidth-optimal `Ring-AllReduce` (Sergeev & Balso, 2018) algorithm scaling linearly with model size. For the 1.7B model, we report standard in-context-learning (`ICL`) benchmarks (Brown et al., 2020). We use a zero-shot setting for `ICL` tasks unless stated otherwise following Allal et al. (2025) and report the best performing communication-efficient method in blue with the best-performing overall in **bold**. To fairly compare optimizer-state changes across decay rates, we measure their *relative* rates of change as $\|s_{t+K} - s_t\|_2/\|s_t\|_2$. For convergence plots, we report final-round means and standard deviations next to labels. We also provide wall-time clock results; we use 4 machines with one H100 for sub-1B models, and 4 machines with 8 H100s each for larger scales. While the links between machines run at 100Gb/s, we observed overheads limiting the practical bandwidth to $60 - 70$ Gb/s. We report stepwise (see Section B.3.1) and timewise convergence. We also provide an analysis on the wall-clock time vs bandwidth in Section G.1.

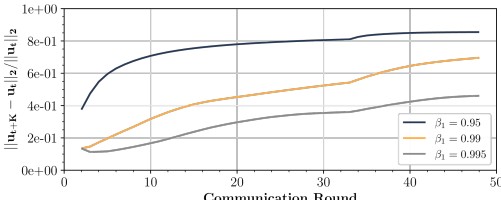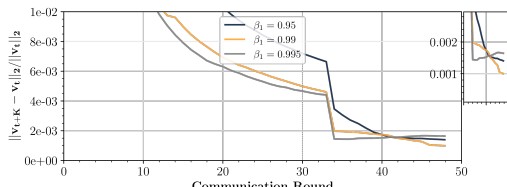

Figure 2: Relative rates of change for first (left) and second (right) momenta across rounds using standard `Local ADOPT` ($K = 64$). For `ADOPT` ($\beta_2 = 0.9999$), increasing $\beta_1 \geq 0.99$ greatly slows the first-momentum rate of change. The second momentum evolves $\sim 100\times$ slower.

## 5  EVALUATION

Our results show optimizer states change at different rates (Section 5.1), forming a clear synchronization hierarchy (Section 5.2). `DES-LOC` reduces communication $2\times$ vs. `Local Adam` (Section 5.3) while converging robustly with adding workers and scaling effectively to large models (Section 5.4).

### 5.1  HIGHER $\beta$ OPTIMIZER STATES HAVE SLOWER EMPIRICAL RATES OF CHANGE (RQ1)

Figure 2 shows that relative rates (of change for the two momenta in `Local ADOPT`/`Adam` scale with their decay rates under gradient clipping ($\rho = 1$). Supported by our theoretical discussions on momenta half-lives (Section 2), the second momentum evolves substantially slower than the first at high-$\beta_2$. For `Local Adam`, the second momentum remains slower even when $\beta_2 \approx \beta_1$, potentially because gradient variance (Kingma & Ba, 2015) evolves slower than the first momentum.

> **Takeaway:** As discussed in Sections 2 and 3, when $\beta_1 \ll \beta_2$, the second momentum evolves slower than the first, proportional to half-life ratio of the two $\frac{\tau_{0.5}(\beta_2)}{\tau_{0.5}(\beta_1)} = \frac{\ln(\beta_1)}{\ln(\beta_2)}$.

### 5.2  PARAMETERS REQUIRE FREQUENT SYNC, MOMENTA SYNC PROPORTIONAL TO $\beta$ (RQ2)

Figure 3 evaluates the effect of independently varying synchronization periods ($K_x, K_u, K_v$) for parameters and optimizer states. We consider two baseline periods ($K_b = 16, 256$), chosen based on the fastest state's half-life ($\tau_{0.5}(0.95) \approx 13.5$). Frequent parameter synchronization ($K_x$) is crucial for performance, while synchronizing momenta ($K_u, K_v$) significantly impacts training only if their half-lives align with the base frequency $K_b$. Otherwise, synchronization frequency primarily influences communication costs rather than model quality. `Adam` results can be seen in Section B.

> **Takeaway:** Parameter synchronization frequency ($K_x$) strongly impacts performance, motivated by the leading term in theoretical bounds (Section 3). Momentum synchronization periods matter empirically only when chosen near their half-lives, consistent with Sections 2 and 3.

### 5.3  `DES-LOC` BRINGS $2\times$ COMMUNICATION REDUCTIONS OVER `LOCAL ADAM` (RQ3)

As shown in Figure 4, `DES-LOC` halves communication versus `Local Adam` (Cheng & Glasgow, 2025) with matching perplexity by syncing momenta less frequently ($K_u = 3K_x, K_v = 6K_x$), exploiting the second-momentum's lower sensitivity to sync frequency (Fig. 3). This yields a $1.37\times$ speedup over `DDP` and $1.1\times$ over `Local Adam` at $K_x = 16$ on 4 H100s. At $K_x = 256$, the speedup over `DDP` increases to $1.47\times$ and both `DES-LOC` and `Local Adam` saturate throughput.

As shown in Figure 4, `DES-LOC` effectively initializes new workers and outperforms the straightforward approach of ad-hoc averaging from checkpoints—whose insufficiency is detailed in Section H.

> **Takeaway:** `DES-LOC` achieves a $2\times$ communication reduction over `Local Adam` by leveraging two insights: optimizer-state sync matters less than parameter sync, and slower-changing states (high $\beta_2$) can sync less often. By eventually syncing all optimizer states, `DES-LOC` matches the robustness of `Local Adam` with $K = \max(K_x, K_u, K_v)$ when adding new workers/responding to system failures.

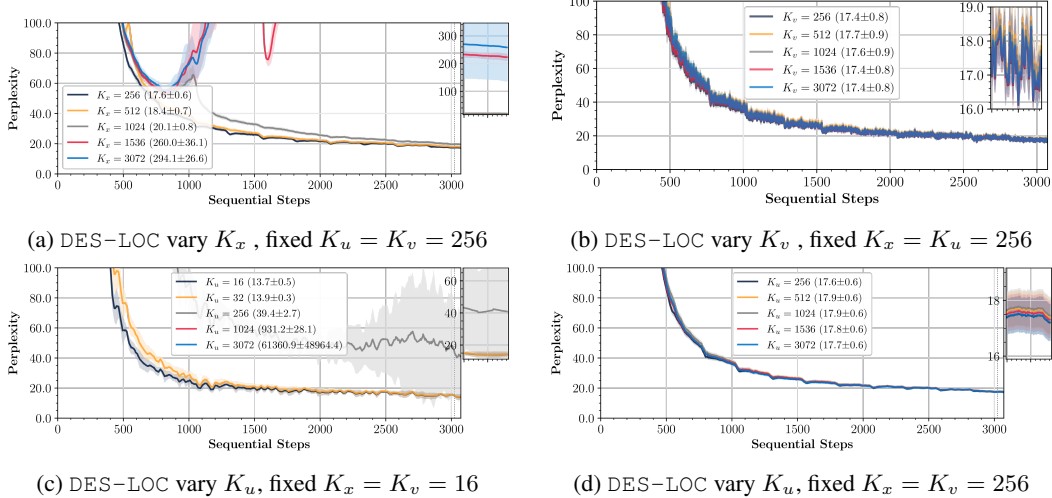

(a) DES-LOC vary $K_x$ , fixed $K_u = K_v = 256$

(b) DES-LOC vary $K_v$ , fixed $K_x = K_u = 256$

(c) DES-LOC vary $K_u$, fixed $K_x = K_v = 16$

(d) DES-LOC vary $K_u$, fixed $K_x = K_v = 256$

Figure 3: Model perplexity for DES-LOC (ADOPT, $\beta_1 = 0.95, \beta_2 = 0.9999$), varying synchronization periods independently (others fixed at $K_b$). Parameter synchronization (a) is critical, with degradation at higher periods. Second-momentum sync (b) minimally affects performance due to its large half-life. First-momentum sync improves perplexity when the freq matches its half-life; contrast (c) and (d). Parameters and second momentum behave similarly across frequencies (Section B)

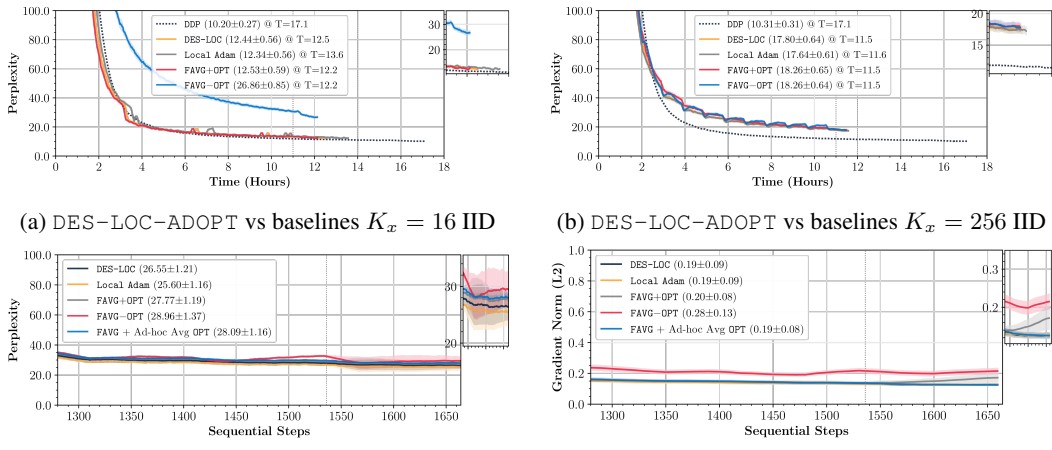

(a) DES-LOC-ADOPT vs baselines $K_x = 16$ IID

(b) DES-LOC-ADOPT vs baselines $K_x = 256$ IID

(c) Perplexity impact of doubling workers, $K_x = 128$.

(d) Gradient norms after doubling workers, $K_x = 128$.

Figure 4: DES-LOC ($K_u, K_v = 3K_x, 6K_x$) reduces comms by $\mathbf{2}\times$ over Local Adam while matching its performance and that of heuristic baselines at high (a) and low (b) frequencies (Section 4.1). We show robustness by doubling worker count at step 1536 (c,d), where DES-LOC and Local Adam maintain stable perplexity/norms, outperforming heuristics and ad-hoc optimizer-state averaging.

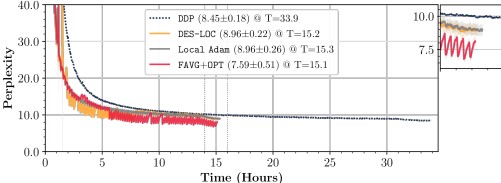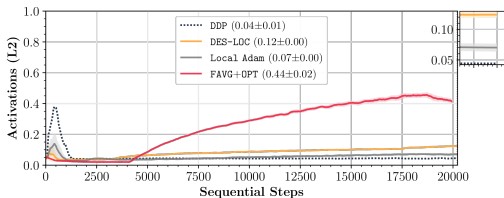

Figure 5: `DES-LOC` matches `Local Adam` perplexity (left) for billion-scale model training at half the communication cost ($K_x = 256, K_u = 3K_x, K_v = 6K_x$), representing a **170×** reduction over `DDP`. Both `DES-LOC` and `Local Adam` converge to competitive perplexity at this scale. `FAVG+OPT` achieves good performance (left) but suffers activation growth (right) and parameter-norm growth (Section B), potentially due to noisy updates, raising concerns for extended training.

Table 1: Our billion-scale model trained with `DES-LOC` matches or surpasses the (`ICL`) performance of models trained with `Local Adam` and `FAVG+OPT`, approaching `DDP` performance. `FAVG+OPT` underperforms compared to its perplexity results from Fig. 5.a, indicating that the activation increases (Fig. 5.b) from the unstable training procedure may have damaged the model.

|  | Arc Challenge | Arc Easy | PIQA | HellaSwag | Avg |
|---|---|---|---|---|---|
| DES-LOC | 31.8 | 59.0 | 70.7 | 44.9 | 51.6 |
| Local Adam | 31.9 | 59.0 | 70.6 | 45.8 | 51.8 |
| FAVG+OPT | 30.1 | 58.0 | 70.0 | 44.8 | 50.7 |
| DDP | **33.8** | **62.5** | **71.1** | **47.8** | **53.8** |

## 5.4 `DES-LOC` IS SUITABLE FOR LARGE-SCALE TRAINING (RQ4)

Figure 5 shows that `DES-LOC` reliably scales to 1B models and very infrequent communication ($K_x = 256$). Evaluating the billion-scale models on the `ICL` tasks (Table 1), `DES-LOC` is competitive with all baselines while reducing communication versus `Local Adam` and `DDP`. The heuristic baseline (Sani et al., 2025) suffers training instabilities (Fig. 5.b) potentially impacting downstream performance (Table 1) and underscoring the advantage of `DES-LOC`'s training stability. We elaborate more on the settings in which we expect `DES-LOC` to improve stability in Section J.

Our method's reduced communication costs result in a $\approx 2.2\times$ training speedup over `DDP` (Figure 5). As show by our benchmarks Table 2, these time savings scale with model size and comms frequency. At the 13B scale with $K_x = 16$, `DES-LOC` would save 13 days over `Local Adam` and 73 days over `DDP`. The advantage over `DDP` widens for $K_x = 256$, where communication-efficient methods maximize throughput. Table 5 shows equivalent results for throughput.

> **Takeaway:** `DES-LOC` enables efficient training of large-scale foundation models, especially at long training horizons, with downstream `ICL` performance competitive with `DDP`. We recommend setting $K_x$ for sufficient throughput based on bandwidth, then setting $K_u, k_v$ as constant multiples (e.g, $3\times, 6\times$) or based on the half-life of their $\beta$ (see Section I).

Table 2: Wall-clock time (days) for 1B-13B models to reach $2\times$ compute-optimal tokens at high ($K = 16$) and low ($K = 256$) frequencies with a 2M token batch size. At high frequency ($K = 16$), `DES-LOC` outperforms Local ADAM by over 13 days on the 13B model and is within 3% of `FAVG+OPT`. At low frequency ($K = 256$), it cuts the 13B's training time $> 93$ days versus `DDP`.

|  | 1B Model | | 7B Model | | 13B Model | |
|---|---|---|---|---|---|---|
| $K_x$ | 16 | 256 | 16 | 256 | 16 | 256 |
| DDP (Baseline) | $1.41 \pm 0.008$ | $1.41 \pm 0.008$ | $38.74 \pm 0.161$ | $38.74 \pm 0.161$ | $175.50 \pm 0.478$ | $175.50 \pm 0.478$ |
| FAVG+OPT | $0.80 \pm 0.007$ | $0.63 \pm 0.006$ | $28.52 \pm 0.095$ | $24.01 \pm 0.088$ | $100.21 \pm 0.544$ | $82.46 \pm 0.513$ |
| Local Adam | $0.96 \pm 0.006$ | $0.64 \pm 0.006$ | $31.46 \pm 0.090$ | $24.18 \pm 0.087$ | $116.10 \pm 0.484$ | $83.34 \pm 0.509$ |
| DES-LOC ($K_u, K_v = 3K_x, 6K_x$) | $0.81 \pm 0.006$ | $0.63 \pm 0.006$ | $28.80 \pm 0.094$ | $24.06 \pm 0.088$ | $102.03 \pm 0.537$ | $82.68 \pm 0.512$ |

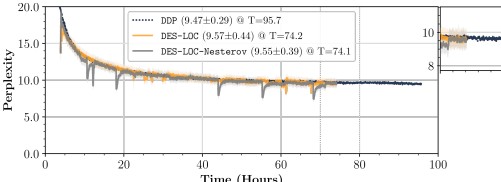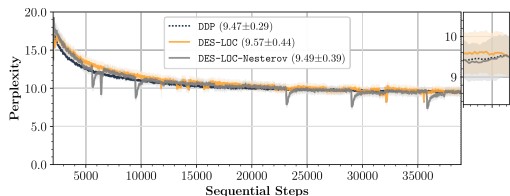

Figure 6: Ablation of the outer optimizer for `DES-LOC` on a 700M parameter model in a medium-frequency communication setting ($K_x = 32$), showing (left) convergence in terms of time and (right) in terms of steps. In this regime, `DES-LOC`'s final perplexity is within $1\%$ of the `DDP` baseline. Using a `Nesterov` outer optimizer provides a further improvement over averaging.

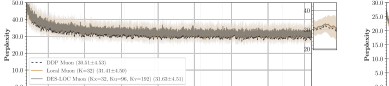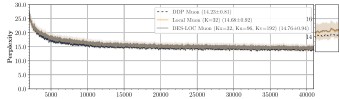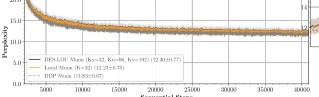

Figure 7: Training loss comparison between `Local Muon` ($K = 32$) and `DES-LOC`-`Muon` ($K_x = 32, K_u = 96$) across model scales (16M, 125M, 360M). `DES-LOC` matches the Local Muon baseline across all scales while communicating more than $1.5\times$ fewer bytes. See Section B.7 for full experimental details.

## 5.5 NESTEROV AS THE OUTER OPTIMIZER (RQ5)

We ablate the outer optimizer for `DES-LOC` on a 700M parameter model, comparing averaging to a `Nesterov` optimizer with momentum of 0.9, outer learning rate of 1.0 tuned following Charles et al. (2025). The experiment ran on 4 H100s and used a medium-synchronization regime ($K_x = 32, K_u = 3K_x, K_v = 6K_x$) where models are initialised from 2048-step `DDP` checkpoints, following Charles et al. (2025). While our convergence bound is not trivially applicable, our analysis of Eq. (4) suggests a higher momentum synchronization frequency ($p_u$) should permit a larger step size ($\eta_0 \propto 1/\sqrt{\psi}$).

As shown in Figure 6, two key points emerge. First, more frequent synchronization ($K_x = 32$) allows `DES-LOC` to come within $1\%$ of the final perplexity of `DDP`, performing much better than in infrequent settings ($K_x = 256$). Second, using `Nesterov` as the outer optimizer improves performance over averaging by $\approx 0.5\%$, with its performance w.r.t `DDP` being similar to the one reported in Charles et al. (2025, Table 4) for models at this scale. The `Nesterov` approach preserves the practical benefits of `DES-LOC`, ensuring effective worker initialization and reducing local optimization noise, which can help prevent issues like exploding activation norms.

> **Takeaway:** Frequent synchronization ($K_x = 32$) allows `DES-LOC` to reach within $1\%$ of the perplexity of `DDP`. Furthermore, using a `Nesterov` outer optimizer improves performance over averaging, while preserving practical benefits like effective worker initialization and reduced optimization noise.

We further investigate the benefits of synchronizing optimizer states under a `Nesterov` outer optimizer relative to purely local-state methods (`DiLoCo`) at the 1B scale in Section B.5.

## 5.6 MUON AS THE INNER OPTIMIZER (RQ6)

To assess the versatility of `DES-LOC` beyond `Adam`-family optimizers, we integrate it with `Muon` (Jordan et al., 2024), a single-momentum optimizer that uses Newton-Schulz iterations for orthogonalization. Since `Muon` preconditions the momentum term directly rather than tracking second-moment estimates, the relevant synchronization periods reduce to parameters ($K_x$) and momentum ($K_u$). We apply `DES-LOC` by synchronizing the `Muon` momentum buffer less frequently than the parameters, setting $K_u = 3K_x$ with $K_x = 32$.

As shown in Figure 7, `DES-LOC` matches the perplexity of the `Local Muon` baseline across model scales ranging from 16M to 360M parameters. By decoupling the synchronization frequencies, `DES-LOC` communicates more than $1.5\times$ fewer bytes than the baseline, confirming that the Half-Life Principle extends to optimizers beyond the `Adam` family. Full experimental details in Section B.7.

> **Takeaway:** `DES-LOC` is compatible with single-momentum optimizers such as `Muon` that rely on Newton-Schulz preconditioning. By reducing the synchronization frequency of the momentum buffer relative to parameters, `DES-LOC` maintains solution quality while significantly lowering communication volume, demonstrating that our approach is inner-optimizer-agnostic.

## 6 RELATED WORK

In synchronous data-parallel training, workers exchange full gradients or parameters *every* iteration, incurring communication costs linear in model size using `Ring-AllReduce` (Sergeev & Balso, 2018). When hardware is weakly connected or widely distributed, communication significantly slows wall-clock training time (Sani et al., 2025) as workers need to wait for synchronization to finish. `Federated Averaging` (FedAvg) (McMahan et al., 2017) and `Local SGD` (Stich, 2019) reduce communication by performing $K$ local optimization steps before averaging parameters, decreasing communication rounds by a factor of $K$. Ad-hoc extensions to adaptive optimizers either keep optimizer states local (Douillard et al., 2023; Charles et al., 2025; Liu et al., 2024) or reset them after each sync (Sani et al., 2024; 2025), both lacking robust convergence guarantees.

`Adam` (Kingma & Ba, 2015) is popular for pre-training as it scales to larger batches than `SGD` (Kunstner et al., 2023; Dubey et al., 2024). It uses moving averages of gradients and their squares, however, its convergence is not guaranteed as it requires $\beta_1 < \sqrt{\beta_2} < 1$, with large, problem-specific $\beta_2$ (Reddi et al., 2018; Zhang et al., 2022). Other optimizers also track gradient moments (Sutskever et al., 2013; Chen et al., 2023; You et al., 2020; Taniguchi et al., 2024). *Local Adam* (Cheng & Glasgow, 2025) reduces communication with local steps but requires syncing optimizer states, which triples the communication cost relative to `Local SGD`/`DDP`, as sync costs scale with the number of states. For further related work, including compression/sparsification and structured updates, check Section K.

## 7 CONCLUSION

`DES-LOC` reconciles communication efficiency with rigorous convergence guarantees in distributed adaptive optimization. By extending theory to the independent synchronization of `Adam` and `SGDM` optimizer states, we empirically demonstrate convergence alongside $170\times$ and $2\times$ communication reductions over `DDP` and prior state-of-the-art methods at billion-scale LLM training, even in environments prone to system failures. Our findings yield clear guidelines: i) **frequently** synchronize parameters, and ii) synchronize optimizer states **less often**, proportional to their half-lives. These insights open avenues for future research, including layer-wise synchronization, adaptive frequencies, compressed updates, as well as emerging applications, such as worldwide cross-data center training and collaborative training. As training workloads scale, we envision `DES-LOC` becoming the standard for efficient, resilient foundation-model training in data centers and distributed environments.

### ACKNOWLEDGMENTS

All costs for the computational resources used for this work were funded by Flower Labs, and the research conducted by a team of researchers from Flower Labs, The University of Cambridge, The Institute of Science and Technology of Austria, The University of Warwick, and Mohamed bin Zayed University of Artificial Intelligence. Support for university-based researchers came from a variety of sources, but in particular, the following funding organizations are acknowledged: the European Research Council (REDIAL), the Royal Academy of Engineering (DANTE), the Ministry of Education of Romania through the Credit and Scholarship Agency, and the European Union's Horizon 2020 research and innovation programme under the Marie Skłodowska-Curie grant agreement No 101034413.

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

# Appendix

# A Table of Contents

# A  EXPERIMENTAL DETAILS AND OPTIMIZER HYPERPARAMETER SWEEPS (SEE SECTION 4.1)

Here we provide additional experimental details complementing those in Section 4.1, including: a) model architecture details and hyperparameters independent of optimizer choice (Section A.1), b) our hyperparameter sweep procedure to select optimizer-specific settings (Section A.2), and c) the optimal hyperparameters with those used in Section 5 highlighted in bold.

## A.1  ARCHITECTURE DETAILS AND HYPERPARAMETERS

Table 3: Model architecture and training parameters. We denote the number of transformer blocks by #Blocks, number of attention heads by #Heads, embedding dimension by $d_{\text{model}}$, vocabulary size by $|\mathcal{V}|$, and feedforward-layer expansion by Exp. Ratio. All models use positional embeddings (Su et al., 2024), the `silu` activation function, and norm-based gradient clipping with clip-bound $\rho$. Global batch size (summed across all workers) is $|\mathcal{B}_{\text{G}}|$, and sequence length is standard for models at these scales. For model initialization we use $\sigma = 1/\sqrt{d_{\text{model}}}$. The total number of steps is denoted by $T$.

| Model Size | Blocks | $d_{\text{model}}$ | $|\mathcal{V}|$ | #Heads | Exp. Ratio | ROPE $\theta$ | ACT | Init $\sigma$ | $\rho$ | Seq Len | $|\mathcal{B}_{\text{G}}|$ | T |
|---|---|---|---|---|---|---|---|---|---|---|---|---|
| 135M | 30 | 576 | 50K | 9 | 4 | 10000 | silu | 0.04 | 1.0 | 2048 | 1024 | 1536, 3072 |
| 720M | 12 | 2048 | 50K | 16 | 4 | 10000 | SiLU | 0.02 | 1.0 | 2048 | 512 | 38912 |
| 1.7B | 24 | 2048 | 50K | 16 | 4 | 10000 | silu | 0.02 | 1.0 | 2048 | 1024 | 20480 |

Table 3 summarizes the architectural details of our models, following established practices for large language models at their respective scales. Unless otherwise stated, we adopt the hyperparameters recommended by Allal et al. (2025) for both the 135M and the 1.7B models. We operate at a batch size of 2M tokens, which is very large for the 135M model at the length of training we perform (Zhang et al., 2025) and industry-standard for the 1.7B model (Touvron et al., 2023), we chose to operate at large batch sizes because adaptive optimizers provide benefits primarily in large-batch training regimes (Kunstner et al., 2023). Moreover, we intend `DES-LOC` for use in cross data-center scenarios, where effectively utilizing available accelerators naturally demands large batch sizes and/or model scales. For both model sizes, we train for approximately $2\times$ the compute-optimal token budget (Hoffmann et al., 2022), placing our evaluations within the context of extended-duration foundation model training (Allal et al., 2025). Our chosen token budget is conservative due to resource constraints; for comparison, Allal et al. (2025) used 11 trillion tokens which is over $4000\times$ compute-optimal for the 135M model, and $300\times$ for the 1.7B.

We select warmup and decay schedules following recommendations from Zhang et al. (2025); Hägele et al. (2024); Allal et al. (2025). For the 135M model, the warmup period is set to $T_{\text{WARM}} = 512$ steps, corresponding to the roughly $40\%$ of the compute-optimal training tokens recommended by Zhang et al. (2025). For the 1.7B model, we use the recommended $T_{\text{WARM}} = 2048$ steps from Allal et al. (2025), roughly $10\%$ of total training. The stable-decay period uses a $1 - \text{SQRT}$ schedule over the final $T_{\text{DECAY}} = 10\% \times T$ steps (Hägele et al., 2024). For shorter runs, such as $T = 1536$ during heterogeneous-data evaluations, we keep the warmup fixed and proportionally scale the decay to ensure well-conditioned parameter updates during the stable learning rate period. The seeds we use for data sampling and for controlling the training algorithms and model are provided in the code accompanying the appendix.

## A.2  OPTIMIZER PARAMETERS SWEEPING PROCEDURE

As detailed in Section 2 and verified empirically in Section 5.2, the choice of decay rates $\beta_1, \beta_2$ strongly influences the effective synchronization frequencies achievable by both `DES-LOC` and `Local Adam`. This relationship arises directly from the half-life of optimizer states, given by $\tau_{0.5} = \frac{\ln(0.5)}{\ln(\beta)}$.

For `Adam`, prior studies such as Wortsman et al. (2023) have demonstrated a critical interplay between the learning rate ($\eta$), batch size, and the second-momentum decay $\beta_2$. Specifically, increasing either

the learning rate or batch size typically demands a lower $\beta_2$ to maintain training stability and avoid loss spikes. Conversely, higher $\beta_2$ values constrain the learning rate and batch size. Such dynamics have also been recently observed between the learning rate and the first-momentum decay $\beta_1$ in Pagliardini et al. (2025). Given that all our experiments use a fixed large batch size of roughly 2 million tokens (appropriate for billion-scale training), we systematically tune the learning rate $\eta$ in response to changes in $\beta_1, \beta_2$. We try values of $\beta_1, \beta_2$ based on previous works (Zhang et al., 2025) and follow the theoretical convergence requirement of Zhang et al. (2022) setting $\beta_1 \leq \sqrt{\beta_2}$.

Due to computational constraints, we cannot jointly optimize synchronization periods, data distributions, and decay parameters, and instead adopt a structured two-stage tuning approach:

1. **Stage 1: Tuning $\eta$ for `DDP`**. Starting from the recommended baseline learning rate ($\eta_0$) from Allal et al. (2025), we conduct a grid search as outlined by Charles et al. (2025): $\{\ldots, \sqrt{2}^{-2}\eta_0, \sqrt{2}^{-1}\eta_0, \eta_0, \sqrt{2}\eta_0, \sqrt{2}^2\eta_0, \ldots\}$ We expand this search until perplexity stops improving, identifying an optimal learning rate $\eta^*_{\text{DDP}}$ for each $(\beta_1, \beta_2)$ configuration.

2. **Stage 2: Tuning $\eta$ for `Local Adam`**. We then repeat this procedure for `Local Adam`, using $\eta^*_{\text{DDP}}$ as the new baseline. To balance generalizability and computational cost, we set the synchronization period to an intermediate value of $K = 64$, between high-frequency ($K = 16$) and low-frequency ($K = 256$) scenarios.

Additionally, following Zhang et al. (2025), we omit weight decay (set to zero) to simplify the hyperparameter tuning process, as it directly affects only model parameters, not optimizer states.

For experiments using `Nesterov`, we follow the hyperparamtere sweeping procedure of Charles et al. (2025), starting with a server learning rate of $1.0$ and a momentum of $0.9$ and only lowering it if it fails to converge

### A.2.1 OPTIMIZERS' HYPERPARAMETER CONFIGURATIONS

Table 4: Optimal learning rates $\eta^*$ for $\beta_1, \beta_2$ configurations of `ADOPT`/`Adam`. The hyperparameter sweep procedure (see Section A.2) involves incrementally adjusting the learning rate by factors of $\sqrt{2}$ around the initial value from Allal et al. (2025) until performance stops improving.

| Optimizer | $\beta_1$ | $\beta_2$ | $\eta^*$ |
|---|---|---|---|
| | 0.9 | 0.9999 | 0.0021 |
| ADOPT | **0.95** | **0.9999** | **0.0021** |
| | 0.99 | 0.9999 | 0.0014 |
| | 0.995 | 0.9999 | 0.0007 |
| | 0.9 | 0.95 | 0.0042 |
| | **0.95** | **0.95** | **0.003** |
| Adam | 0.9 | 0.99 | 0.003 |
| | 0.95 | 0.99 | 0.003 |
| | 0.99 | 0.99 | 0.0021 |

Our hyperparameter sweep (Table 4) indicates that the optimal learning rate $\eta^*$ under the warmup-stable-decay scheduler (Hägele et al., 2024) strongly depends on both optimizer type and the chosen $\beta_1, \beta_2$ values. For `Adam`, optimal learning rates and second-momentum decay ($\beta_2$) align closely with recommendations from Allal et al. (2025), though a slightly higher first-momentum decay ($\beta_1$) consistently performs better, in agreement with prior findings (Zhang et al., 2025). For `ADOPT` (default $\beta_2$), we observe a lower optimal learning rate compared to `Adam`, but similar best-performing $\beta_1$ values. We also find that the optimal learning rate does not differ between `DDP` and `Local Adam` for given $\beta_1, \beta_2$ when $K = 64$ and using a $\sqrt{2}$ sweep, higher learning rates either do not provide a benefit or diverge while lower learning rates are only necessary when pushing $K$ far closer to the complete training duration.

We find that increasing $\beta_1$ for `ADOPT`, and $\beta_1, \beta_2$ for `Adam`, leads to rapid performance degradation, particularly at or above $0.99$. Since the half-life at $\beta = 0.99$ ($\tau_{0.5} \approx 69$) is not sufficiently longer than at $\beta = 0.95$ ($\tau_{0.5} \approx 13.5$) to justify the observed performance drop, we select $\beta_1 = 0.95$ for all experiments, along with the default $\beta_2$ for `ADOPT` and $\beta_2 = 0.95$ for `Adam`.

> **Takeaway:** Increasing an optimizer state's $\beta$ significantly affects performance. Since linear increases in $\beta$ cause only logarithmic changes in half-life $\tau_{0.5}$, raising $\beta$ beyond the optimal value degrades performance without substantially improving the achievable synchronization frequency (Section 5.2).

## B    COMPLEMENTARY RESULTS TO SECTIONS 2.1 AND 5

We now provide additional results supplementing those presented in the main text. Specifically:

1. **Section B.1** complements Section 2.1 by including results on the heterogeneous data distribution described in Section 4.1. This highlights DES-LOC's robustness under imperfect sampling or strongly Non-IID federated scenarios (see Kairouz et al., 2021, Sec 3.1).

2. **Section B.2.1** complements Fig. 3 by showing the separate impact of varying synchronization frequencies for parameters and the second momentum when the base frequency is $K_b = 16$. It supports our claim that parameters and second momentum exhibit similar behavior across different synchronization regimes, unlike the first momentum.

3. **Section B.2.2** extends Fig. 3 by evaluating DES-LOC-Adam. We confirm that the parameter synchronization frequency is the most important, as predicted by our theory. In contrast, the momenta sync frequency is far less impactful, especially for low parameter sync frequencies.

4. **Section B.3.2** complements Fig. 4 by showing DES-LOC-ADOPT's perplexity against baseline methods on heterogeneous data (as defined in Section 4.1). This validates our claim from Contribution 2 regarding DES-LOC's effectiveness on heterogeneous datasets.

5. **Section B.3.3** presents an ablation study examining alternative low-communication configurations of DES-LOC, justifying our choice of $K_u = 3K_x, K_v = 6K_x$ used in Fig. 4.

6. **Section B.3.4** repeats the baseline comparison from Fig. 4 for DES-LOC-Adam, demonstrating that DES-LOC achieves similar communication reductions and performance when using Adam instead of ADOPT.

7. **Section B.4** provides additional metrics illustrating training instabilities for the FAVG+OPT baseline, including rapidly growing parameter norms, supporting observations in Fig. 5.b.

8. **Section B.5** presents the full 1B-scale comparison of DES-LOC Nesterov against DiLoCo, demonstrating the benefits of periodic optimizer-state synchronization under a Nesterov outer optimizer.

9. **Section B.6** benchmarks DES-LOC under extremely low bandwidth (10 Gbit/s), showing $9.42\times$ wall-clock speedups over DDP.

10. **Section B.7** provides full experimental details for the Muon experiments presented in Section 5.6.

11. **Section B.8** evaluates DES-LOC on the Flux vision model, demonstrating generality beyond LLMs.

12. **Section B.9** measures training throughput at the 7B parameter scale on B200 GPUs.

### B.1    TOY PROBLEM ON NON-IID DATA (SEE SECTION 2.1)

> **Toy Example Non-IID:** Fig. 8 simulates the scenario from Section 3, where each worker $m$ optimizes a distinct loss $f_m$ on heterogeneous data. Both DES-LOC and Local Adam show more stable convergence and get closer to the optimum than heuristic baselines.

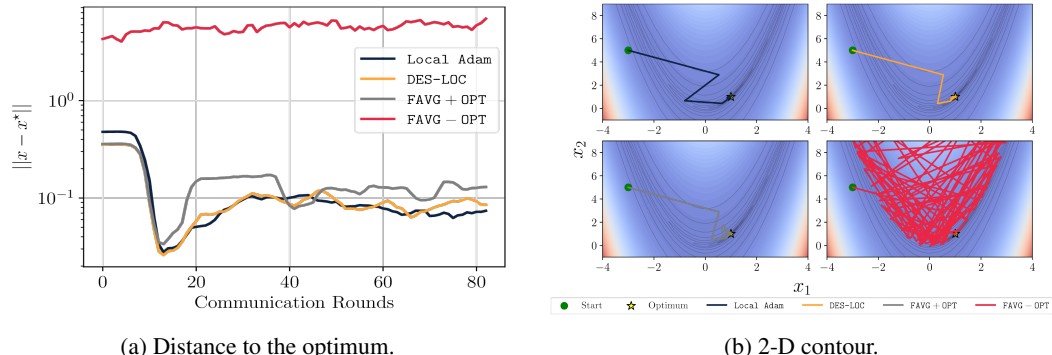

(a) Distance to the optimum.

(b) 2-D contour.

Figure 8: We present a toy problem in a `Non-IID` setting, where `DES-LOC` (with synchronization periods $K_x = 192, K_u = 192, K_v = 692$) and Local Adam (with $K = K_x$) converge to a superior solution compared to methods that keep optimizer states local ■ (Douillard et al., 2023; Sani et al., 2025) or periodically reset them ■ (Sani et al., 2024; Iacob et al., 2025). Like the `IID` scenario, resetting optimizer states prevents convergence due to repeated oscillations caused by reinitializations. Additionally, as seen in panel (a) between rounds 15 and 40, methods keeping optimizer states local suffer from larger oscillations further away from the optimum. The function optimized is $f(x_1, x_2) = (1 - x_1)^2 + 100(x_2 - x_1^2)^2$, and we simulate $M = 256$ workers, each adding Gaussian noise with worker-specific standard deviation $\sigma^m \sim \mathcal{N}(0, 3)$.

## B.2   RQ2: INDEPENDENT SYNC FREQUENCIES

This section provides supplementary results for **RQ2**, complementing Section 5.2. Section B.2.1 shows that perplexity has similar sensitivity to the first and second momentum synchronization frequencies at both high and low base synchronization frequencies. Additionally, Section B.2.2 repeats the comparison from Fig. 3 for `DES-LOC-Adam`, revealing similar trends regarding the importance of the parameters, with a reduced importance for the momenta due to lower $\beta_2$.

### B.2.1   PARAMETER AND SECOND MOMENTUM AT $K_b = 16$ (SEE FIG. 3.A, FIG. 3.B)

Figure 9 examines the effects of independently varying synchronization periods $(K_x, K_v)$ for parameters and second momentum under `DES-LOC-ADOPT` in the high-frequency regime ($K_b = 16$), chosen based on the first momentum's half-life ($\tau_{0.5} \approx 13.5$). Similar to the low-frequency results in Fig. 3.a, parameter synchronization frequency ($K_x$) strongly influences perplexity, while the second momentum ($K_v$) has minimal impact due to its long half-life. This contrasts with the first momentum, whose half-life closely matches the high-frequency period.

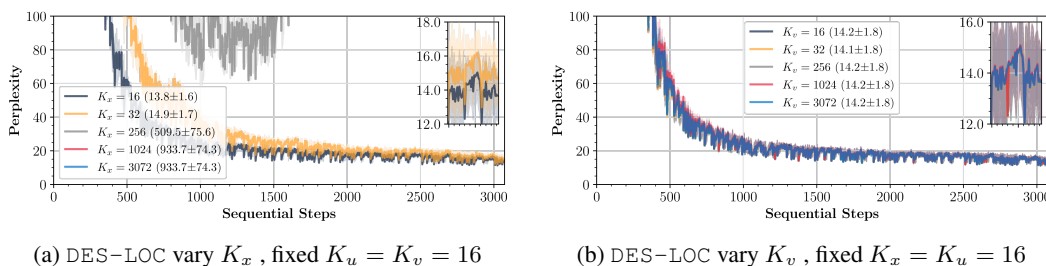

(a) `DES-LOC` vary $K_x$, fixed $K_u = K_v = 16$

(b) `DES-LOC` vary $K_v$, fixed $K_x = K_u = 16$

Figure 9: Model perplexity for `DES-LOC` (`ADOPT`, $\beta_1 = 0.95, \beta_2 = 0.9999$), independently varying synchronization periods at a high baseline frequency ($K_b = 16$). Similar to Fig. 3, parameter synchronization (a) is critical, with performance sharply degrading at higher periods, while second-momentum synchronization (b) has minimal impact due to its large half-life ($\tau_{0.5}(\beta_2) \gg K_b$).

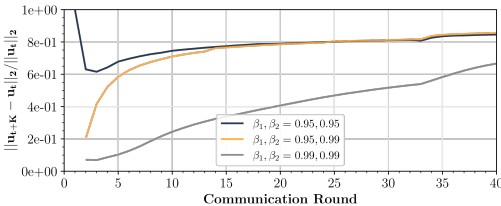

(a) First momentum change rate for `Local Adam`.

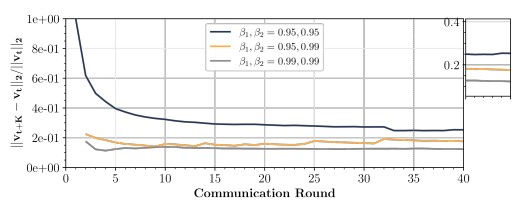

(b) Second momentum change rate for `Local Adam`.

Figure 10: Relative rates of change for first and second momenta across rounds using standard `Local Adam` ($K = 64$). Increasing $\beta_1$ substantially reduces the rate of change of the first momentum, while increasing either $\beta_1, \beta_2$ decreases the rate of change of the second.

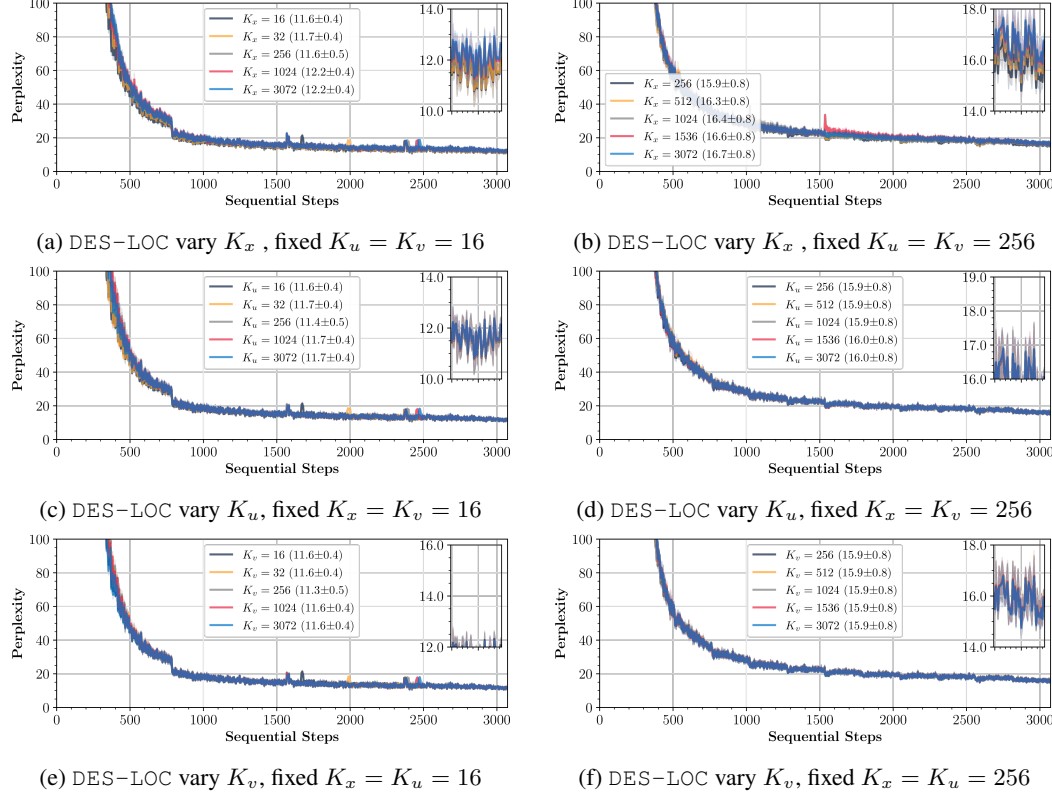

(a) `DES-LOC` vary $K_x$, fixed $K_u = K_v = 16$

(b) `DES-LOC` vary $K_x$, fixed $K_u = K_v = 256$

(c) `DES-LOC` vary $K_u$, fixed $K_x = K_v = 16$

(d) `DES-LOC` vary $K_u$, fixed $K_x = K_v = 256$

(e) `DES-LOC` vary $K_v$, fixed $K_x = K_u = 16$

(f) `DES-LOC` vary $K_v$, fixed $K_x = K_u = 256$

Figure 11: Model perplexity for `DES-LOC-Adam` ($\beta_1 = \beta_2 = 0.95$) when independently varying sync periods ($K_x, K_u, K_v$) while fixing others at baseline $K_b$. Parameter synchronization (a,b) influences performance in both high ($K_b = 16$) and low ($K_b = 256$) frequency regimes. Momenta synchronization minimally impacts perplexity due to both states' high adaptivity (low $\beta$), with potentially minor effects during the early stages of training in high-frequency regimes (c,e).

> **Takeaway:** In high-frequency synchronization regimes, the importance of parameters and the second momentum remains similar to the low-frequency regime shown in Section 5.2,

### B.2.2 ADAM RESULTS (SEE FIG. 3)

Fig. 10 show the rate of change results for `Adam` momenta at various $\beta$ values.

Figure 11 provides complementary results to Figs. 3 and 9 using `DES-LOC-Adam` with $\beta_1 = \beta_2 = 0.95$. Unlike `ADOPT`, the relatively low $\beta$ result in both the first and second momentum quickly adapting to the local gradients, reducing the impact of their sync frequency.

> **Takeaway:** For `DES-LOC-Adam`, parameter synchronization remains critical, consistent with theory. However, due to reduced $\beta_2$, momenta synchronization is less impactful since both the numerator and denominator of `Adam` updates are driven by local worker gradients after a few initial steps.

### B.3 RQ3: COMMUNICATION REDUCTION AND BASELINE COMPARISONS

This section provides supplementary results for **RQ3**, complementing Section 5.3. Section B.3.2 shows the perplexity of different configurations providing a $2\times$ communication reduction over `Local Adam`. Additionally, Section B.3.4 repeats the comparison against baselines from Section 5.3 for `DES-LOC-Adam`, showing similar communication reductions relative to `Local Adam`.

#### B.3.1 STEPWISE PLOTS FOR BASELINE COMPARISON

Figures 12 and 13 show stepwise plots for wall-clock results in the main text, they are the counterparts to Figs. 4 and 5.

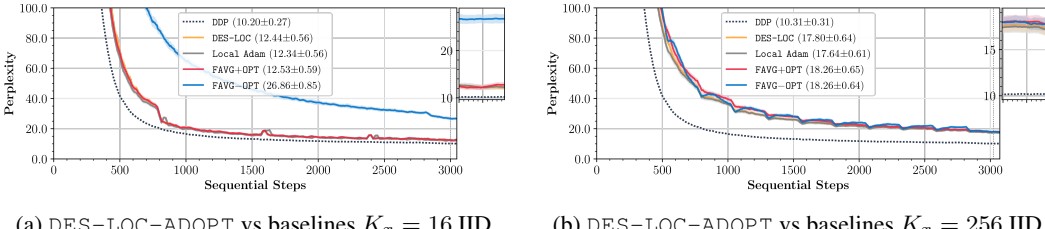

(a) `DES-LOC-ADOPT` vs baselines $K_x = 16$ IID     (b) `DES-LOC-ADOPT` vs baselines $K_x = 256$ IID

Figure 12: Setting $K_x = K$, $K_u = 3K_x$, and $K_v = 6K_x$, `DES-LOC` achieves a **$2\times$** communication reduction over `Local Adam`, matching performance at high (a) and low (b) frequencies for `Local Adam` and heuristic baselines (see Section 4.1). Using stepwise converges shows that `DES-LOC` matches `Local Adam` on a per-step basis.

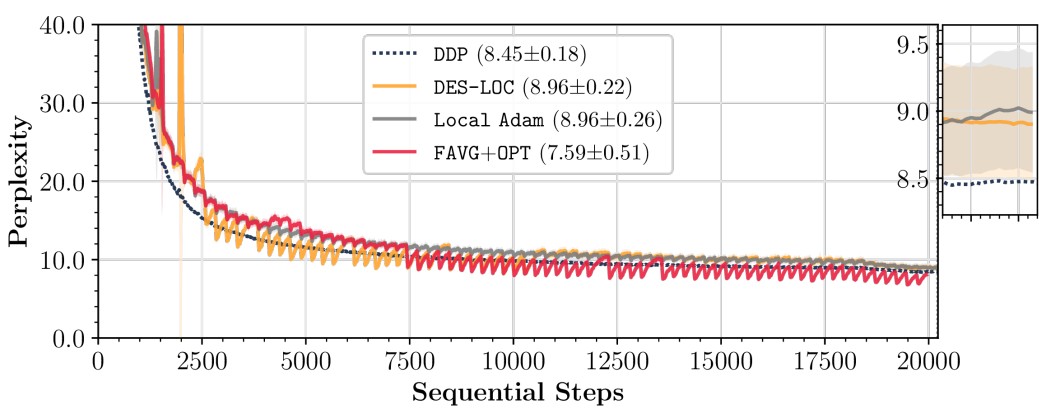

(a) `DES-LOC-ADOPT` 1B-model perplexity

Figure 13: `DES-LOC` matches `Local Adam` perplexity for billion-scale model training at half the communication cost ($K_x = 256, K_u = 3K_x, K_v = 6K_x$), representing a **$170\times$** reduction over `DDP`. Plot shows that stepwise convergence matches between `Local Adam` and `DES-LOC`).

> **Takeaway:** `DES-LOC` matches the stepwise convergence of `Local Adam` and approaches the convergence speed of `DDP`.

### B.3.2 DES-LOC ON HETEROGENEOUS DATA (SEE CONTRIBUTION 2)

Figure 14 evaluates the robustness of DES-LOC against baselines under heterogeneous (Non-IID) data distributions as described in Section 4.1. We set synchronization periods to $K_x = K$, $K_u = 3K_x$, and $K_v = 6K_x$ to achieve a targeted $2\times$ communication reduction over Local Adam.

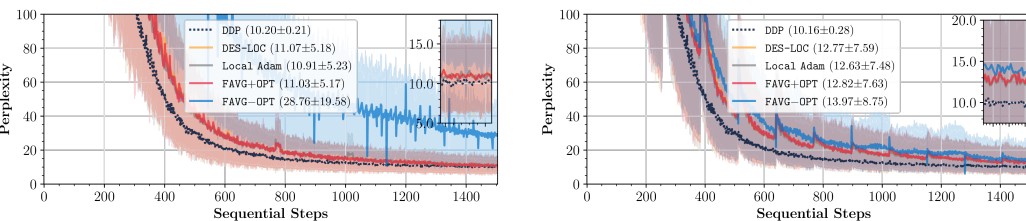

(a) DES-LOC-Adam, high sync frequency $K_x = 16$.    (b) DES-LOC-Adam, low sync frequency $K_x = 128$.

Figure 14: Comparison of perplexity under Non-IID conditions for DES-LOC, Local Adam ($K_x = K_u = K_v$), and heuristic baselines (defined in Section 4.1) at high (a) and low (b) synchronization frequencies. Due to higher cross-worker variance caused by heterogeneous data, parameters require slightly more frequent synchronization in the low-frequency regime ($K_x = 128 < 256$). Experiments are limited to $T = 1536$ steps ($\sim$compute-optimal) for computational feasibility.

> **Takeaway:** DES-LOC effectively converges on heterogeneous data distributions, maintaining the $2\times$ communication reduction observed in homogeneous settings. This aligns with our theoretical convergence results for heterogeneous losses (Section 3) and shows applicability in federated scenarios.

### B.3.3 DES-LOC LOW COMMUNICATION CONFIGURATIONS ABLATION (SEE FIG. 4)

Figure 15 explores alternative synchronization configurations enabling DES-LOC to achieve improved communication efficiency over Local Adam. Motivated by theoretical insights (Sections 2 and 3) and empirical evidence (Sections 5.1 and 5.2), we only consider settings where parameter synchronization is most frequent ($K_x \leq \min(K_u, K_v)$). This constraint follows from experiments in Section 5.2, which show that infrequent parameter synchronization significantly degrades perplexity, while momentum synchronization frequency has a smaller impact. For a fixed $2\times$ communication reduction over Local Adam, our findings confirm that synchronizing the first momentum more frequently than the second aligns with their respective half-lives and maintains performance close to Local Adam.

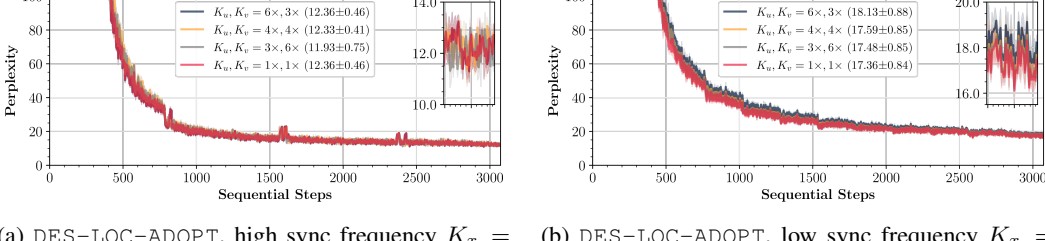

(a) DES-LOC-ADOPT, high sync frequency $K_x = 16$.    (b) DES-LOC-ADOPT, low sync frequency $K_x = 256$.

Figure 15: Configurations of DES-LOC targeting $2\times$ lower communication than Local Adam ($K_x = K_u = K_v$), setting $K_u, K_v$ as multiples of $K_x$. In both high (a) and low-frequency (b) regimes, performance depends on how communication is split between momenta for $\beta_1 \ll \beta_2$. Syncing the first momentum less often ($K_u = 6K_x, K_v = 3K_x$) degrades performance, wasting communication on the slow second momentum. Conversely, syncing it frequently ($K_u = 3K_x, K_v = 6K_x$) yields performance comparable to Local Adam. Setting $K_u = K_v = 4K_x$ produces intermediate results.

> **Takeaway:** For a given parameter synchronization period $K_x$ determined by bandwidth constraints, choose momentum synchronization periods $K_u, K_v$ as multiples of $K_x$. When $\beta_1 \ll \beta_2$, set $K_u < K_v$, with $K_u = 3 \times K_x$ and $K_v = 6 \times K_x$ providing robust default choices.

### B.3.4 ADAM RESULTS (SEE FIG. 4)

We now present results for `DES-LOC-Adam` with $\beta_1 = \beta_2 = 0.95$. `DES-LOC-Adam` achieves similar communication reductions over `Local Adam` and `DDP` as `ADOPT`. However, due to the lower $\beta_2$, the second-momentum half-life ($\tau_{0.5}(0.95) \approx 13.5$) is significantly shorter than for `ADOPT` ($\tau_{0.5}(0.9999) \approx 6931$). Figure 16 shows that with both momenta evolving at similar rates, the benefit of selecting $K_u < K_v$ diminishes. For consistency and due to meaningful empirical differences in rates of change (Section 5.1), we keep $K_u = 3 \times K_x$ and $K_v = 6 \times K_x$ in subsequent comparisons.

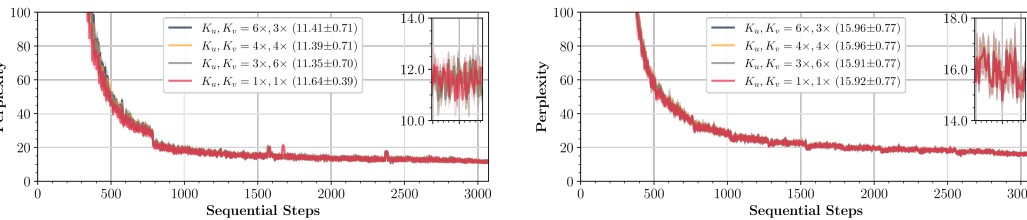

(a) `DES-LOC-Adam`, high sync frequency $K_x = 16$.  (b) `DES-LOC-Adam`, low sync frequency $K_x = 256$.

Figure 16: Configurations of `DES-LOC` targeting $2\times$ lower communication than `Local Adam` ($K_x = K_u = K_v$), using `Adam` ($\beta_1 = \beta_2 = 0.95$). In contrast to `DES-LOC-ADOPT` (where $\beta_1 \ll \beta_2$ yields an advantage for $K_u < K_v$ as shown in Fig. 15), the similar half-lives in `Adam` make perplexity insensitive to how communication is split between momenta for high (a) and low-frequencies (b).

Figure 17 shows `DES-LOC-Adam` achieves a $2\times$ communication reduction over the prior state-of-the-art `Local Adam` (Cheng & Glasgow, 2025) without significant perplexity degradation. Due to the much faster evolution of the optimizer states using `Adam` compared to `ADOPT`, local worker gradients drive the optimization reducing the benefit of allocating more of the communication budget to the first momentum.

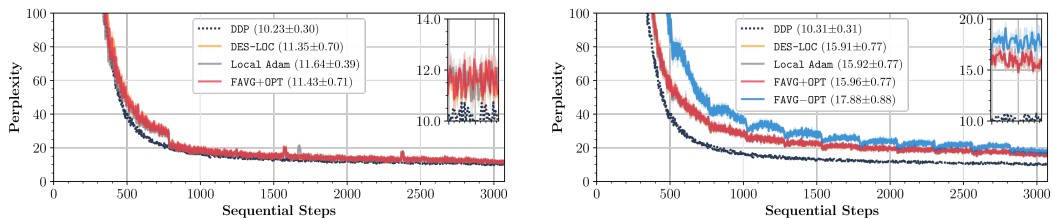

(a) `DES-LOC-Adam`, high sync frequency $K_x = 16$.  (b) `DES-LOC-Adam`, low sync frequency $K_x = 256$.

Figure 17: Setting $K_x = K$, $K_u = 3K_x$, and $K_v = 6K_x$, `DES-LOC-Adam` achieves a **$2\times$** communication reduction over `Local Adam`, matching performance at high (a) and low (b) frequencies for `Local Adam` and heuristic baselines (see Section 4.1).

> **Takeaway:** `DES-LOC-Adam` achieves a similar $2\times$ communication reduction over `Local Adam` as `DES-LOC-ADOPT` by exploiting the reduced importance of optimizer-state synchronization relative to parameters. However, due to the smaller $\beta_2$ in `Adam`, there is limited benefit from assigning different synchronization frequencies to the first and second momenta compared to `ADOPT`.

### B.4 **RQ4:** ADDITIONAL METRICS AND TRAINING INSTABILITIES OF `FAVG+OPT` (SEE FIG. 5.B)

Figure 18 complements Fig. 5.b by showing parameter and update norms for `DES-LOC` and baseline methods when training billion-scale models. Both `DES-LOC` and `Local Adam` regularize updates by synchronizing optimizer states, effectively reducing update norms due to averaging across workers (triangle inequality). In contrast, the heuristic baseline (Sani et al., 2025) experiences large updates, leading to uncontrolled parameter growth, increased activations (Fig. 5.b), and degraded performance on downstream `ICL` tasks (Table 1) relative to its perplexity (Fig. 5.a).

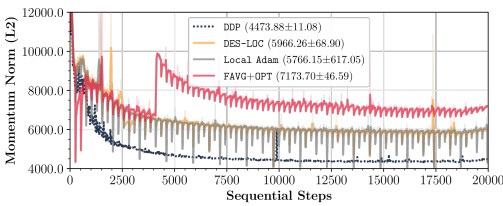
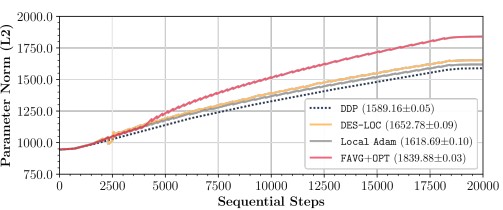

(a) `DES-LOC-ADOPT` 1B-update norms   (b) `DES-LOC-ADOPT` 1B-parameter norms

Figure 18: Comparison of update (a) and parameter norms (b) for billion-scale models trained with `DES-LOC` ($K_x = 256, K_u = 768, K_v = 1536$), `Local Adam` ($K = 256$), `DDP`, and Federated Averaging with persistent optimizer states (`FAVG+OPT`). Frequent synchronization in `Local Adam` and `DDP` consistently reduces update and parameter norms. Similarly, `DES-LOC` achieves comparable reductions at intervals corresponding to multiples of `lcm`($K_x, K_u, K_v$), with smaller intermediate drops. Conversely, `FAVG+OPT`, which does not synchronize optimizer states, experiences persistently larger and noisier updates, becoming vulnerable to spikes (notably before step 5000). This leads to uncontrolled parameter growth (b).

Table 5: Throughput (batches/sec) for 1B-13B models to reach $2\times$ compute-optimal tokens at high ($K = 16$) and low ($K = 256$) frequencies with a 2M token batch size. All local methods achieve significant throughput gains over the `DDP` baseline. At high frequency ($K = 16$), `DES-LOC` boosts throughput by over $1.7\times$ on the 13B model. At low frequency ($K = 256$), this advantage grows to over $2.1\times$ versus `DDP`.

| | 1B Model | | 7B Model | | 13B Model | |
|---|---|---|---|---|---|---|
| **Method** | $K_x = 16$ | $K_x = 256$ | $K_x = 16$ | $K_x = 256$ | $K_x = 16$ | $K_x = 256$ |
| DDP (Baseline) | $171.9 \pm 0.94$ | $171.9 \pm 0.9$ | $25.1 \pm 0.10$ | $25.1 \pm 0.10$ | $11.1 \pm 0.03$ | $11.1 \pm 0.03$ |
| `FAVG+OPT` | $304.9 \pm 2.51$ | $385.4 \pm 3.7$ | $34.0 \pm 0.11$ | $40.4 \pm 0.15$ | $19.4 \pm 0.11$ | $23.5 \pm 0.15$ |
| `Local Adam` | $253.1 \pm 1.59$ | $380.1 \pm 3.6$ | $30.9 \pm 0.09$ | $40.2 \pm 0.15$ | $16.7 \pm 0.07$ | $23.3 \pm 0.14$ |
| `DES-LOC` ($K_u, K_v = 3K_x, 6K_x$) | $299.2 \pm 2.39$ | $384.1 \pm 3.7$ | $33.7 \pm 0.11$ | $40.4 \pm 0.15$ | $19.0 \pm 0.10$ | $23.5 \pm 0.15$ |

> **Takeaway:** Unlike heuristic methods, which maintain purely local optimizer states leading to unstable, noisy updates, `DES-LOC` provides stable regularization similar to `Local Adam` and `DDP` by periodically synchronizing parameters and momenta, reducing training instabilities.

### B.5 `DES-LOC` NESTEROV VS. DILOCO AT 1B SCALE

Having shown that a `Nesterov` outer optimizer improves `DES-LOC` (Section 5.5), we ask whether synchronizing optimizer states still helps relative to the local-state ($K_u = K_v = \infty$) `Nesterov` method `DiLoCo`. Charles et al. (2025) has shown that at **large** scale ($> 1B$ parameters) `DiLoCo` can match or outperform `DDP` with `Adam`. We adopt their outer hyper-params and train a 1B model with the same experimental design as Section 5.5 using $4 \times 8$ `H100s` for 40,960 steps ($\approx 4\times$ compute-optimal) with inner `Adam`, comparing `Adam-DDP`, `Local Adam`, `DiLoCo`, and `DES-LOC` `Nesterov`, with $K_x = 32$ for `Local Adam` and `DiLoCo` and $K_u = 4K_x, K_v = 8K_x$ for `DES-LOC` `Nesterov`.

Figure 19 (top left) shows that `Nesterov`-based methods outperform AdamW: `DiLoCo` achieves $7.63\pm0.20$ validation perplexity, improving over `Adam-DDP` by $\approx 2\%$, while `DES-LOC` `Nesterov`

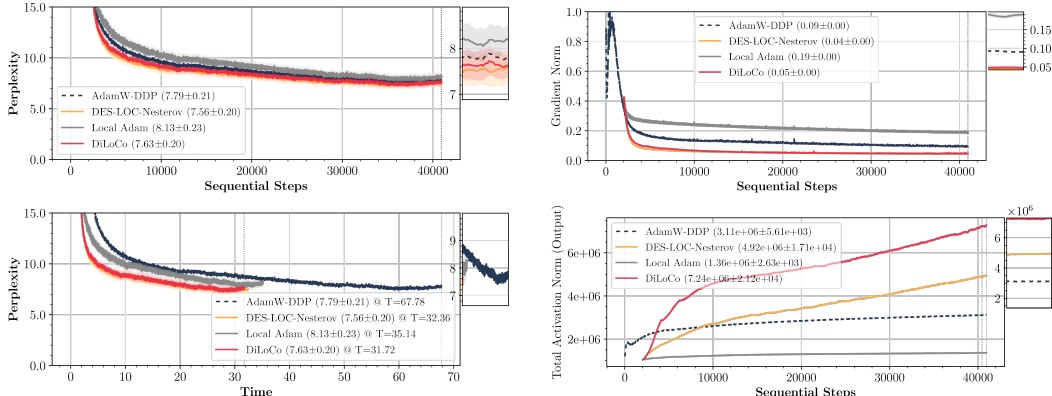

Figure 19: Comparison of `Adam-DDP`, `Local Adam`, `DiLoCo`, and `DES-LOC Nesterov` on a 1B-parameter model trained for 40,960 steps with an AdamW inner optimizer. Top left: train perplexity vs steps. Top right: worker gradient norms. Bottom left: train perplexity vs time, bottom right: whole-model output activation norms. Shaded regions show std across workers. `DES-LOC Nesterov` outperforms `Adam`, it also outperforms `DiLoCo` at the cost of more communication. Error bars show variance across workers, accounting for compounding local drift.

reaches $7.56 \pm 0.20$, a $\approx 0.9\%$ gain over `DiLoCo`; both outperform `Local Adam` ($8.13 \pm 0.23$). Note that this comparison pits `Nesterov`-based local updates against `DDP` with standard `Adam`; as Charles et al. (2025) note, once `DDP` also uses `Nesterov`, this gap can shrink or reverse depending on model size and worker count. These results show that synchronizing optimizer states preserves the benefits of `Nesterov` while retaining the advantages of state averaging.

We analyze the interaction between the optimizer states and the outer optimizer by measuring the gradient norms and activation statistics. In Figure 19 (top and bottom right), for both `DiLoCo` and `DES-LOC Nesterov`, gradient norms drop rapidly relative to `Adam-DDP` and `Local Adam` and remain roughly $2\times$ smaller than `DDP` thereafter, suggesting that `Nesterov` may steer optimization toward smoother regions of the loss landscape. State synchronization slightly accelerates the decrease in gradient norm over `DiLoCo`, after which the curves coincide.

Under `DiLoCo`, total output activation norms grow monotonically to more than $2\times$ the `Adam-DDP` values, whereas `DES-LOC Nesterov` substantially slows this growth, ending $\approx 32\%$ below `DiLoCo` (bottom row). This resembles the stabilization seen for `DES-LOC` versus `FedAvg` baselines without optimizer-state averaging (Fig. 5), where periodic averaging also curbed activation growth. This supports viewing finite synchronization as a regularizer that limits worker drift and optimizer-state noise, yielding better-controlled activations while offering `Nesterov`'s benefits. `DES-LOC Nesterov` does incur additional communication costs relative to `DiLoCo` at fixed $K_x$, being $\approx 2\%$ slower than `DiLoCo` under these bandwidth conditions (Fig. 19 bottom left) and $K_u, K_v$ settings while being $\approx 8\%$ faster than `Local Adam`. The extra cost can be made arbitrarily small by increasing the optimizer-state sync periods ($K_u, K_v$): in the limit $K_u, K_v \to \infty$ it recovers `DiLoCo` in both performance and communication, while any finite sync period partly inherits the robustness and fault-tolerance benefits of synchronizing optimizer states, modulated by the chosen $\beta$'s.

> **Takeaway:** At the 1B scale and long horizons, `Nesterov`-based local-update methods (`DiLoCo`, `DES-LOC Nesterov`) outperform `Adam-DDP`, consistent with prior scaling-law results. Relative to `DiLoCo`, `DES-LOC Nesterov` matches or improves perplexity while substantially reducing gradient and activation norms via periodic optimizer-state synchronization, yielding a tunable point on the communication–performance Pareto frontier.

## B.6 VERY LOW BANDWIDTH EXPERIMENTS

While perplexity is invariant to network bandwidth, wall-clock time is highly sensitive to it. To practically showcase this, we perform a benchmark with a 1B model to measure time under extremely low bandwidth conditions (10 Gbit/s). This setup simulates a scenario with affordable, consumer-

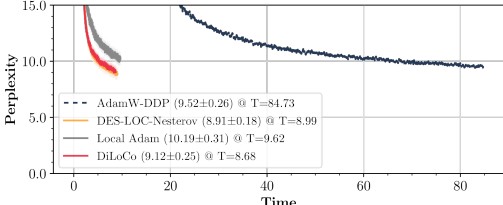 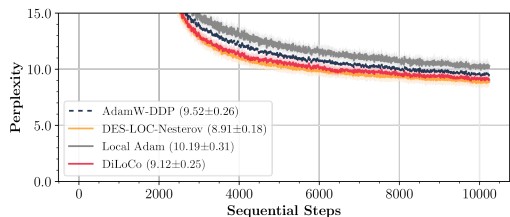

Figure 20: Training efficiency benchmark on a 1B model under restricted bandwidth (10 Gbit/s). **Left:** Perplexity versus wall-clock time. `DES-LOC Nesterov` effectively decouples training time from bandwidth, finishing in $\approx 9$ hours compared to the projected $\approx 3.5$ days for `DDP` (dashed blue). **Right:** Perplexity versus sequential steps. While step-wise convergence is comparable, the communication overhead of `DDP` creates a massive bottleneck in the time domain.

grade interconnects rather than data-centers. Due to the extreme gradient synchronization delay inherent to `DDP` in this regime, the benchmark was limited to a 10,240 step horizon to remain feasible.

As shown in Figure 20, `DES-LOC Nesterov` dramatically reduces training time by $\approx 9.42\times$ compared to `DDP`, completing the run in 8.99 hours versus 84.73 hours (3.5 days) for `DDP`, even with the constant overheads of our unoptimized implementation. Furthermore, `DES-LOC Nesterov` is more efficient than `Local Adam`, finishing $\approx 7\%$ faster (8.99h vs. 9.62h) while achieving significantly lower perplexity (8.91 vs. 10.19). When compared to the ultra-lightweight `DiLoCo` baseline (8.68h), `DES-LOC Nesterov` incurs a time penalty of $\approx 3.6\%$ due to the additional optimizer state synchronization. However, this yields performance gains, improving final perplexity by $\approx 2.3\%$ (8.91 vs. 9.12) over `DiLoCo`.

> **Takeaway:** In extremely low bandwidth settings (10 Gbit/s), `DES-LOC Nesterov` eliminates the communication bottleneck, reducing training time by $9.42\times$ over `DDP`. It strikes a balance on the Pareto frontier: its wall-clock time is in-between those of `Local Adam` and `DiLoCo` while outperforming them both in perplexity.

## B.7 MUON EXPERIMENTAL DETAILS (SEE SECTION 5.6)

This section provides the full experimental details for the `Muon` experiments presented in Section 5.6. Although a comprehensive theoretical treatment of preconditioned local updates under Newton-Schulz iterations is outside the scope of this work, the `DES-LOC` design is inherently compatible with such structures, as the momentum buffer follows standard EMA dynamics regardless of the subsequent preconditioning step.

**Experimental Details.** We utilize the standard PyTorch implementation of `Muon` with Nesterov momentum enabled and a weight decay of $0.1$. Following the recommendations of Liu et al. (2025), we apply the `match_rms_norm` adjustment to learning rates. We adopt the conventional split optimization strategy for `Muon`: `AdamW` handles embeddings and layer normalizations, while `Muon` optimizes all 2D matrices (Jordan et al., 2024). The momentum parameter for `Muon` is set to $\beta = 0.9$, while the `Adam` component retains the $\beta_1 = 0.9, \beta_2 = 0.999$ settings used elsewhere. Gradient clipping thresholds are scaled by model size: $1.0$ for 16M, $0.5$ for 125M, and $0.25$ for 360M. For the `Local Muon` baseline, all optimizer states (Muon momentum; Adam first/second momenta) synchronize every 32 steps. In contrast, `DES-LOC` delays state synchronization: the first momentum (for both optimizers) synchronizes every 96 steps ($3\times$ reduction), and `Adam`'s second momentum synchronizes every 192 steps ($6\times$ reduction).

## B.8 EXPERIMENTS ON THE FLUX VISION MODEL

To demonstrate the universality of `DES-LOC` across different modalities and architectures beyond standard decoder-only LLMs, we evaluate its performance on `Flux` (Labs et al., 2025), a Rectified Flow Transformer designed for text-to-image generation. This architecture differs significantly from the causal language models evaluated in previous sections, serving as a robust test for the generalizability of our decoupled synchronization approach.

**Experimental Setup.** We utilize the 280M parameter variant of `Flux` provided by `torchtitan`, training with a global batch size of 256. The inner optimizer is `AdamW` with $\beta_1 = 0.9, \beta_2 = 0.999$. We compare three settings: (1) `DDP`, (2) `Local Adam` with a synchronization period of $K = 32$, and (3) `DES-LOC` with a parameter sync period of $K_x = 32$. For `DES-LOC`, we decouple the momentum synchronization significantly, setting $K_u = 3K_x$ and $K_v = 6K_x$ (192 steps).

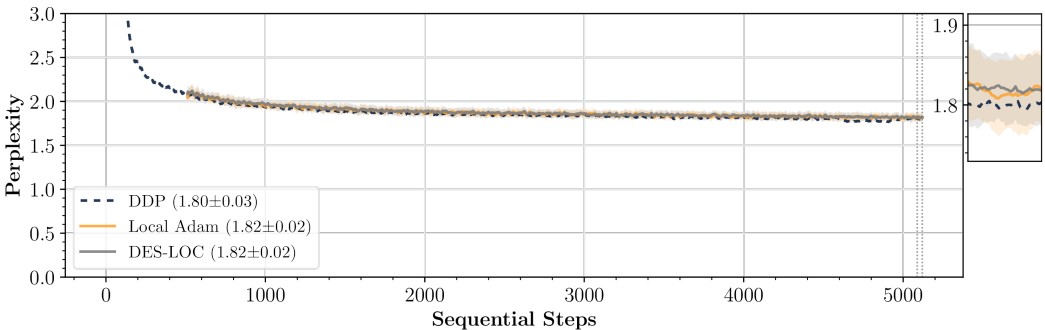

Figure 21: Training loss comparison on the 280M parameter `Flux` model (Rectified Flow Transformer). `DES-LOC` ($K_x = 32, K_u = 192, K_v = 192$) effectively matches the convergence trajectory of both the fully synchronous `DDP` baseline and `Local Adam` ($K = 32$).

Our results, visualized in Fig. 21, indicate that `DES-LOC` generally matches the performance of `Local Adam` and approaches the `DDP` upper bound.

> **Takeaway:** The efficacy of `DES-LOC` extends beyond LLMs to Rectified Flow Transformers (`Flux`). The method generally matches the performance of `DDP` and `Local Adam` while reducing communication by $2\times$ over `Local Adam`, demonstrating the universality of the approach. We leave the scaling of this result to larger vision models for future work.

## B.9 THROUGHPUT AT 7B SCALE

To assess the practical scalability of our method on state-of-the-art hardware and at large model scales, we measure the training throughput of a 7B parameter model distributed across 8 independent NVIDIA B200 GPUs.

**Throughput Analysis.** As illustrated in Fig. 22, during the local update phases, each GPU operates at the peak efficiency of a fully isolated local run, achieving identical tokens-per-second throughput as a single B200 with zero synchronization overhead. Distinct drops in throughput are observed only at the sparse synchronization boundaries ($K_x = 32, K_u = 96, K_v = 192$), where the system pauses to aggregate model parameters and optimizer states.

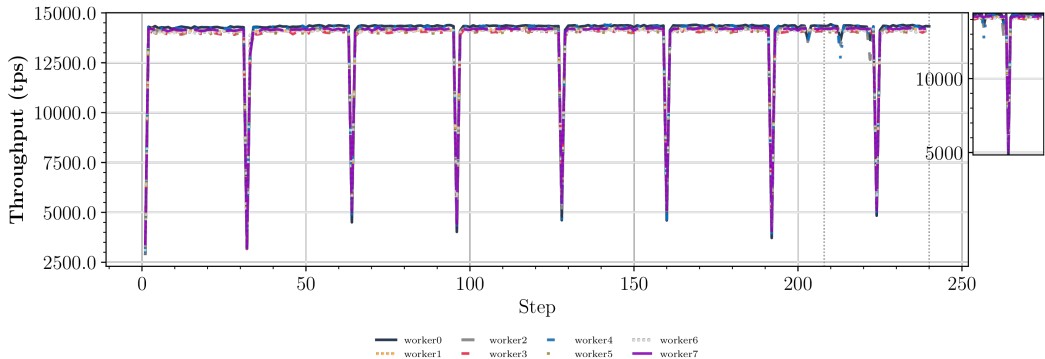

Figure 22: Instantaneous throughput (tokens/sec) for a 7B model on 8x B200s. `DES-LOC` maintains peak "local-only" speed for the vast majority of steps, with throughput dips occurring only at synchronization intervals $(32, 96, 192)$. In contrast, standard `DDP` would incur a synchronization penalty at *every* step, permanently depressing the throughput curve.

Crucially, standard `DDP` incurs this communication penalty at *every single training step*, significantly lowering the average tokens/sec. Even with our current unoptimized "stop-the-world" implementation—which explicitly pauses computation to communicate and does not leverage computation-communication overlap—`DES-LOC` significantly increases aggregate throughput by amortizing these costs over long local training windows.

> **Takeaway:** On high-performance B200 hardware, `DES-LOC` enables near-linear scaling by keeping workers in a high-throughput local regime for the majority of training. By restricting communication overhead to sparse intervals, it delivers significant wall-clock speedups over `DDP`, even without low-level implementation optimizations like communication overlap.

## C    FURTHER ALGORITHMIC DETAILS OF `DES-LOC`

### C.1    EXTENSION TO `FEDOPT`

Although Cheng & Glasgow (2025) show provable convergence for adaptive inner optimizers in a federated optimization framework, their result rests on the assumption that after a period of local work, the new global model is created by averaging the local client models. In relation to the larger `FedOpt` literature (Reddi et al., 2021), the scheme chosen by Cheng & Glasgow (2025) resembles that of `FedAvg`, or where the server optimizer is `SGD` with the outer learning rate set to one (Reddi et al., 2021). Naturally, the question of whether alternate server optimizers than have been used in prior works can also be implemented for `Local Adam`, and thus `DES-LOC`, arises.

We argue that indeed `DES-LOC`'s principles can be effectively applied to any `FedOpt` method and not just `FedAvg`. While using an alternate server optimizer does not have proven convergence guarantees as yet, we show in Algorithm 1 that the choice of the `ServerOpt` is not constrained from a practical point-of-view. However, the improvements that `DES-LOC` provide are related to the local optimization procedure, which is orthogonal to the outer optimizer choice. Choosing the correct, and most effective, outer optimizer is an open research area (Khaled et al., 2025), and we leave the investigations of the interactions between `DES-LOC` and outer optimziers to future work.

## C.2 Deterministic Optimizer-specific Variants of Algorithm 1

---

**Algorithm 2** `DES-LOC-Adam`

---

**Require: Model tensors, Hyper-parameters**
1:    $x_0, u_{-1}, v_{-1} \in \mathbb{R}^d$ — initial parameter vector, seeds for first and second moments
2:    $\{\eta_t\}_{t=0}^{T-1} \subset \mathbb{R}_{>0}$ — step-size schedule
3:    $\beta_1, \beta_2 \in [0,1)$ — Adam decay factors
4:    $\rho, \lambda \in \mathbb{R}_{>0}$ — gradient clipping term, $\ell_2$ stability term
5:    $T, M \in \mathbb{N}_+$ — total iterations, number of workers
6:    $K_x, K_u, K_v \in \mathbb{N}_+$ — sync periods for parameters, first and second moments

**Ensure:** $x_T,\ u_{T-1},\ v_{T-1}$

7: **for each worker** $m$: $x_0^m = x_0,\ u_{-1}^m = v_{-1}^m = 0$      *local init ($t = -1$ seeds)*
8: **for** $t = 0, \ldots, T-1$ **do**      *training loop*
9:    **for all** workers $m = 0, \ldots, M-1$ **in parallel do**
10:      $g_t^m \leftarrow \nabla F(x_t^m; \xi_t^m)$      *stochastic gradient*
11:      $\widehat{g}_t^m \leftarrow \mathbf{clip}(g_t^m, \rho)$      *clip to radius $\rho$*
12:      **if** $t \bmod K_u = 0$ **then**      *sync $u$*
13:        $u_t^m \leftarrow \beta_1 \mathbb{E}_m[u_{t-1}^m] + (1-\beta_1)\widehat{g}_t^m$
14:      **else**
15:        $u_t^m \leftarrow \beta_1 u_{t-1}^m + (1-\beta_1)\widehat{g}_t^m$
16:      **if** $t \bmod K_v = 0$ **then**      *sync $v$*
17:        $v_t^m \leftarrow \beta_2 \mathbb{E}_m[v_{t-1}^m] + (1-\beta_2)(\widehat{g}_t^m \odot \widehat{g}_t^m)$
18:      **else**
19:        $v_t^m \leftarrow \beta_2 v_{t-1}^m + (1-\beta_2)(\widehat{g}_t^m \odot \widehat{g}_t^m)$
20:      $d_t^m \leftarrow \dfrac{\eta_t}{\sqrt{v_t^m + \lambda^2}} \odot u_t^m$      *bias-corrected step*
21:      **if** $t \bmod K_x = 0$ **then**      *sync $x$*
22:        $x_{t+1}^m \leftarrow \mathbb{E}_m[x_t^m] - d_t^m$
23:      **else**
24:        $x_{t+1}^m \leftarrow x_t^m - d_t^m$

---

**Algorithm 3** `DES-LOC-ADOPT`

---

**Require: Model tensors, Hyper-parameters**
1:    $x_0, m_{-1}, v_{-1} \in \mathbb{R}^d$ — initial parameter vector and momenta
2:    $\{\eta_t\}_{t=0}^{T-1} \subset \mathbb{R}_{>0}$ — learning rate schedule
3:    $\beta_1, \beta_2 \in [0,1)$ — decay factors
4:    $\rho, \epsilon \in \mathbb{R}_{>0}$ — gradient clipping term, small stability constant
5:    $T, M \in \mathbb{N}_+$ — total iterations, number of workers
6:    $K_x, K_m, K_v \in \mathbb{N}_+$ — sync periods for parameters, first and second moments

**Ensure:** $x_T,\ m_{T-1},\ v_{T-1}$

7: **for each worker** $m$: $x_0^m = x_0,\ m_{-1}^m = v_{-1}^m = 0$      *local initialization*
8: **for** $t = 0, \ldots, T-1$ **do**
9:    **for all** workers $m = 0, \ldots, M-1$ **in parallel do**
10:      $g_t^m \leftarrow \nabla F(x_t^m; \xi_t^m)$      *stochastic gradient*
11:      $\widehat{g}_t^m \leftarrow \mathbf{clip}(g_t^m, \rho)$      *gradient clipping*
12:      **if** $t \bmod K_v = 0$ **then**
13:        $v_t^m \leftarrow \beta_2 \mathbb{E}_m[v_{t-1}^m] + (1-\beta_2)(\widehat{g}_t^m \odot \widehat{g}_t^m)$
14:      **else**
15:        $v_t^m \leftarrow \beta_2 v_{t-1}^m + (1-\beta_2)(\widehat{g}_t^m \odot \widehat{g}_t^m)$
16:      **if** $t \bmod K_m = 0$ **then**
17:        $m_t^m \leftarrow \beta_1 \mathbb{E}_m[m_{t-1}^m] + (1-\beta_1)\dfrac{\widehat{g}_t^m}{\max\{\sqrt{v_{t-1}^m}, \epsilon\}}$
18:      **else**
19:        $m_t^m \leftarrow \beta_1 m_{t-1}^m + (1-\beta_1)\dfrac{\widehat{g}_t^m}{\max\{\sqrt{v_{t-1}^m}, \epsilon\}}$
20:      $d_t^m \leftarrow \eta_t m_t^m$      *ADOPT update*
21:      **if** $t \bmod K_x = 0$ **then**
22:        $x_{t+1}^m \leftarrow \mathbb{E}_m[x_t^m] - d_t^m$
23:      **else**
24:        $x_{t+1}^m \leftarrow x_t^m - d_t^m$

---

# D    CONVERGENCE ANALYSIS OF DES-LOC-SGDM (IN EXPECTATION BOUNDS)

Here we provide a non-convex convergence analysis of the proposed DES-LOC approach applied to the SGDM optimizer which has a single state ($N = 1$, momentum). The complete description of the algorithm can be found in Algorithm 4.

---

**Algorithm 4** DES-LOC-SGDM

**Require: Model tensors**
1:    $x_0 \in \mathbb{R}^d$ — initial parameter vector
2:    $u_{-1} \in \mathbb{R}^d$ — seed for the momentum, initialised to $\mathbf{0}$
**Require: Hyper-parameters**
3:    $\{\eta_t\}_{t=0}^{T-1} \subset \mathbb{R}_{>0}$ — step-size schedule
4:    $\beta \in [0, 1)$ — Momentum decay factor
5:    $T \in \mathbb{N}_+$ — total optimisation iterations
6:    $M \in \mathbb{N}_+$ — number of workers
7:    $p_x = \frac{1}{K_x}, p_u = \frac{1}{K_u} \in [0, 1]$ — synchronization probabilities for parameters and momentum
**Ensure:** $x_T, u_{T-1}, v_{T-1}$
8: **for each worker** $m$: $x_0^m = x_0,\ u_{-1}^m = v_{-1}^m = 0$                         local init ($t = -1$ seeds)
9: **for** $t = 0, \ldots, T - 1$ **do**                                                                         training loop
10:      **for all workers** $m = 0, \ldots, M - 1$ **in parallel do**
11:          $g_t^m \leftarrow \nabla F_m(x_t^m; \xi_t^m)$                                                stochastic gradient
12:          $u_t^m \leftarrow \begin{cases} \mathbb{E}_m[\beta u_{t-1}^m + (1 - \beta)g_t^m], & \text{with probability } p_u \\ \beta u_{t-1}^m + (1 - \beta)g_t^m, & \text{with probability } 1 - p_u \end{cases}$                    sync $u$
13:          $x_{t+1}^m \leftarrow \begin{cases} \mathbb{E}_m[x_t^m - \eta_t u_t^m], & \text{with probability } p_x \\ x_t^m - \eta_t u_t^m, & \text{with probability } 1 - p_x \end{cases}$                    sync $x$

---

In order to facilitate the technical presentation, we model synchronization frequencies by assigning probabilities to each averaging event. For example, the parameters $x_t^m$ are synchronized with the probability $p_x = \frac{1}{K_x}$, which is statistically equivalent to performing the averaging in every $\frac{1}{p_x} = K_x$ iteration. Similarly, momentum $u_t^m$ synchronization happens with probability $p_u = \frac{1}{K_u}$, which can differ from $p_x$.

Step 1 (virtual iterates). For each step $t \geq 0$, denote the average parameters, momentum and gradient as follows:

$$x_t \stackrel{\text{def}}{=} \mathbb{E}_m[x_t^m], \quad u_t \stackrel{\text{def}}{=} \mathbb{E}_m[u_t^m], \quad g_t \stackrel{\text{def}}{=} \mathbb{E}_m[g_t^m].$$

Then these averaged variables follow the "standard" centralized SGDM dynamics:

$$\begin{aligned} u_t &= \beta u_{t-1} + (1 - \beta)g_t \\ x_{t+1} &= x_t - \eta u_t. \end{aligned}$$

Letting $x_{-1} = x_0$, define the global virtual iterations as follows

$$z_t \stackrel{\text{def}}{=} \frac{1}{1 - \beta}x_t - \frac{\beta}{1 - \beta}x_{t-1}, \quad t \geq 0.$$

The key property of this virtual iterates we are going to exploit in the next steps is that they follow averaged gradients, namely for any $t \geq 0$ we have

$$\begin{aligned} z_{t+1} - z_t &= \frac{1}{1 - \beta}(x_{t+1} - x_t) - \frac{\beta}{1 - \beta}(x_t - x_{t-1}) \\ &= -\frac{\eta}{1 - \beta}u_t + \frac{\eta\beta}{1 - \beta}u_{t-1} = -\frac{\eta}{1 - \beta}(u_t - \beta u_{t-1}) = -\eta g_t. \end{aligned}$$

**Step 2 (smoothness over virtual iterates).** Then we apply smoothness of the global loss function $f$ over these global virtual iterates.

$$
\begin{aligned}
f(z_{t+1}) &\leq f(z_t) + \langle \nabla f(z_t), z_{t+1} - z_t \rangle + \frac{L}{2}\|z_{t+1} - z_t\|^2 \\
&= f(z_t) + \underbrace{\langle \nabla f(x_t), z_{t+1} - z_t \rangle}_{I} + \underbrace{\langle \nabla f(z_t) - \nabla f(x_t), z_{t+1} - z_t \rangle}_{II} + \underbrace{\frac{L}{2}\|z_{t+1} - z_t\|^2}_{III}.
\end{aligned}
$$

In the next step, we separately bound each term appearing in the above bound.

**Step 3a (one step progress).** Bounding term I.

$$
\begin{aligned}
&\mathbb{E}\langle \nabla f(x_t), z_{t+1} - z_t \rangle \\
&= -\eta\mathbb{E}\left\langle \nabla f(x_t), \frac{1}{M}\sum_{m=1}^{M} g_t^m \right\rangle = -\eta\mathbb{E}\left\langle \nabla f(x_t), \frac{1}{M}\sum_{m=1}^{M} \nabla f_m(x_t^m) \right\rangle \\
&= -\frac{\eta}{2}\mathbb{E}\|\nabla f(x_t)\|^2 - \frac{\eta}{2}\mathbb{E}\left\|\frac{1}{M}\sum_{m=1}^{M}\nabla f_m(x_t^m)\right\|^2 + \frac{\eta}{2}\mathbb{E}\left\|\nabla f(x_t) - \frac{1}{M}\sum_{m=1}^{M}\nabla f_m(x_t^m)\right\|^2 \\
&= -\frac{\eta}{2}\mathbb{E}\|\nabla f(x_t)\|^2 - \frac{\eta}{2}\mathbb{E}\left\|\frac{1}{M}\sum_{m=1}^{M}\nabla f_m(x_t^m)\right\|^2 + \frac{\eta}{2}\mathbb{E}\left\|\frac{1}{M}\sum_{m=1}^{M}\nabla f_m(x_t) - \nabla f_m(x_t^m)\right\|^2 \\
&\leq -\frac{\eta}{2}\mathbb{E}\|\nabla f(x_t)\|^2 - \frac{\eta}{2}\mathbb{E}\left\|\frac{1}{M}\sum_{m=1}^{M}\nabla f_m(x_t^m)\right\|^2 + \frac{\eta}{2M}\sum_{m=1}^{M}\mathbb{E}\|\nabla f_m(x_t) - \nabla f_m(x_t^m)\|^2 \\
&\leq -\frac{\eta}{2}\mathbb{E}\|\nabla f(x_t)\|^2 - \frac{\eta}{2}\mathbb{E}\left\|\frac{1}{M}\sum_{m=1}^{M}\nabla f_m(x_t^m)\right\|^2 + \frac{\eta L^2}{2M}\sum_{m=1}^{M}\underbrace{\mathbb{E}\|x_t - x_t^m\|^2}_{\text{Lemma 3}}.
\end{aligned}
$$

**Step 3b (one step progress).** Bounding term II.

$$
\begin{aligned}
\mathbb{E}\langle \nabla f(z_t) - \nabla f(x_t), z_{t+1} - z_t \rangle &= -\eta\mathbb{E}\left\langle \nabla f(z_t) - \nabla f(x_t), \frac{1}{M}\sum_{m=1}^{M}\nabla f_m(x_t^m) \right\rangle \\
&\leq \frac{\eta\rho}{2}\mathbb{E}\|\nabla f(z_t) - \nabla f(x_t)\|^2 + \frac{\eta}{2\rho}\mathbb{E}\left\|\frac{1}{M}\sum_{m=1}^{M}\nabla f_m(x_t^m)\right\|^2 \\
&\leq \frac{\eta\rho L^2}{2}\underbrace{\mathbb{E}\|z_t - x_t\|^2}_{\text{Lemma 2}} + \frac{\eta}{2\rho}\mathbb{E}\left\|\frac{1}{M}\sum_{m=1}^{M}\nabla f_m(x_t^m)\right\|^2.
\end{aligned}
$$

**Step 3c (one step progress).** Bounding term III.

$$
\begin{aligned}
\frac{L}{2}\mathbb{E}\|z_{t+1} - z_t\|^2 &= \frac{\eta^2 L}{2}\mathbb{E}\left\|\frac{1}{M}\sum_{m=1}^{M} g_t^m\right\|^2 \\
&= \frac{\eta^2 L}{2}\mathbb{E}\left\|\frac{1}{M}\sum_{m=1}^{M} g_t^m - \nabla f_m(x_t^m)\right\|^2 + \frac{\eta^2 L}{2}\mathbb{E}\left\|\frac{1}{M}\sum_{m=1}^{M}\nabla f_m(x_t^m)\right\|^2 \\
&= \frac{\eta^2 L}{2M^2}\sum_{m=1}^{M}\mathbb{E}\|g_t^m - \nabla f_m(x_t^m)\|^2 + \frac{\eta^2 L}{2}\mathbb{E}\left\|\frac{1}{M}\sum_{m=1}^{M}\nabla f_m(x_t^m)\right\|^2 \\
&\leq \frac{\eta^2 L}{2M}\sigma^2 + \frac{\eta^2 L}{2}\mathbb{E}\left\|\frac{1}{M}\sum_{m=1}^{M}\nabla f_m(x_t^m)\right\|^2.
\end{aligned}
$$

Step 3abc (one step progress). Combining previous bounds.

$$
\begin{aligned}
\mathbb{E}f(z_{t+1}) - \mathbb{E}f(z_t) \;\leq\; & \mathbb{E}\underbrace{\langle \nabla f(x_t), z_{t+1} - z_t \rangle}_{I} + \mathbb{E}\underbrace{\langle \nabla f(z_t) - \nabla f(x_t), z_{t+1} - z_t \rangle}_{II} + \mathbb{E}\underbrace{\frac{L}{2}\|z_{t+1} - z_t\|^2}_{III} \\
\leq\; & -\frac{\eta}{2}\mathbb{E}\|\nabla f(x_t)\|^2 - \frac{\eta}{2}\mathbb{E}\left\|\frac{1}{M}\sum_{m=1}^{M}\nabla f_m(x_t^m)\right\|^2 + \frac{\eta L^2}{2M}\sum_{m=1}^{M}\underbrace{\mathbb{E}\|x_t - x_t^m\|^2}_{\text{Lemma 3}} \\
& + \frac{\eta \rho L^2}{2}\underbrace{\mathbb{E}\|z_t - x_t\|^2}_{\text{Lemma 2}} + \frac{\eta}{2\rho}\mathbb{E}\left\|\frac{1}{M}\sum_{m=1}^{M}\nabla f_m(x_t^m)\right\|^2 \\
& + \frac{\eta^2 L}{2K}\sigma^2 + \frac{\eta^2 L}{2}\mathbb{E}\left\|\frac{1}{M}\sum_{m=1}^{M}\nabla f_m(x_t^m)\right\|^2 \\
\leq\; & -\frac{\eta}{2}\mathbb{E}\|\nabla f(x_t)\|^2 - \frac{\eta}{2}\left(1 - \frac{1}{\rho} - \eta L\right)\mathbb{E}\left\|\frac{1}{M}\sum_{m=1}^{M}\nabla f_m(x_t^m)\right\|^2 \\
& + \frac{\eta \rho L^2}{2}\underbrace{\mathbb{E}\|z_t - x_t\|^2}_{\text{Lemma 2}} + \frac{\eta L^2}{2M}\sum_{m=1}^{M}\underbrace{\mathbb{E}\|x_t - x_t^m\|^2}_{\text{Lemma 3}} + \frac{\eta^2 L}{2M}\sigma^2.
\end{aligned}
$$

Step 4 (final). Now we average over the iterates and apply the bounds derived in Lemmas 1,2.

$$
\begin{aligned}
\frac{\mathbb{E}[f(z_T) - f(z_0)]}{T} \;=\; & \frac{1}{T}\sum_{t=0}^{T-1}\mathbb{E}[f(z_{t+1}) - f(z_t)] \\
\leq\; & -\frac{\eta}{2T}\sum_{t=0}^{T-1}\mathbb{E}\|\nabla f(x_t)\|^2 - \frac{\eta}{2}\left(1 - \frac{1}{\rho} - \eta L\right)\frac{1}{T}\sum_{t=0}^{T-1}\mathbb{E}\left\|\frac{1}{M}\sum_{m=1}^{M}\nabla f_m(x_t^m)\right\|^2 \\
& + \frac{\eta \rho L^2}{2}\underbrace{\frac{1}{T}\sum_{t=0}^{T-1}\mathbb{E}\|z_t - x_t\|^2}_{\text{Lemma 1}} + \frac{\eta L^2}{2}\underbrace{\frac{1}{TM}\sum_{t=0}^{T-1}\sum_{m=1}^{M}\mathbb{E}\|x_t - x_t^m\|^2}_{\text{Lemma 2}} + \frac{\eta^2 L}{2M}\sigma^2 \\
\leq\; & -\frac{\eta}{2T}\sum_{t=0}^{T-1}\mathbb{E}\|\nabla f(x_t)\|^2 - \frac{\eta}{2}\left(1 - \frac{1}{\rho} - \eta L\right)\frac{1}{T}\sum_{t=0}^{T-1}\mathbb{E}\left\|\frac{1}{M}\sum_{m=1}^{M}\nabla f_m(x_t^m)\right\|^2 + \frac{\eta^2 L}{2M}\sigma^2 \\
& + \frac{\eta \rho L^2}{2}\left(\frac{\eta^2 \beta^2}{(1-\beta)^2 M}\sigma^2 + \frac{\eta^2 \beta^2}{(1-\beta)^2}\frac{1}{T}\sum_{\tau=0}^{T-1}\mathbb{E}\left\|\frac{1}{M}\sum_{m=1}^{M}\nabla f_m(x_\tau^m)\right\|^2\right) \\
& + \frac{\eta L^2}{2}\left(12\eta^2(B^2 - 1)\psi \cdot \frac{1}{T}\sum_{t=0}^{T-1}\mathbb{E}\|\nabla f(\theta^t)\|^2 + 4\eta^2 \psi(\sigma^2 + 3G^2)\right) \\
\leq\; & -\frac{\eta}{2}\left(1 - 12\eta^2 L^2(B^2 - 1)\psi\right)\frac{1}{T}\sum_{t=0}^{T-1}\mathbb{E}\|\nabla f(x_t)\|^2 \\
& - \frac{\eta}{2}\left(1 - \frac{1}{\rho} - \eta L - \frac{\eta^2 \beta^2 \rho L^2}{(1-\beta)^2}\right)\frac{1}{T}\sum_{t=0}^{T-1}\mathbb{E}\left\|\frac{1}{M}\sum_{m=1}^{M}\nabla f_m(x_t^m)\right\|^2 \\
& + \frac{\eta^2 L}{2M}\sigma^2 + \frac{\eta^3 \rho L^2 \beta^2}{2(1-\beta)^2 M}\sigma^2 + 2\eta^3 L^2 \psi(\sigma^2 + 3G^2).
\end{aligned}
$$

Next, we choose $\rho = 2$ and step size $\eta$ such that

$$12\eta^2 L^2(B^2 - 1)\psi \leq \frac{1}{2} \quad \Longleftrightarrow \quad \text{to bound the first term}$$

$$\eta L + \frac{2\eta^2\beta^2 L^2}{(1-\beta)^2} \leq \frac{1}{2} \quad \Longleftrightarrow \quad \text{to bound the second term}$$

$$12\eta^2 L^2\psi \leq \frac{1}{2} \quad \Longleftrightarrow \quad \text{from Lemma 3}$$

Note that

$$\eta_0 \overset{\text{def}}{=} \frac{1}{4L}\min\left(1 - \beta, \frac{1}{6\sqrt{\psi\max(1, B^2 - 1)}}\right)$$

satisfies all three bounds. Then, with any $\eta \leq \eta_0$ we get

$$
\begin{aligned}
\frac{\mathbb{E}[f(z_T) - f(z_0)]}{T} \quad \leq \quad & -\frac{\eta}{4T}\sum_{t=0}^{T-1}\mathbb{E}\|\nabla f(x_t)\|^2 \\
& + \frac{\eta^2 L}{2M}\sigma^2 + \frac{\eta^3\rho L^2\beta^2}{2(1-\beta)^2 M}\sigma^2 + 2\eta^3 L^2\psi(\sigma^2 + 3G^2).
\end{aligned}
$$

Noticing that $z_0 = x_0$ and $f^* \leq f(z_T)$, we have

$$\frac{1}{T}\sum_{t=0}^{T-1}\mathbb{E}\|\nabla f(x_t)\|^2 \leq \frac{4(f(x_0) - f^*)}{\eta T} + \frac{2\eta L}{M}\sigma^2 + \frac{4\eta^2 L^2\beta^2}{(1-\beta)^2 M}\sigma^2 + 8\eta^2 L^2\psi(\sigma^2 + 3G^2).$$

Furthermore, choosing $\eta = \min(\eta_0, \frac{1}{\sqrt{T}})$, we get the following rate:

$$
\begin{aligned}
& \frac{1}{T}\sum_{t=0}^{T-1}\mathbb{E}\|\nabla f(x_t)\|^2 \\
\leq \quad & \max\left(1, \frac{1}{\eta_0\sqrt{T}}\right)\frac{4(f(x_0) - f^*)}{\sqrt{T}} + \frac{2L\sigma^2}{M\sqrt{T}} + \frac{4L^2\beta^2\sigma^2}{(1-\beta)^2 MT} + \frac{8L^2\psi(\sigma^2 + 3G^2)}{T} \\
\leq \quad & \frac{4(f(x_0) - f^*)}{\sqrt{T}} + \frac{2L\sigma^2}{M\sqrt{T}} + \frac{4(f(x_0) - f^*)}{\eta_0 T} + \frac{4L^2\beta^2\sigma^2}{(1-\beta)^2 MT} + \frac{8L^2\psi(\sigma^2 + 3G^2)}{T} \\
= \quad & \frac{4}{\sqrt{T}}\left(f(x_0) - f^* + \frac{L\sigma^2}{2M}\right) + \mathcal{O}\left(\frac{1 + \psi}{T}\right).
\end{aligned}
$$

### D.1 EXTENSION TO ADAM OPTIMIZER

Here we discuss extension of the previous analysis for the Adam optimizer including the second-order momentum in the analysis. The addition is similar to the first-order momentum while the synchronization probability $p_v$ can differ from other probabilities $p_u$ and $p_u$. The complete description of the algorithm can be found in Algorithm 5. Instead of bounded heterogeneity Assumption 3, in this analysis we use stronger condition mentioned below:

**Assumption 4** (Bounded gradient). *For any iterate $t \geq 0$ and worker $m$, the local stochastic gradient is bounded, namely $\|g_t^m\|_2 \leq G$.*

This condition facilitates the analysis by providing uniform upper bounds for gradients/momentum variables and is commonly used in the analysis of adaptive optimization.

Step 1 (preconditioning and virtual iterates). Let $\Gamma_t^m \overset{\text{def}}{=} \mathbf{diag}^{-1/2}(\tilde{v}_t^m + \lambda^2)$ be the preconditioning matrix and for each step $t \geq 0$, denote the averaged variables

$$x_t \overset{\text{def}}{=} \mathbb{E}_m[x_t^m], \quad u_t \overset{\text{def}}{=} \mathbb{E}_m[u_t^m], \quad v_t \overset{\text{def}}{=} \mathbb{E}_m[v_t^m], \quad \tilde{v}_t \overset{\text{def}}{=} \mathbb{E}_m[\tilde{v}_t^m], \quad g_t \overset{\text{def}}{=} \mathbb{E}_m[g_t^m].$$

Then

$$
\begin{aligned}
u_t &= \beta_1 u_{t-1} + (1 - \beta_1)g_t \\
x_{t+1} &= x_t - d_t = x_t - \eta\mathbb{E}_m[\Gamma_t^m u_t^m].
\end{aligned}
$$

---

**Algorithm 5** `DES-LOC-Adam` (with probabilistic synchronization)

---

**Require: Model tensors**
1:    $x_0 \in \mathbb{R}^d$ — initial parameter vector
2:    $u_{-1}, v_{-1} \in \mathbb{R}^d$ — seeds for first and second moments, initialised to $\mathbf{0}$

**Require: Hyper-parameters**
3:    $\{\eta_t\}_{t=0}^{T-1} \subset \mathbb{R}_{>0}$ — step-size schedule
4:    $\beta_1, \beta_2 \in [0, 1)$ — Adam decay factors
5:    $\lambda \in \mathbb{R}_{\geq 0}$ — $\ell_2$ stability term
6:    $T \in \mathbb{N}_+$ — total optimisation iterations
7:    $M \in \mathbb{N}_+$ — number of workers
8:    $p_x = \frac{1}{K_x}, p_u = \frac{1}{K_u}, p_v = \frac{1}{K_v} \in [0, 1]$ — synchronization probabilities for parameters and momentums

**Ensure:** $x_T, u_{T-1}, v_{T-1}$

9: **for each worker** $m$: $x_0^m = x_0$, $u_{-1}^m = v_{-1}^m = 0$            *local init ($t = -1$ seeds)*

10: **for** $t = 0, \ldots, T - 1$ **do**            *training loop*

11:      **for all** workers $m = 0, \ldots, M - 1$ **in parallel do**

12:          $g_t^m \leftarrow \nabla F(x_t^m; \xi_t^m)$            *stochastic gradient*

13:          $u_t^m \leftarrow \begin{cases} \mathbb{E}_m[\beta_1 u_{t-1}^m + (1 - \beta_1)g_t^m], & \text{with probability } p_u \\ \beta_1 u_{t-1}^m + (1 - \beta_1)g_t^m, & \text{with probability } 1 - p_u \end{cases}$    *sync u*

14:          $v_t^m \leftarrow \begin{cases} \mathbb{E}_m[\beta_2 v_{t-1}^m + (1 - \beta_2)(g_t^m \odot g_t^m)], & \text{with probability } p_v \\ \beta_2 v_{t-1}^m + (1 - \beta_2)(g_t^m \odot g_t^m), & \text{with probability } 1 - p_v \end{cases}$    *sync u*

15:          $\tilde{v}_t^m \leftarrow \max(v_t^m, \tilde{v}_{t-1}^m)$           *AMSGrad Normalization, $\tilde{v}_{-1} = v_{-1}$*

16:          $d_t^m \leftarrow \dfrac{\eta_t}{\sqrt{\tilde{v}_t^m + \lambda^2}} \odot u_t^m$           *bias-corrected update*

17:          $x_{t+1}^m \leftarrow \begin{cases} \mathbb{E}_m[x_t^m - d_t^m], & \text{with probability } p_x \\ x_t^m - d_t^m, & \text{with probability } 1 - p_x \end{cases}$    *sync x*

---

Consider the same averaged iterates $x_t$ and virtual iterates $z_t$ as before:

$$z_t = \frac{1}{1 - \beta_1}x_t - \frac{\beta_1}{1 - \beta_1}x_{t-1}.$$

In particular, $z_0 = x_0$. Then,

$$
\begin{aligned}
z_{t+1} - z_t &= \frac{1}{1 - \beta_1}(x_{t+1} - x_t) - \frac{\beta_1}{1 - \beta_1}(x_t - x_{t-1}) \\
&= -\frac{\eta}{1 - \beta_1}\mathbb{E}_m[\Gamma_t^m u_t^m] + \frac{\eta\beta_1}{1 - \beta_1}\mathbb{E}_m[\Gamma_{t-1}^m u_{t-1}^m] \\
&= -\frac{\eta}{1 - \beta_1}\mathbb{E}_m[\Gamma_t^m u_t^m] + \frac{\eta\beta_1}{1 - \beta_1}\mathbb{E}_m[\Gamma_{t-1}^m u_{t-1}^m] \pm \frac{\eta\beta_1}{1 - \beta_1}\mathbb{E}_m[\Gamma_t^m u_{t-1}^m] \\
&= -\frac{\eta}{1 - \beta_1}\mathbb{E}_m[\Gamma_t^m(u_t^m - \beta_1 u_{t-1}^m)] + \frac{\eta\beta_1}{1 - \beta_1}\mathbb{E}_m[(\Gamma_{t-1}^m - \Gamma_t^m)u_{t-1}^m] \\
&= -\eta\mathbb{E}_m[\Gamma_t^m \widetilde{g}_t^m] + \frac{\eta\beta_1}{1 - \beta_1}\mathbb{E}_m[(\Gamma_{t-1}^m - \Gamma_t^m)u_{t-1}^m] \\
&= -\eta\mathbb{E}_m[\Gamma_t^m g_t] + \eta\mathbb{E}_m[\Gamma_t^m(g_t - \widetilde{g}_t^m)] + \frac{\eta\beta_1}{1 - \beta_1}\mathbb{E}_m[(\Gamma_{t-1}^m - \Gamma_t^m)u_{t-1}^m] \\
&= -\eta\Gamma_t g_t + \eta \cdot \underbrace{\mathbb{E}_m[\Gamma_t^m(g_t - \widetilde{g}_t^m)]}_{\overset{\text{def}}{=}U_t} + \eta \cdot \underbrace{\frac{\beta_1}{1 - \beta_1}\mathbb{E}_m[(\Gamma_{t-1}^m - \Gamma_t^m)u_{t-1}^m]}_{\overset{\text{def}}{=}V_t},
\end{aligned}
$$

where $\Gamma_t \overset{\text{def}}{=} \mathbb{E}_m[\Gamma_t^m]$ and $\widetilde{g}_t^m \overset{\text{def}}{=} \frac{u_t^m - \beta_1 u_{t-1}^m}{1 - \beta_1}$ for which, $\mathbb{E}_m[\widetilde{g}_t^m] = \mathbb{E}_m[g_t^m] = g_t$.

Step 2 (smoothness over virtual iterates). Then we apply smoothness of the global loss function $f$ over these global virtual iterates.

$$
\begin{aligned}
f(z_{t+1}) - f(z_t) &\leq \langle \nabla f(z_t), z_{t+1} - z_t \rangle + \frac{L}{2}\|z_{t+1} - z_t\|^2 \\
&= -\eta\langle \nabla f(z_t), \Gamma_t g_t \rangle + \eta\langle \nabla f(z_t), U_t \rangle + \eta\langle \nabla f(z_t), V_t \rangle + \frac{L}{2}\|z_{t+1} - z_t\|^2 \\
&= \underbrace{-\eta\langle \nabla f(x_t), \Gamma_t g_t \rangle}_{I} + \underbrace{\eta\langle \nabla f(z_t), U_t \rangle}_{II} + \underbrace{\eta\langle \nabla f(z_t), V_t \rangle}_{III} \\
&\quad + \underbrace{\frac{\eta^2 L}{2}\|\Gamma_t g_t - U_t - V_t\|^2}_{IV} + \underbrace{\eta\langle \nabla f(x_t) - \nabla f(z_t), \Gamma_t g_t \rangle}_{V}.
\end{aligned}
$$

In the next step, we separately bound each term appearing in the above bound. For clarity, we are also going to use $\|\nabla f(x_t)\| \leq G$ and $\|\nabla f(z_t)\| \leq G$. However, these conditions can be avoided through linking $\nabla f(z_t)$ term to $\nabla f(x_t)$, and $\nabla f(x_t)$ term to $\mathbb{E}_m \nabla f_m(x_t^m)$ with the bound for $\mathbb{E}[\|x_t - x_t^m\|^2]$.

Step 3a (one step progress). Bounding term I.

$$
\begin{aligned}
I &= -\eta\langle \nabla f(x_t), \Gamma_t g_t]\rangle \\
&= -\eta\mathbb{E}\left[\langle \nabla f(x_t), \Gamma_{t-1} g_t \rangle\right] + \eta\mathbb{E}\left[\langle \nabla f(x_t), (\Gamma_{t-1} - \Gamma_t)g_t \rangle\right] \\
&\leq -\eta\mathbb{E}\left[\left\langle \nabla f(x_t), \frac{1}{M}\sum_{m=1}^{M} \nabla f_m(x_t^m)\right\rangle_{\Gamma_{t-1}}\right] + \eta G^2 \mathbb{E}[\|\Gamma_{t-1} - \Gamma_t\|]. \\
&\leq -\frac{\eta}{2}\mathbb{E}\left[\|\nabla f(x_t)\|_{\Gamma_{t-1}}^2\right] - \frac{\eta}{2}\mathbb{E}\left[\left\|\frac{1}{M}\sum_{m=1}^{M} \nabla f_m(x_t^m)\right\|_{\Gamma_{t-1}}^2\right] \\
&\quad + \frac{\eta}{2}\mathbb{E}\left[\left\|\nabla f(x_t) - \frac{1}{M}\sum_{m=1}^{M} \nabla f_m(x_t^m)\right\|_{\Gamma_{t-1}}^2\right] + \eta G^2 \mathbb{E}[\|\Gamma_{t-1} - \Gamma_t\|] \\
&\leq -\frac{\eta}{2}\|\Gamma_{t-1}\|_{\min}\mathbb{E}\|\nabla f(x_t)\|^2 - \frac{\eta}{2}\mathbb{E}\left[\left\|\frac{1}{M}\sum_{m=1}^{M} \nabla f_m(x_t^m)\right\|_{\Gamma_{t-1}}^2\right] \\
&\quad + \frac{\eta}{2}\|\Gamma_{t-1}\|_{\max}\mathbb{E}\left[\left\|\frac{1}{M}\sum_{m=1}^{M} \nabla f_m(x_t) - \nabla f_m(x_t^m)\right\|^2\right] + \eta G^2 \mathbb{E}[\|\Gamma_{t-1} - \Gamma_t\|] \\
&\leq -\frac{\eta}{2C_0}\mathbb{E}\|\nabla f(x_t)\|^2 - \frac{\eta}{2}\mathbb{E}\left[\left\|\frac{1}{M}\sum_{m=1}^{M} \nabla f_m(x_t^m)\right\|_{\Gamma_{t-1}}^2\right] \\
&\quad + \frac{\eta}{2\lambda M}\sum_{m=1}^{M}\mathbb{E}\left[\|\nabla f_m(x_t) - \nabla f_m(x_t^m)\|^2\right] + \eta G^2 \mathbb{E}[\|\Gamma_{t-1} - \Gamma_t\|] \\
&\leq -\frac{\eta}{2C_0}\mathbb{E}\|\nabla f(x_t)\|^2 + \frac{\eta L^2}{2\lambda M}\sum_{m=1}^{M}\mathbb{E}\left[\|x_t - x_t^m\|^2\right] + \eta G^2 \mathbb{E}[\|\Gamma_{t-1} - \Gamma_t\|],
\end{aligned}
$$

where $\|\cdot\|$ indicates the spectral norm for matrices, and we used the following inequalities:

$$
\|\Gamma_{t-1}\|_{\min} = \left\|\frac{1}{M}\sum_{m=1}^{M}\Gamma_{t-1}^m\right\|_{\min} = \frac{1}{M}\sum_{m=1}^{M}\Gamma_{t-1}^m[i,i] = \frac{1}{M}\sum_{m=1}^{M}\frac{1}{\sqrt{\tilde{v}_{t-1}[i]} + \lambda^2} \geq \frac{1}{\sqrt{G^2 + \lambda^2}} \stackrel{\text{def}}{=} \frac{1}{C_0}.
$$

Step 3b (one step progress). Bounding term II.

$$II \;=\; \eta \langle \nabla f(z_t), U_t \rangle \le \eta \|\nabla f(z_t)\| \|U_t\| \le \frac{\eta G}{M} \sum_{m=1}^{M} \|\Gamma_t^m (g_t - \widetilde{g}_t^m)\|$$

$$\le \;\; \frac{\eta G}{\lambda M} \sum_{m=1}^{M} \|g_t - \widetilde{g}_t^m\|.$$

Step 3c (one step progress). Bounding term III.

$$III \;=\; \eta \langle \nabla f(z_t), V_t \rangle \le \eta \|\nabla f(z_t)\| \|V_t\| \le \frac{\eta \beta_1}{1-\beta_1} \frac{G}{M} \sum_{m=1}^{M} \|(\Gamma_{t-1}^m - \Gamma_t^m) u_{t-1}^m\|$$

$$\le \;\; \frac{\eta \beta_1}{1-\beta_1} \frac{G^2}{M} \sum_{m=1}^{M} \|\Gamma_{t-1}^m - \Gamma_t^m\|.$$

Step 3d (one step progress). Bounding term IV.

$$IV \;=\; \frac{\eta^2 L}{2} \|\Gamma_t g_t - U_t - V_t\|^2$$

$$\le \;\; \frac{3\eta^2 L}{2} \|\Gamma_t g_t\|^2 + \frac{3\eta^2 L}{2} \|U_t\|^2 + \frac{3\eta^2 L}{2} \|V_t\|^2$$

$$\le \;\; \frac{3\eta^2 L G^2}{2\lambda^2} + \frac{3\eta^2 L}{2\lambda^2 M} \sum_{m=1}^{M} \|g_t - \widetilde{g}_t^m\|^2 + \frac{3\eta^2 \beta_1 L G}{2(1-\beta_1) M} \sum_{m=1}^{M} \|\Gamma_{t-1}^m - \Gamma_t^m\|^2$$

Step 3e (one step progress). Bounding term V.

$$\begin{aligned}
V \;=\;& \eta \left\langle \nabla f(x_t) - \nabla f(z_t), \Gamma_t g_t \right\rangle \\
=\;& \eta \mathbb{E}\left[ \langle \nabla f(x_t) - \nabla f(z_t), \Gamma_{t-1} g_t \rangle \right] + \eta \mathbb{E}\left[ \langle \nabla f(x_t) - \nabla f(z_t), (\Gamma_t - \Gamma_{t-1}) g_t \rangle \right] \\
\le\;& \eta \mathbb{E}\left[ \left\langle \nabla f(x_t) - \nabla f(z_t), \frac{1}{M} \sum_{m=1}^{M} \nabla f_m(x_t^m) \right\rangle_{\Gamma_{t-1}} \right] + \frac{\eta^2 L \beta_1}{1-\beta_1} \mathbb{E}\left[ \|\mathbb{E}_m[\Gamma_{t-1}^m u_{t-1}^m]\| \, \|(\Gamma_t - \Gamma_{t-1}) g_t\| \right] \\
\le\;& \eta \mathbb{E}\left[ \langle \nabla f(x_t) - \nabla f(z_t), \nabla f(x_t) \rangle_{\Gamma_{t-1}} \right] \\
& + \eta \mathbb{E}\left[ \left\langle \nabla f(x_t) - \nabla f(z_t), \frac{1}{M} \sum_{m=1}^{M} \nabla f_m(x_t^m) - \nabla f_m(x_t) \right\rangle_{\Gamma_{t-1}} \right] + \frac{\eta^2 L \beta_1 G^2}{(1-\beta_1)\lambda} \mathbb{E}\left[ \|\Gamma_t - \Gamma_{t-1}\| \right] \\
\le\;& \frac{\eta}{\lambda} \mathbb{E}\left[ \|\nabla f(x_t) - \nabla f(z_t)\| \|\nabla f(x_t)\| \right] \\
& + \frac{\eta}{\lambda} \mathbb{E}\left[ \|\nabla f(x_t) - \nabla f(z_t)\| \cdot \frac{1}{M} \sum_{m=1}^{M} \|\nabla f_m(x_t^m) - \nabla f_m(x_t)\| \right] + \frac{\eta^2 L \beta_1 G^2}{(1-\beta_1)\lambda} \mathbb{E}\left[ \|\Gamma_t - \Gamma_{t-1}\| \right] \\
\le\;& \frac{\eta}{\lambda} \mathbb{E}\left[ \frac{1}{2\rho} \|\nabla f(x_t) - \nabla f(z_t)\|^2 + \frac{\rho}{2} \|\nabla f(x_t)\|^2 \right] \\
& + \frac{\eta}{\lambda} \mathbb{E}\left[ \frac{1}{2} \|\nabla f(x_t) - \nabla f(z_t)\|^2 + \frac{1}{2} \frac{L^2}{M} \sum_{m=1}^{M} \|x_t^m - x_t\|^2 \right] + \frac{\eta^2 L \beta_1 G^2}{(1-\beta_1)\lambda} \mathbb{E}\left[ \|\Gamma_t - \Gamma_{t-1}\| \right],
\end{aligned}$$

where we used the following uniform bound on $\|\nabla f(x_t) - \nabla f(z_t)\|$:

$$\begin{aligned}
\|\nabla f(x_t) - \nabla f(z_t)\| \;\le\;& L \|x_t - z_t\| \le \frac{\beta_1 L}{1-\beta_1} \|x_t - x_{t-1}\| = \frac{\eta \beta_1 L}{1-\beta_1} \|\mathbb{E}_m[\Gamma_{t-1}^m u_{t-1}^m]\| \\
\le\;& \frac{\eta \beta_1 L}{1-\beta_1} \mathbb{E}_m[\|\Gamma_{t-1}^m\| \|u_{t-1}^m\|] \le \frac{\eta \beta_1 L}{1-\beta_1} \frac{G}{\lambda}.
\end{aligned}$$

Therefore, ignoring the constants, we have the following bounds:

$$
\begin{aligned}
V &\leq \mathcal{O}\left(\frac{\eta^2}{\rho}\right) + \frac{\eta\rho}{2\lambda} \cdot \mathbb{E}[\|\nabla f(x_t)\|^2] + \mathcal{O}\left(\eta\right) \cdot \frac{1}{M}\sum_{m=1}^{M}\mathbb{E}[\|x_t^m - x_t\|^2] + \mathcal{O}\left(\eta^2\right) \\
IV &\leq \mathcal{O}\left(\eta^2\right) \\
III &\leq \mathcal{O}\left(\eta\right) \cdot \frac{1}{M}\sum_{m=1}^{M}\mathbb{E}[\|\Gamma_{t-1}^m - \Gamma_t^m\|] \\
II &\leq \mathcal{O}\left(\eta\right) \cdot \frac{1}{M}\sum_{m=1}^{M}\mathbb{E}[\|g_t - \widetilde{g}_t^m\|] \\
I &\leq -\frac{\eta}{2C_0}\mathbb{E}\|\nabla f(x_t)\|^2 + \mathcal{O}\left(\eta\right) \cdot \frac{1}{M}\sum_{m=1}^{M}\mathbb{E}\left[\|x_t - x_t^m\|^2\right] + \mathcal{O}\left(\eta\right) \cdot \frac{1}{M}\sum_{m=1}^{M}\mathbb{E}[\|\Gamma_{t-1}^m - \Gamma_t^m\|]
\end{aligned}
$$

To get the $\mathcal{O}\left(\frac{1}{\sqrt{T}}\right)$ bound for the averaged gradients $\mathbb{E}[\|\nabla f(x_t)\|^2]$, note that we are left to choose small value for $\rho = \frac{\lambda}{2C_0}$ and show the following bounds:

$$
\begin{aligned}
&\frac{1}{TM}\sum_{t=0}^{T-1}\sum_{m=1}^{M}\mathbb{E}[\|x_t^m - x_t\|^2] = \mathcal{O}(\eta^2), && \text{(extension of Lemma 3)} \\
&\sum_{t=0}^{T-1}\mathbb{E}[\|\Gamma_{t-1}^m - \Gamma_t^m\|] = \mathcal{O}(1), && \text{(follows from AMSGrad normalization)} \\
&\frac{1}{M}\sum_{t=0}^{T-1}\sum_{m=1}^{M}\mathbb{E}[\|g_t - \widetilde{g}_t^m\|] = \mathcal{O}(1), && \text{(see below).}
\end{aligned}
$$

For the last bound, we can use similar steps as in Lemma 3, namely

$$
\begin{aligned}
\mathbb{E}[\|u_t - u_t^m\|] &= p_u \cdot 0 + (1-p_u)\mathbb{E}[\|\beta_1 u_{t-1} + (1-\beta_1)g_t - (\beta_1 u_{t-1}^m + (1-\beta_1)g_t^m)\|] \\
&\leq (1-p_u)\beta_1\mathbb{E}[\|u_{t-1} - u_{t-1}^m)\|] + (1-p_u)(1-\beta_1)\mathbb{E}[\|g_t - g_t^m)\|] \\
&\leq (1-p_u)(1-\beta_1)\sum_{\tau=0}^{t}((1-p_u)\beta_1)^{t-\tau}\mathbb{E}[\|g_\tau - g_\tau^m\|]. \\
\mathbb{E}[\|g_t - \widetilde{g}_t^m\|] &= \mathbb{E}\left\|\frac{u_t - \beta_1 u_{t-1}}{1-\beta_1} - \frac{u_t^m - \beta_1 u_{t-1}^m}{1-\beta_1}\right\| \\
&\leq \frac{\beta_1}{1-\beta_1}\mathbb{E}\|u_{t-1} - u_{t-1}^m\| + \frac{1}{1-\beta_1}\mathbb{E}[\|u_t - u_t^m\|] \\
&= \frac{1}{1-\beta_1}\sum_{\tau=t-1}^{t}\beta_1^{t-\tau}\mathbb{E}\|u_\tau - u_\tau^m\| \\
&= (1-p_u)\sum_{\tau=t-1}^{t}\sum_{\nu=0}^{\tau}\beta_1^{t-\tau}((1-p_u)\beta_1)^{\tau-\nu}\mathbb{E}[\|g_\nu - g_\nu^m\|] \\
&= \sum_{\tau=t}^{t+1}\sum_{\nu=0}^{\tau-1}\beta_1^{t-\tau}\underbrace{((1-p_u)\beta_1)^{\tau-\nu}}_{=q_2}\mathbb{E}[\|g_\nu - g_\nu^m\|],
\end{aligned}
$$

which has the same double geometric sum structure as (7).

### D.2 KEY LEMMAS

**Lemma 2.** *For all $T \geq 1$, we have*

$$\sum_{t=0}^{T-1} \|z_t - x_t\|^2 \leq \frac{\eta^2 \beta^2}{(1-\beta)^2 M} T\sigma^2 + \frac{\eta^2 \beta^2}{(1-\beta)^2} \sum_{t=0}^{T-1} \mathbb{E} \left\| \frac{1}{M} \sum_{m=1}^{M} \nabla f_m(x_t^m) \right\|^2. \tag{6}$$

*Proof.* Since $u_{-1} = 0$, unrolling the update rule of momentum, for any $t \geq 0$ we get

$$u_t = \beta u_{t-1} + (1-\beta)g_t = (1-\beta) \sum_{\tau=0}^{t} \beta^{t-\tau} g^\tau.$$

Using this and the definition of the average iterates, we have

$$z_t - x_t = \frac{\beta}{1-\beta}(x_t - x_{t-1}) = -\frac{\beta\eta}{1-\beta} u_t = -\beta\eta \sum_{\tau=0}^{t} \beta^{t-\tau} g_\tau.$$

Using convexity of squared norm function and letting $s_t \stackrel{\text{def}}{=} \sum_{\tau=0}^{t} \beta^{t-\tau} = \frac{1-\beta^{t+1}}{1-\beta}$, for all $t \geq 0$, we have

$$\|z_t - x_t\|^2 = \eta^2 \beta^2 s_t^2 \left\| \sum_{\tau=0}^{t} \frac{\beta^{t-\tau}}{s_t} g_\tau \right\|^2 \leq \eta^2 \beta^2 s_t^2 \sum_{\tau=0}^{t} \frac{\beta^{t-\tau}}{s_t} \|g_\tau\|^2 \leq \frac{\eta^2 \beta^2}{1-\beta} \sum_{\tau=0}^{t} \beta^{t-\tau} \|g_\tau\|^2.$$

Summing over the iterates yields

$$
\begin{aligned}
\sum_{t=0}^{T-1} \mathbb{E}\|z_t - x_t\|^2 &\leq \frac{\eta^2 \beta^2}{1-\beta} \sum_{t=0}^{T-1} \sum_{\tau=0}^{t} \beta^{t-\tau} \mathbb{E}\|g_\tau\|^2 \\
&= \frac{\eta^2 \beta^2}{1-\beta} \sum_{\tau=0}^{T-1} \sum_{t=\tau}^{T-1} \beta^{t-\tau} \mathbb{E}\|g_\tau\|^2 \\
&= \frac{\eta^2 \beta^2}{1-\beta} \sum_{\tau=0}^{T-1} \frac{1-\beta^{T-\tau}}{1-\beta} \mathbb{E}\|g_\tau\|^2 \\
&\leq \frac{\eta^2 \beta^2}{(1-\beta)^2} \sum_{\tau=0}^{T-1} \mathbb{E}\|g_\tau\|^2 \\
&= \frac{\eta^2 \beta^2}{(1-\beta)^2} \sum_{\tau=0}^{T-1} \mathbb{E} \left\| \frac{1}{M} \sum_{m=1}^{M} g_\tau^m - \nabla f_m(x_\tau^m) \right\|^2 + \frac{\eta^2 \beta^2}{(1-\beta)^2} \sum_{\tau=0}^{T-1} \mathbb{E} \left\| \frac{1}{M} \sum_{m=1}^{M} \nabla f_m(x_\tau^m) \right\|^2 \\
&= \frac{\eta^2 \beta^2}{(1-\beta)^2 M^2} \sum_{\tau=0}^{T-1} \sum_{m=1}^{M} \mathbb{E} \|g_\tau^m - \nabla f_m(x_\tau^m)\|^2 + \frac{\eta^2 \beta^2}{(1-\beta)^2} \sum_{\tau=0}^{T-1} \mathbb{E} \left\| \frac{1}{M} \sum_{m=1}^{M} \nabla f_m(x_\tau^m) \right\|^2 \\
&= \frac{\eta^2 \beta^2}{(1-\beta)^2 M} T\sigma^2 + \frac{\eta^2 \beta^2}{(1-\beta)^2} \sum_{\tau=0}^{T-1} \mathbb{E} \left\| \frac{1}{M} \sum_{m=1}^{M} \nabla f_m(x_\tau^m) \right\|^2.
\end{aligned}
$$

$\square$

**Lemma 3.** *If $24\eta^2 L^2 \psi \leq 1$, then*

$$\frac{1}{MT} \sum_{t=0}^{T-1} \sum_{m=1}^{M} \mathbb{E}\|x_t - x_t^m\|^2 \leq 12\eta^2(B^2-1)\psi \cdot \frac{1}{T} \sum_{t=0}^{T-1} \mathbb{E}\|\nabla f(x_t)\|^2 + 4\eta^2\psi(\sigma^2 + 3G^2),$$

*where*

$$\psi = \frac{4(1-p_x)}{p_x^2} \cdot \frac{(1-\beta)(1-p_u)}{1-(1-p_u)\beta}$$

*Proof.* Let us expand the term $\mathbb{E}\|x_{t+1} - x_{t+1}^m\|^2$ using $x_{t+1}^m$'s probabilistic update rule:

$$
\begin{aligned}
\mathbb{E}\|x_{t+1} - x_{t+1}^m\|^2 &= p_x \cdot 0 + (1 - p_x) \cdot \mathbb{E}\|x_t - \eta u_t - (x_t^m - \eta u_t^m)\|^2 \\
&= (1 - p_x) \cdot \mathbb{E}\|x_t - x_t^m - \eta(u^t - u_t^m)\|^2 \\
&\leq (1 - p_x)(1 + s)\mathbb{E}\|x_t - x_t^m\|^2 + \eta^2(1 - p_x)(1 + 1/s)\mathbb{E}\|u_t - u_t^m\|^2 \\
&\leq \eta^2(1 - p_x)(1 + 1/s)\sum_{\tau=1}^{t}((1 - p_x)(1 + s))^{t-\tau}\mathbb{E}\|u_\tau - u_\tau^m\|^2.
\end{aligned}
$$

where $s > 0$ will be chosen later. Next we expand the term $\mathbb{E}\|u_t - u_t^m\|^2$ using $u_t^m$'s probabilistic update rule:

$$
\begin{aligned}
\mathbb{E}\|u_t - u_t^m\|^2 &= p_u \cdot 0 + (1 - p_u) \cdot \mathbb{E}\left\|\frac{1}{M}\sum_{m=1}^{M}(\beta u_{t-1}^m + (1 - \beta)g_{t-1}^m) - (\beta u_{t-1}^m + (1 - \beta)g_{t-1}^m)\right\|^2 \\
&= (1 - p_u)\mathbb{E}\|\beta(u_{t-1} - u_{t-1}^m) + (1 - \beta)(g_{t-1} - g_{t-1}^m)\|^2 \\
&\leq (1 - p_u)\beta\mathbb{E}\|(u_{t-1} - u_{t-1}^m)\|^2 + (1 - p_u)(1 - \beta)\mathbb{E}\|g_{t-1} - g_{t-1}^m\|^2 \\
&\leq (1 - p_u)(1 - \beta)\sum_{\tau=0}^{t-1}((1 - p_u)\beta)^{t-1-\tau}\mathbb{E}\|g_\tau - g_\tau^m\|^2 \\
&\leq \frac{1 - \beta}{\beta}\sum_{\tau=0}^{t-1}((1 - p_u)\beta)^{t-\tau}\mathbb{E}\|g_\tau - g_\tau^m\|^2
\end{aligned}
$$

Denote $q_1 = (1 - p_x)(1 + s)$ and $q_2 = (1 - p_u)\beta$. Combining the previous two bounds, we get

$$
\begin{aligned}
&\frac{1}{M}\sum_{m=1}^{M}\mathbb{E}\|x_t - x_t^m\|^2 \\
&\leq \eta^2(1 - p_x)(1 + 1/s)\sum_{\tau=1}^{t}((1 - p)(1 + s))^{t-\tau}\frac{1}{M}\sum_{m=1}^{M}\mathbb{E}\|u_\tau - u_\tau^m\|^2 \qquad (7) \\
&\leq \eta^2(1 - p_x)(1 + 1/s)\sum_{\tau=1}^{t}((1 - p_u)(1 + s))^{t-\tau}\frac{1}{M}\sum_{m=1}^{M}\left[\frac{1 - \beta}{\beta}\sum_{\nu=0}^{\tau-1}((1 - p_u)\beta)^{\tau-\nu}\mathbb{E}\|g_\nu - g_\nu^m\|^2\right] \\
&= \eta^2(1 - p_x)(1 + 1/s)\frac{1 - \beta}{\beta}\sum_{\tau=1}^{t}\sum_{\nu=0}^{\tau-1}q_1^{t-\tau}q_2^{\tau-\nu}\left[\frac{1}{M}\sum_{m=1}^{M}\mathbb{E}\|g_\nu - g_\nu^m\|^2\right] \\
&= \eta^2(1 - p_x)(1 + 1/s)\frac{1 - \beta}{\beta}\sum_{\nu=0}^{t-1}\sum_{\tau=\nu+1}^{t}q_1^{t-\tau}q_2^{\tau-\nu}\left[\frac{1}{M}\sum_{m=1}^{M}\mathbb{E}\|g_\nu - g_\nu^m\|^2\right] \\
&= \eta^2(1 - p_x)(1 + 1/s)\frac{1 - \beta}{\beta}\sum_{\nu=0}^{t-1}q_2\frac{q_1^{t-\nu} - q_2^{t-\nu}}{q_1 - q_2}\left[\frac{1}{M}\sum_{m=1}^{M}\mathbb{E}\|g_\nu - g_\nu^m\|^2\right], \\
&= \eta^2\underbrace{(1 - p_x)(1 + 1/s)(1 - \beta)(1 - p_u)}_{\stackrel{\text{def}}{=}\phi}\sum_{\nu=0}^{t-1}\frac{q_1^{t-\nu} - q_2^{t-\nu}}{q_1 - q_2}\left[\frac{1}{M}\sum_{m=1}^{M}\mathbb{E}\|g_\nu - g_\nu^m\|^2\right].
\end{aligned}
$$

Next, we bound the gradient term above.

$$
\begin{aligned}
\frac{1}{M}\sum_{m=1}^{M}\mathbb{E}\|g_t^m - g_t\|^2 
&= \frac{1}{M}\sum_{m=1}^{M}\mathbb{E}\left\|g_t^m - \frac{1}{M}\sum_{i=1}^{K}g_i^t\right\|^2 \\
&\leq \frac{2}{K}\sum_{m=1}^{M}\mathbb{E}\left\|g_t^m - \nabla f_m(x_t^m) - \frac{1}{M}\sum_{m=1}^{M}(g_t^m - \nabla f_m(x_t^m))\right\|^2 \\
&\quad + \frac{2}{M}\sum_{m=1}^{M}\mathbb{E}\left\|\nabla f_m(x_t^m) - \frac{1}{M}\sum_{m=1}^{M}\nabla f_m(x_t^m)\right\|^2 \\
(\text{Lemma } 4) \quad &\leq \frac{2}{M}\sum_{m=1}^{M}\mathbb{E}\|g_t^m - \nabla f_m(x_t^m)\|^2 - 2\mathbb{E}\left\|\frac{1}{M}\sum_{m=1}^{M}(g_t^m - \nabla f_m(x_t^m))\right\|^2 \\
&\quad + \frac{12L^2}{M}\sum_{m=1}^{M}\mathbb{E}\|x_t - x_t^m\|^2 + 6(B^2 - 1)\mathbb{E}\|\nabla f(x_t)\|^2 + 6G^2 \\
&\leq 2\sigma^2 + \frac{12L^2}{M}\sum_{m=1}^{M}\mathbb{E}\|x_t - x_t^m\|^2 + 6(B^2 - 1)\mathbb{E}\|\nabla f(x_t)\|^2 + 6G^2.
\end{aligned}
$$

Again, plugging this bound to the previous one, we get

$$
\begin{aligned}
&\frac{1}{MT}\sum_{t=0}^{T-1}\sum_{m=1}^{M}\mathbb{E}\|x_t - x_t^m\|^2 \\
&\leq \frac{1}{MT}\sum_{t=1}^{T}\sum_{m=1}^{M}\mathbb{E}\|x_t - x_t^m\|^2 \\
&\leq \frac{\eta^2\phi}{T}\sum_{t=1}^{T}\sum_{\tau=0}^{t-1}\frac{q_1^{t-\tau} - q_2^{t-\tau}}{q_1 - q_2}\left[\frac{1}{M}\sum_{m=1}^{M}\mathbb{E}\|g_\tau - g_\tau^m\|^2\right] \\
&= \frac{\eta^2\phi}{T}\sum_{\tau=0}^{T-1}\sum_{t=\tau+1}^{T}\frac{q_1^{t-\tau} - q_2^{t-\tau}}{q_1 - q_2}\left[\frac{1}{M}\sum_{m=1}^{M}\mathbb{E}\|g_\tau - g_\tau^m\|^2\right] \\
&= \frac{\eta^2\phi}{T}\sum_{\tau=0}^{T-1}\frac{1}{q_1 - q_2}\left(\frac{q_1(1 - q_1^{T-\tau})}{1 - q_1} - \frac{q_2(1 - q_2^{T-\tau})}{1 - q_2}\right)\left[\frac{1}{M}\sum_{m=1}^{M}\mathbb{E}\|g_\tau - g_\tau^m\|^2\right] \\
&\leq \frac{\eta^2\phi}{T}\sum_{\tau=0}^{T-1}\frac{1}{q_1 - q_2}\left(\frac{q_1}{1 - q_1} - \frac{q_2}{1 - q_2}\right)\left[\frac{1}{M}\sum_{m=1}^{M}\mathbb{E}\|g_\tau - g_\tau^m\|^2\right] \\
&= \frac{\eta^2\phi}{(1 - q_1)(1 - q_2)}\frac{1}{T}\sum_{\tau=0}^{T-1}\left[\frac{1}{M}\sum_{m=1}^{M}\mathbb{E}\|g_\tau - g_\tau^m\|^2\right].
\end{aligned}
$$

Now, let us optimize the factor

$$
\frac{\phi}{(1 - q_1)(1 - q_2)} = \frac{(1 - p_x)(1 + 1/s)(1 - \beta)(1 - p_u)}{(1 - (1 - p_x)(1 + s))(1 - (1 - p_u)\beta)} = \frac{(1 - p_x)(1 + 1/s)}{1 - (1 - p_x)(1 + s)} \cdot \frac{(1 - \beta)(1 - p_u)}{1 - (1 - p_u)\beta}
$$

by choosing optimal value for $s$ introduced earlier. By the first order optimality condition, we find that the optimal value is $s^* = \frac{1}{\sqrt{1-p_x}} - 1$. Hence, the minimal value of the factor is

$$
\begin{aligned}
\frac{\phi}{(1-q_1)(1-q_2)} &= \frac{1-p_x}{(1-\sqrt{1-p_x})^2} \cdot \frac{(1-\beta)(1-p_u)}{1-(1-p_u)\beta} \\
&= \frac{(1-p_x)(1-\sqrt{1-p_x})^2}{(1-\sqrt{1-p_x})^2(1+\sqrt{1-p_x})^2} \cdot \frac{(1-\beta)(1-p_u)}{1-(1-p_u)\beta} \\
&= \frac{(1-p_x)(1+\sqrt{1-p_x})^2}{p_x^2} \cdot \frac{(1-\beta)(1-p_u)}{1-(1-p_u)\beta} \\
&\leq \frac{4(1-p_x)}{p_x^2} \cdot \frac{(1-\beta)(1-p_u)}{1-(1-p_u)\beta} \stackrel{\text{def}}{=} \psi.
\end{aligned}
$$

Continuing the chain of bounds

$$
\begin{aligned}
&\frac{1}{MT} \sum_{t=0}^{T-1} \sum_{m=1}^{M} \mathbb{E}\|x_t - x_t^m\|^2 \\
\leq\ & \eta^2 \psi \cdot \frac{1}{T} \sum_{t=0}^{T-1} \left[ \frac{1}{K} \sum_{m=1}^{M} \mathbb{E}\|g_t - g_t^m\|^2 \right] \\
\leq\ & \eta^2 \psi \cdot \frac{1}{T} \sum_{t=0}^{T-1} \left[ \frac{12L^2}{M} \sum_{m=1}^{M} \mathbb{E}\|x_t - x_t^m\|^2 + 6(B^2-1)\mathbb{E}\|\nabla f(x_t)\|^2 + 2\sigma^2 + 6G^2 \right] \\
\leq\ & 12\eta^2 L^2 \psi \cdot \frac{1}{TM} \sum_{t=0}^{T-1} \sum_{m=1}^{M} \mathbb{E}\|x_t - x_t^m\|^2 \\
& + 6\eta^2(B^2-1)\psi \cdot \frac{1}{T} \sum_{t=0}^{T-1} \mathbb{E}\|\nabla f(x_t)\|^2 + 2\eta^2 \psi (\sigma^2 + 3G^2).
\end{aligned}
$$

Assuming $12\eta^2 L^2 \psi \leq 1/2$ and reordering the first term in the bound, we arrive

$$
\frac{1}{MT} \sum_{t=0}^{T-1} \sum_{m=1}^{M} \mathbb{E}\|x_t - x_t^m\|^2 \leq 12\eta^2(B^2-1)\psi \cdot \frac{1}{T} \sum_{t=0}^{T-1} \mathbb{E}\|\nabla f(x_t)\|^2 + 4\eta^2 \psi (\sigma^2 + 3G^2).
$$

$\square$

**Lemma 4.** *Under smoothness and bounded heterogeneity assumptions 1 and 3, we have*

$$
\frac{1}{M} \sum_{m=1}^{M} \left\| \nabla f_m(x_t^m) - \frac{1}{K} \sum_{i=1}^{K} \nabla f_i(x_t^i) \right\|^2 \leq \frac{6L^2}{M} \sum_{m=1}^{M} \|x_t - x_t^m\|^2 + 3(B^2-1)\|\nabla f(x_t)\|^2 + 3G^2.
$$

*Proof.* The bound follows from simple algebraic manipulations and Jensen's inequality.

$$
\frac{1}{K} \sum_{m=1}^{M} \|\nabla f_m(x_t^m) - \frac{1}{K} \sum_{i=1}^{N} \nabla f_i(x_t^i)\|^2
$$

$$
= \frac{1}{K} \sum_{m=1}^{M} \left\| \nabla f_m(x_t^m) - \nabla f_m(x_t) + \nabla f_m(x_t) - \nabla f(x_t) + \nabla f(x_t) - \frac{1}{K} \sum_{i=1}^{N} \nabla f_i(x_t^i) \right\|^2
$$

$$
\leq \frac{3}{K} \sum_{m=1}^{M} \|\nabla f_m(x_t^m) - \nabla f_m(x_t)\|^2 + \frac{3}{K} \sum_{m=1}^{M} \|\nabla f_m(x_t) - \nabla f(x_t)\|^2
$$

$$
+ \frac{3}{K} \sum_{m=1}^{M} \left\| \nabla f(x_t) - \frac{1}{K} \sum_{i=1}^{K} \nabla f_i(x_t^i) \right\|^2
$$

$$
\leq \frac{3L^2}{K} \sum_{m=1}^{M} \|x_t^m - x_t\|^2 + \frac{3}{K} \sum_{m=1}^{M} \|\nabla f_m(x_t) - \nabla f(x_t)\|^2 + \frac{3L^2}{K} \sum_{i=1}^{K} \|x_t - x_t^i\|^2
$$

$$
= \frac{6L^2}{K} \sum_{m=1}^{M} \|x_t^m - x_t\|^2 + \frac{3}{K} \sum_{m=1}^{M} \|\nabla f_m(x_t) - \nabla f(x_t)\|^2
$$

$$
= \frac{6L^2}{K} \sum_{m=1}^{M} \|x_t^m - x_t\|^2 + 3G^2 + 3(B^2 - 1)\|\nabla f(x_t)\|^2.
$$

□

# E    CONVERGENCE ANALYSIS OF DES-LOC-ADAM (HIGH-PROBABILITY BOUNDS)

For this section, we refer to Algorithm 1 as DES-LOC-OPT($K_x, K_1, \ldots, K_N$). Let us consider the second algorithm DES-LOC-OPT($K, K, \ldots, K$) with $K = \text{lcm}\{K_x, K_1, \ldots, K_N\}$. These two algorithms have a property that they both fully synchronize, i.e., all states and current iterates are the same, if $T = rK$ for some $r \in \mathbb{N}$.

Commonly, the analysis of DES-LOC-OPT($K, K, \ldots, K$) proceeds in the following way. In each step, construct an ideal update as if you were running DES-LOC-OPT($1, 1, \ldots, 1$) using virtual iterates (see the proof in the prior section for the example of analysis with virtual iterates), and bound the drift from this idealized scenario. For the case of DES-LOC-OPT($K, K, \ldots, K$), the bound typically depends on the distance of the current iterate from the last full synchronization. Below, we show that the drift of OPT($K_x, K_1, \ldots, K_N$) is not larger than DES-LOC-OPT($K, K, \ldots, K$), since OPT($K_x, K_1, \ldots, K_N$) synchronize more often. Therefore, the convergence rate of OPT($K_x, K_1, \ldots, K_N$) is not worse than the convergence rate for DES-LOC-OPT($K, K, \ldots, K$) as its analysis also applies to OPT($K_x, K_1, \ldots, K_N$), i.e., all final upper bounds derived for DES-LOC-OPT($K, K, \ldots, K$) are also valid for OPT($K_x, K_1, \ldots, K_N$). For instance, a typical way to estimate drift is to have an assumption of type $\|s_i^n - s_{i-1}^n\| \leq U$ for all $i \in \{1, 2, \ldots, k\}$, and $n \in \{1, 2, \ldots, M\}$, where $s_i^n$ is some state on client $n$ at step $i$ and $s_0 = s_0^1 = \ldots = s_0^M$ the synchronized state. Then, drift is usually expressed as $\|s_k^n - s_0\|$. For DES-LOC-OPT($K, K, \ldots, K$), we can simply bound

$$
\|s_k^n - s_0\| = \left\| \sum_{i=1}^{k} s_i^n - s_{i-1}^n \right\| \leq \sum_{i=1}^{k} \|s_i^n - s_{i-1}^n\| \leq kU.
$$

For DES-LOC-OPT$(K_x, K_1, \ldots, K_N)$, we can obtain the same bound, where we for simplicity assume that $s$ is synchronized every $K_s$ steps and $k \in \{K_s + 1, \ldots, 2K_s\}$.

$$\|s_k^n - s_0\| = \left\| \sum_{i=K_s+1}^{k} (s_i^n - s_{i-1}^n) + s_{K_s} - s_0 \right\|$$

$$\leq \sum_{i=K_s+1}^{k} \|s_i^n - s_{i-1}^n\| + \left\| \frac{1}{M} \sum_{m=1}^{M} \sum_{i=1}^{K_s} s_i^m - s_{i-1}^m \right\|$$

$$\leq \sum_{i=K_s+1}^{k} \|s_i^n - s_{i-1}^n\| + \frac{1}{M} \sum_{m=1}^{M} \sum_{i=1}^{K_s} \|s_i^m - s_{i-1}^m\|$$

$$\leq kU.$$

In a more general case, we would apply the above recursively. Such type of adjustments is the only requirement to adapt analysis of DES-LOC-OPT$(K, K, \ldots, K)$ to obtain the same rate for DES-LOC-OPT$(K_x, K_1, \ldots, K_N)$ for the type of the analysis described above.

We do not claim any novelty for this analysis. We mainly include these results for completeness, to showcase that our method converges under different settings. The main theoretical results showing that some of the optimizer states can be synchronized less frequently are presented in the prior section above. We would also like to highlight that this result might be relatively weak and not tight since we only show that DES-LOC-OPT$(K, K, \ldots, K)$ and DES-LOC-OPT$(K_x, K_1, \ldots, K_N)$ have the same worst-case convergence, but DES-LOC-OPT$(K, K, \ldots, K)$ requires less communication than DES-LOC-OPT$(K_x, K_1, \ldots, K_N)$ under this analysis, which is not the case in practice nor in the analyses presented above.

Finally, detailed inspection of the analysis of DES-LOC-Adam $(K, K, \ldots, K)$ Cheng & Glasgow (2025) reveals that this analysis satisfies the above criteria. Thus, we can directly apply their results under the following assumptions and preliminaries.

We aim to optimize a neural network $x$ under the loss function $f$

$$\min_{x \in \mathbb{R}^d} f(x) := \mathbb{E}_{\xi \sim \mathcal{D}}[F(x; \xi)]. \tag{8}$$

using $M$ workers, each of which has access to the stochastic gradient of $f$, $\nabla F(x; \xi)$ with $\xi$ independently drawn from the data distribution $D$. We define the auxiliary sequence,

$$z_{t+1}^m = \begin{cases} \frac{1}{1-\beta_1} x_{t+1}^m - \frac{\beta_1}{1-\beta_1} x_t^m & \text{if } t \bmod K \neq -1, \\ \frac{1}{1-\beta_1} x_{t+1}^m - \frac{\beta_1}{1-\beta_1} \overline{x}_t & \text{otherwise.} \end{cases} \tag{9}$$

where, $\overline{x}_{t+1} = \mathbb{E}_m[x_{t+1}^m]$. We also define $\overline{z}_{t+1} = \mathbb{E}_m[z_{t+1}^m]$.

We make the following standard assumptions.

**Assumption 5** (Lower-boundedness). *$f$ is closed, twice continuously differentiable and* $\inf_{x \in \mathbb{R}^d} f(x) =: f(x_*) =: f_* > -\infty$.

**Assumption 6** (Smoothness). *There exists some set $\Omega \subset \mathbb{R}^d$ and $L > 0$, such that for any $x, y \in \Omega$,*

$$\|\nabla f(x) - \nabla f(y)\| \leq L\|x - y\|, \tag{10}$$

$$\|\nabla f(x)\|^2 \leq 2L(f(x) - f_*). \tag{11}$$

**Assumption 7** (Bounded $\alpha$-moment noise). *There exists some set $\Omega \subset \mathbb{R}^d$, $\alpha \geq 4$ and constant vector $\boldsymbol{\sigma} \succeq 0$ such that for any $x \in \Omega$,*

$$\mathbb{E}_{\xi \sim \mathcal{D}} |\nabla F(x; \xi) - \nabla f(x)|^\alpha \preceq \boldsymbol{\sigma}^\alpha. \tag{12}$$

*Let $\sigma_\infty := \|\boldsymbol{\sigma}\|_\infty = \max_i\{\sigma_i\}$, $\sigma := \|\boldsymbol{\sigma}\| = (\sigma_1^2 + \cdots + \sigma_d^2)^{1/2}$.*

**Assumption 8** (Weak convexity). *There exists constant $\tau > 0$ such that $f$ is $\tau$-weakly convex, i.e., for any $x, y \in \mathbb{R}^d$,*

$$\langle \nabla f(x) - \nabla f(y), x - y \rangle \geq -\tau\|x - y\|^2, \tag{13}$$

$$f(y) \geq f(x) + \langle \nabla f(x), y - x \rangle - \frac{\tau}{2}\|x - y\|^2, \quad \nabla^2 f(x) \succeq -\tau I_d. \tag{14}$$

Based on these assumptions, the `DES-LOC-Adam` variant of Adam converges as stated in the following theorem.

**Theorem 5.** *Let the Assumptions 5,6 ,7, 8, hold for* $\Omega = \text{conv}(\mathbf{B}_{R_0}(\Omega_0))$, *where* $\Omega_0 := \{x : f(x) - f_* \leq 4\Delta\}$, $\mathbf{B}_{R_0}(\Omega_0) = \{x \in R^d : \exists y : \|x - y\|_2 \leq R_0\}$, $R_0 = \sqrt{\frac{\Delta}{80L}}$, $K_{\text{lcm}} = \text{lcm}\{K_x, K_u, K_v\}$, *and the same assumptions as in Theorem D.3 of (*Cheng & Glasgow, 2025*), then with probability* $\geq 1 - \delta$, `DES-LOC-Adam` *yields,*

$$\frac{\lambda}{K_{\text{lcm}}R} \sum_{r=0}^{R-1} \sum_{k=0}^{K_{\text{lcm}}-1} \|\nabla f(\bar{z}_{r,k})\|^2 = \tilde{\mathcal{O}}\left(\frac{\tau\Delta}{R} + \frac{L\Delta}{K_{\text{lcm}}R} + \sqrt{\frac{L\Delta\sigma^2}{MK_{\text{lcm}}R}} + \frac{(L\Delta\sigma)^{\frac{2}{3}}}{K_{\text{lcm}}^{\frac{1}{3}}R^{\frac{2}{3}}} + \left(\frac{L\Delta\sigma^{\frac{a}{a-1}}}{K_{\text{lcm}}R}\right)^{\frac{2(a-1)}{3a-2}}\right)$$

*Proof.* The above corresponds to Theorem D.3 of (Cheng & Glasgow, 2025) for `DES-LOC-Adam` $(K_{\text{lcm}}, \ldots, K_{\text{lcm}})$. $\square$

Note that for sufficiently large $R$, the leading term in the rate is:

$$\frac{1}{K_{\text{lcm}}R} \sum_{r=0}^{R-1} \sum_{k=0}^{K_{\text{lcm}}-1} \|\nabla f(\bar{z}_{r,k})\|^2 = \tilde{\mathcal{O}}\left(\sqrt{\frac{L\Delta\sigma^2}{MK_{\text{lcm}}R}}\right), \tag{15}$$

In both cases, Theorem 5 shows that for the convergence bounds to hold for the high probability analysis of `DES-LOC-Adam`, synchronization needs to be a finite lcm

## F    DERIVATION OF EQS. (1) AND (2): MAXIMUM MOMENTUM CHANGE WITH CLIPPING

**Lemma.**    Let the gradient at each step satisfy $\|g_t\|_\infty \leq \rho$ for some constant $\rho > 0$. Assume the first-momentum state in Adam is initialized at $u_{-1} = 0$ and updated by

$$u_t = \beta_1 u_{t-1} + (1 - \beta_1)g_t, \quad \beta_1 \in [0, 1). \tag{16}$$

Then, for all $t \geq 0$, the momentum is bounded and satisfies

$$\|u_t\|_\infty \leq \rho, \quad \text{and} \quad \|u_{t+K} - u_t\|_\infty \leq 2\rho\left(1 - \beta_1^K\right) \quad \forall K \geq 1. \tag{17}$$

**Proof.**

**STEP 1: BOUND ON** $\|u_t\|_\infty$.    We first show by induction that the momentum is always bounded by $\rho$.

**Base Case** ($t = 0$): Since $u_{-1} = 0$, we have:

$$\|u_0\|_\infty = \|\beta_1 u_{-1} + (1 - \beta_1)g_0\|_\infty \leq (1 - \beta_1)\|g_0\|_\infty \leq \rho. \tag{18}$$

**Inductive Hypothesis (I.H.):** Assume $\|u_t\|_\infty \leq \rho$ for some $t \geq 0$.

**Inductive Step** ($t \to t + 1$): Then,

$$\|u_{t+1}\|_\infty = \|\beta_1 u_t + (1 - \beta_1)g_{t+1}\|_\infty \tag{19}$$
$$\leq \beta_1\|u_t\|_\infty + (1 - \beta_1)\|g_{t+1}\|_\infty \tag{20}$$
$$\leq \beta_1\rho + (1 - \beta_1)\rho = \rho. \tag{21}$$

Thus, by induction, we have the desired result:

$$\|u_t\|_\infty \leq \rho, \quad \forall t \geq 0. \tag{22}$$

**STEP 2: BOUND ON $\|u_{t+K} - u_t\|_\infty$.** Now we bound the change in the momentum over $K$ steps explicitly. Unrolling the recursion, we have:

$$u_{t+K} = \beta_1^K u_t + (1 - \beta_1) \sum_{k=0}^{K-1} \beta_1^k g_{t+K-k}. \tag{23}$$

Subtracting $u_t$ from both sides, we obtain:

$$u_{t+K} - u_t = (\beta_1^K - 1)u_t + (1 - \beta_1) \sum_{k=0}^{K-1} \beta_1^k g_{t+K-k}. \tag{24}$$

Applying the triangle inequality gives:

$$\|u_{t+K} - u_t\|_\infty \le |1 - \beta_1^K| \|u_t\|_\infty + (1 - \beta_1) \sum_{k=0}^{K-1} \beta_1^k \|g_{t+K-k}\|_\infty. \tag{25}$$

Using the bounds $\|u_t\|_\infty \le \rho$ and $\|g_t\|_\infty \le \rho$, we simplify to:

$$\|u_{t+K} - u_t\|_\infty \le (1 - \beta_1^K)\rho + (1 - \beta_1)\rho \sum_{k=0}^{K-1} \beta_1^k. \tag{26}$$

The geometric series simplifies as:

$$\sum_{k=0}^{K-1} \beta_1^k = \frac{1 - \beta_1^K}{1 - \beta_1}. \tag{27}$$

Substituting this back into the expression yields:

$$\|u_{t+K} - u_t\|_\infty \le (1 - \beta_1^K)\rho + (1 - \beta_1^K)\rho = 2\rho(1 - \beta_1^K). \tag{28}$$

Thus, the momentum difference satisfies:

$$\|u_{t+K} - u_t\|_\infty \le 2\rho(1 - \beta_1^K), \quad \forall K \ge 1. \tag{29}$$

**SECOND-MOMENT BOUND.** Applying the exact same logic to the second momentum $v_t$, with $\beta_1$ replaced by $\beta_2$ and the bounded gradient squared term $\|g_t \odot g_t\|_\infty \le \rho^2$, immediately gives:

$$\|v_{t+K} - v_t\|_\infty \le 2\rho^2(1 - \beta_2^K). \tag{30}$$

This completes the proof. $\qquad\square$

## G   WALL-CLOCK TIME MODELING

Understanding the practical benefits of our proposal beyond the theoretical aspects and empirical convergence curves is crucial. This section addresses the practical implications of adopting our method for training state-of-the-art (SOTA) large language models (LLMs) in large-scale distributed training infrastructures. The most critical metrics are based on total wall-clock time, communication time, and resource utilization, i.e., how much of the wall-clock time is spent using the compute available instead of waiting for the communication to complete. We provide the following simplified model for estimating total wall-clock time (Section G.1), computation time (Section G.1.1), and communication time (Section G.1.2) that applies to any method based on distributed data parallelism (DDP). The notation used here is consistent with that in Algorithm 1. We conclude this section with the results obtained with this modeling and their discussion.

### G.1 ESTIMATING TOTAL WALL-CLOCK TIME

The total wall-clock time for completing an LLM pre-training is based on the number of tokens processed $D$ (dataset size), the model size $d$ (the number of trainable parameters), the number of compute units $M$ (data-parallel/local workers), the floating point operations per second $S$ that these compute units can perform, the Model FLOPS Utilization (MFU), the average peer-to-peer (P2P) bandwidth $B$ and the latency $l$ between compute units. We separate the total wall-clock time discussion into computational time (Section G.1.1) and communication time (Section G.1.2). In our modeling, the total wall-clock time is the sum of computational time and communication time:

$$t_{\text{total}} = t_{\text{compute}} + t_{\text{comms}} \tag{31}$$

We next derive $t_{\text{compute}}$ and $t_{\text{comms}}$ separately, and then instantiate $t_{\text{total}}$ for specific training methods.

### G.1.1 ESTIMATING COMPUTATION TIME

The total time spent computing $T_{\text{compute}}$ depends on the number of compute units $M$, their floating point operations per second $S$, the MFU of the training pipeline, and the total number of FLOPs $C$ that the training pipeline requires. Following the same approach as in Kaplan et al. (2020); Hoffmann et al. (2022), the total number of FLOPs required to train an LLM can be estimated as $C = 6dD$, where $d$ is the number of model parameters and $D$ the total number of tokens (dataset size). Since the MFU can be considered a measure of efficiency, i.e., $\text{MFU} \in [0, 1]$, we can estimate the total time spent computing as:

$$t_{\text{compute}} = \frac{C}{\text{MFU} \cdot S \cdot M} = \frac{6 \cdot d \cdot D}{\text{MFU} \cdot S \cdot M} \tag{32}$$

In other words, if the hardware can perform $S \cdot M$ FLOPs/sec at peak and is utilized at MFU fraction of peak, the training FLOPs $C$ translate to that many seconds of compute.

In practice, MFU strongly depends on how the pipeline's parallelization is locally configured across the workers $M$. For the sake of fairness in our comparisons, we can assume that the per-batch MFU of a data-parallel worker is the same as the per-batch MFU of a worker in our proposal and other local adaptive methods. Importantly, this holds in cases where either such workers refer to a single GPU or each worker locally performs more advanced parallelism techniques, such as the ones proposed by Rajbhandari et al. (2020); Zhao et al. (2023).

**Resources Utilization and MFU.** Theoretically estimating the resource utilization in large-scale training of LLMs is very challenging despite prior knowledge of the number of hardware accelerators (GPUs), their theoretical peak FLOPs, and the total amount of FLOPs $C$ required to perform the task is available. Following previous well-established proposals (Chowdhery et al., 2023), we leverage MFU and the theoretical peak FLOPs of the hardware accelerators we used in our experiments. Recent systems research (Shoeybi et al., 2019) has shown it is possible to reach 50% of peak FLOPs even for trillion-parameter models by carefully combining data, tensor, and pipeline parallelism. This emphasizes that our model's assumptions (e.g., each worker sees full $d$) can be adapted to those scenarios by treating a model-parallel group as one worker with higher $S$ and similar MFU. For the sake of a fair comparison, our analysis in this section compares different methods assuming that the local workers operate with the same theoretical peak FLOPs and the same MFU. The results reported in Section G.2 describe how such values were obtained.

### G.1.2 ESTIMATING COMMUNICATION TIME

Communication time is the most critical factor when comparing standard data-parallel approaches to our proposal, since the computation time will be the same, given that they train the same model size on the same number of tokens using the same computing infrastructure. At each communication step, the workers $W$ synchronize a set of parameters $M$, the amount of which depends on the method used. For example, distributed data-parallel synchronization occurs at every batch step on the complete set of gradients produced by the $M$ workers, each exchanging a payload at batch step $i$ of $P_{\text{DDP},i} = d$ parameters. In our proposal, the synchronization involves model parameters and optimizer states at different frequencies, making such estimation slightly more complex. Since their time costs simply

add up, we treat the parameter sync and momentum sync contributions independently. For instance, if parameters are synced every $K_x$ steps and momenta every $K_u, K_v$ steps, we sum the time for each series of syncs.

Any of such payloads can be exchanged and averaged using bandwidth-efficient AllReduce methods, such as RingAllReduce (Sergeev & Balso, 2018), which scales only with the speed of the slowest P2P link. Given the slowest P2P bandwidth $B$ and a latency $l$, a single communication at timestamp $i$ is performed synchronously and in parallel across the $M$ workers, taking a total time of:

$$t_{\text{comms},i} = \frac{2P_i}{B}\left(1 - \frac{1}{M}\right) + l, \tag{33}$$

where $P_i$ is the payload size of the communication happening at the timestamp $i$, which depends on the optimization method adopted as described above.

**DDP.** In the `DDP` training approach, each of the $T$ optimization steps to train on $D$ tokens requires communicating at every step for a total training time of:

$$t_{\text{total,DDP}} = t_{\text{compute}} + T \cdot \left[\frac{2d}{B}\left(1 - \frac{1}{M}\right) + l\right] \tag{34}$$

**FedAvg.** The approach of the `FedAvg` method is that of synchronizing with frequency $K$ only the model parameters across the $M$ workers. This, the total training time can be estimated as:

$$t_{\text{total,FedAvg}} = t_{\text{compute}} + \frac{T}{K} \cdot \left[\frac{2d}{B}\left(1 - \frac{1}{M}\right) + l\right] \tag{35}$$

This optimization procedure will communicate less than `DDP` when $K < T$.

**Local Adam.** Using a local adaptive optimizer such as Cheng & Glasgow (2025) with a synchronization frequency of $K$ local steps, requires training for a total training time of:

$$t_{\text{total,Local Adam}} = t_{\text{compute}} + \frac{3T}{K} \cdot \left[\frac{2d}{B}\left(1 - \frac{1}{M}\right) + l\right] \tag{36}$$

This means that, as long as $3K < T$, `Local Adam` will always take less wall clock time than `DDP`.

**Our Method (DES-LOC).** Adopting our proposal (`DES-LOC-Adam` and `DES-LOC-ADOPT` specifically, which we shall use interchangeably for the purposes of this analysis) requires synchronizing model parameters $x$, fist momentum $u$ and second momentum $v$ with frequencies $k_x, K_u, K_v$, respectively. Assuming each of these sets is synchronized independently, we can compose by adding their communication time contribution to the total training wall-clock time, which results:

$$t_{\text{total,DES-LOC-Adam}} = t_{\text{compute}} + \left(\frac{T}{K_x} + \frac{T}{K_u} + \frac{T}{K_v}\right) \cdot \left[\frac{2d}{B}\left(1 - \frac{1}{M}\right) + l\right] \tag{37}$$

This means that, as long as $\frac{1}{K_x} + \frac{1}{K_u} + \frac{1}{K_v} < \frac{3}{K} \wedge \frac{1}{K_x} + \frac{1}{K_u} + \frac{1}{K_v} < 1$, our method will always take less wall-clock time than `Local Adam` and `DDP`.

**Limitations.** We critically discuss here the limitations of the proposed modeling in order to shed light on their relevance when it comes to deploying such training algorithms in real-world scenarios.

First, our modeling approach adopts constants for several system components, such as computing capabilities and interconnects. In particular, MFU in the real world always oscillates around some average value depending on the operational performance of high-bandwidth memories (HBMs), DRAM caches, and processing units in the hardware accelerators. At the same time, the P2P bandwidth and latency between accelerators also fluctuate around average values.

Second, most efficient implementations adopted in the field take advantage of the possibility of overlapping communication and computation, reducing the communication time. Notably, overlapping communication with computation can drastically reduce effective communication costs, for

example, PyTorch's DDP implementation can overlap $95\%$ of the communication (Romero et al., 2022). Our model currently assumes synchronous communications, but could incorporate such approaches by reducing the effective $l$ or $B$ impact. One extension could be adding a parameter $\alpha \in [0, 1]$ representing the fraction of communication time that is not overlapped, so total time per step $i$ is $t_{\text{total},i} = t_{\text{compute}} + \alpha t_{\text{comm}}$. Setting $\alpha = 0$ would recover the fully overlapped ideal (communication is entirely hidden by computation), and $\alpha = 1$ is the current no-overlap assumption. This would keep the model framework-agnostic but allow tuning to specific training setups.

Techniques in Rajbhandari et al. (2020); Zhao et al. (2023) complement our analysis by reducing memory usage and communication volume, effectively scaling down payload $P_i$ or increasing MFU. Our approach focuses on synchronization timing rather than data partitioning; combining our method with fragmented updates (e.g., ZeRO) could further improve wall-clock time.

Despite limitations, our model was designed so that any gap with real-world performance evenly affects all methods analyzed, assuming thoughtful implementation. Thus, results in Section G.2 illustrate potential improvements from adopting `DES-LOC`, and our model can help practitioners estimate performance at larger scales.

## G.2 MODELING RESULTS

Figures 23 and 24 analyze the wall-clock time, communication overhead, and GPU utilization of `DES-LOC` compared to `DDP`, `Local Adam`, and heuristic baselines for training our 1.7B model. By setting synchronization periods as $K_x = 256, K_u = 768, K_v = 1536$, `DES-LOC` significantly reduces communication and improves GPU utilization relative to `Local Adam` ($K = 256$), closely approaching the efficiency of heuristic methods, especially in bandwidth-constrained settings.

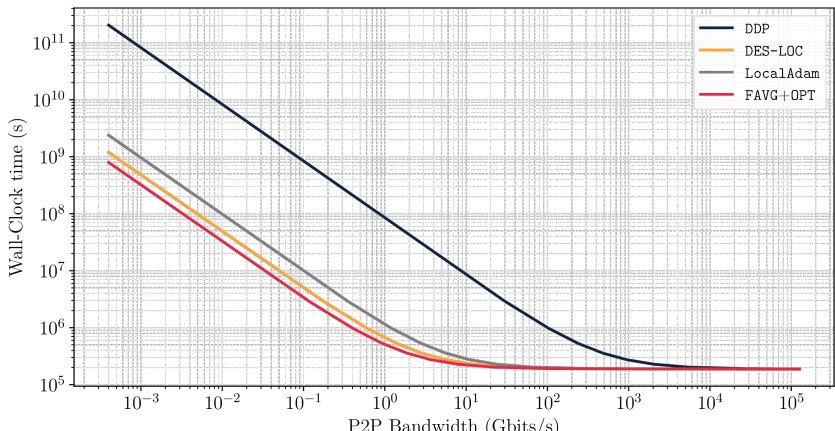

Figure 23: Estimated wall-clock time for training the 1.7B model with `DES-LOC` ($K_x = 256, K_u = 768, K_v = 1536$), compared to `Local Adam` ($K = 256$), `DDP`, and Federated Averaging with persistent optimizer states (`FAVG+OPT`, $K = 256$). At low bandwidth ($< 10^3$), all communication-efficient methods substantially reduce wall-clock time compared to `DDP`. `DES-LOC` closely approaches the maximum efficiency of `FAVG+OPT`, significantly outperforming `Local Adam`, which synchronizes all optimizer states frequently. Moreover, `DES-LOC` maintains stable and convergent training behavior (Fig. 5). At high bandwidth ($> 10^3$), `DDP` becomes competitive or preferable.

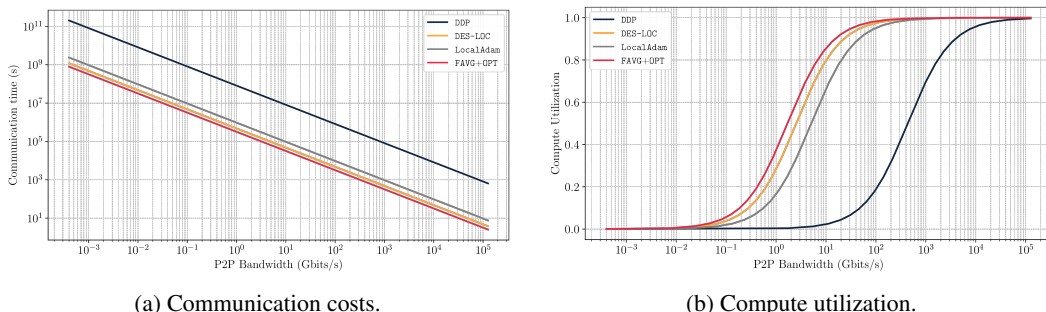

(a) Communication costs.                          (b) Compute utilization.

Figure 24: Communication overhead (a) and GPU utilization (b) for training the 1.7B model with synchronization periods $K_x = 256, K_u = 768, K_v = 1536$. `DES-LOC` reduces communication costs by $170\times$ compared to `DDP`, outperforming the $85\times$ reduction achieved by `Local Adam` while `FAVG+OPT`, communicating only parameters, achieves a theoretical maximum reduction ($256\times$). The improved communication efficiency of `DES-LOC` translates to higher GPU utilization at low bandwidths ($< 10^3$), significantly improving over `DDP` and `Local Adam`.

> **Takeaway:** By synchronizing optimizer states less frequently, `DES-LOC` enhances GPU utilization and total wall-clock time compared to `DDP` and `Local Adam`, especially under bandwidth constraints.

## H    CHECKPOINTING VS. PERIODIC STATE SYNCHRONIZATION

A natural question is whether simply checkpointing local optimizer states suffices for dealing with variable or elastic compute. This approach is inadequate for two reasons. *Quality:* Initializing new workers from a single stored state yields worse convergence as shown in Fig. 4.(c) ($\sim$15% higher perplexity in our tests in the follow-up round) compared to DES-LOC's averaging. *Elasticity:* When the worker count changes from $N$ to $M$, checkpointing lacks a principled mapping, forcing arbitrary state duplication or sub-selection, which either amplifies outliers or discards information.

A more principled ad-hoc strategy is averaging the $N$ existing states. To formalize the comparison, let the local states $\theta_i$ be i.i.d. variables with mean $\mu$ (the ideal global state) and variance $\sigma^2$ (local drift). The statistical risk is the Mean Squared Error, $\text{Risk}(\hat{\mu}) = \mathbb{E}[(\hat{\mu} - \mu)^2]$.

The **random selection estimator** (checkpointing), $\hat{\mu}_{rand} = \theta_k$, has a risk equal to the full sample variance:

$$\text{Risk}(\hat{\mu}_{rand}) = \sigma^2 \tag{38}$$

The **averaging estimator**, $\hat{\mu}_{avg} = \frac{1}{N} \sum_{i=1}^{N} \theta_i$, reduces this risk by a factor of $N$:

$$\text{Risk}(\hat{\mu}_{avg}) = \frac{\sigma^2}{N} \tag{39}$$

Averaging is thus more robust to the divergence of any single worker. However, even this principled ad-hoc approach underperforms DES-LOC. The crucial distinction is that `DES-LOC` builds periodic averaging into the training loop, treating it as a core mechanism rather than an external recovery tool. This proactively constrains the variance of local drift ($\sigma^2$) throughout training, ensuring all workers remain in a low-variance consensus state and making the system inherently robust to elasticity.

## I    CHOOSING SYNCHRONIZATION FREQUENCIES

Our results suggest a simple and principled rule-of-thumb for setting the synchronization periods ($K$) for model parameters and optimizer momentum states, grounded in the dynamics of exponential moving averages (EMAs). This methodology provides actionable defaults for practitioners seeking to balance model convergence with communication efficiency.

The core principle is that the synchronization frequency of any given optimizer state should be based on its empirical **half-life**—the time horizon over which its EMA "forgets" half of its past information.

This ensures that states are synchronized before they drift too far apart, maintaining training stability. For a state with a decay rate $\beta$ approaching 1, the half-life can be calcualted as:

$$t_{1/2} \approx \frac{\ln 2}{1 - \beta}$$

Based on this, we propose the following two-step methodology for setting the synchronization periods $K_x$ (for parameters), $K_u$ (for first moments), and $K_v$ (for second moments).

**Parameters First ($K_x$).** The synchronization of model parameters is paramount to training quality. The period $K_x$ should be chosen to match the end-of-training quality of fully-synchronous DDP at a target step budget while still materially reducing communication. In practice, starting points like $K_x = 16$ **or** $K_x = 32$ are effective as shown in our own work and in Charles et al. (2025). Parameters should always be synchronized at least as frequently as any momentum state.

**Momentum by Half-Life ($K_u, K_v$).** For any optimizer momentum state with a decay rate $\beta$, its synchronization period $K$ should be set near its calculated half-life, i.e., $K \approx t_{1/2}$. For common optimizers like Adam or ADOPT with well-tuned decay rates $\beta_1$ and $\beta_2$, this simplifies to setting the sync periods for the first and second moments as: $K_u \approx \frac{\ln 2}{1-\beta_1}$ and $K_v \approx \frac{\ln 2}{1-\beta_2}$.

Following this heuristic can yield a minimum $5\times$ reduction in communication cost over DDP for $K_x \geq 16$, significantly decreasing wall-clock time while achieving convergence speed and final model quality comparable to DDP.

## J    CRITICAL BATCH SIZE AND REGIME POSITIONING

To formally contextualize the regimes where DES-LOC is most beneficial, we begin with the statistical properties of gradient estimation. Let the true gradient over the full data distribution for a loss function $\mathcal{L} : \mathbb{R}^d \to \mathbb{R}$ be $G(\theta) = \nabla \mathcal{L}(\theta)$. In practice, a mini-batch of size $B$ provides an estimate, $G_{\text{est}}(\theta)$. The variance of this estimator scales inversely with the batch size:

$$\text{cov}(G_{\text{est}}(\theta)) = \frac{1}{B}\Sigma(\theta)$$

where $\Sigma(\theta)$ is the per-example gradient covariance. This relationship establishes a fundamental trade-off: smaller per-worker batch sizes $B$ result in higher-variance, or "noisier," gradient estimates.

Analyses of large-scale training have formalized the concept of a **critical batch size**, $B_{\text{crit}}$ (McCandlish et al., 2018; Zhang et al., 2025). This represents the point at which the benefits of increasing batch size begin to diminish.

When the batch size $B < B_{\text{crit}}$, the gradient estimate $G_{\text{est}}(\theta)$ is noisy, and increasing $B$ yields substantial improvements in convergence speed per step.

When $B \gg B_{\text{crit}}$, the gradient estimate $G_{\text{est}}(\theta)$ becomes a highly accurate estimate of the true gradient $G(\theta)$, and further increases to $B$ provide negligible returns.

In modern distributed settings with $N$ workers, the goal is often to operate at a **compute-optimal global batch size** ($G = N \times B$), which is typically near $B_{\text{crit}}$ for the given model and training duration. In massively parallel environments where $N$ is large, maintaining an optimal $G$ necessitates that the per-worker batch size $B = G/N$ becomes small. Consequently, large-scale, compute-optimal training often forces individual workers into a regime where $B \ll B_{\text{crit}}$, thereby exposing them to high levels of gradient noise.

For local-update methods (e.g., Local SGD, FedAvg with local optimizers), this high-variance regime is particularly challenging. Each worker performs multiple optimization steps using its own noisy gradient estimates, causing its local parameter replica $\theta_i$ to diverge from the other workers. This inter-worker drift can destabilize training and severely degrade final model quality.

DES-LOC is designed to counteract this divergence precisely in the high-noise, compute-optimal regime. By periodically synchronizing not only the model parameters but also the optimizer states

(e.g., Adam's momentum and variance accumulators), `DES-LOC` acts as a powerful consensus-enforcing mechanism. This periodic averaging reduces the variance of the distributed state, effectively dampening the destabilizing effects of high-variance local gradients and materially improving stability and final model quality. This allows the system to retain the communication savings of local updates without succumbing to parametric drift.

While the benefit of desynchronized momentum syncing may shrink in very-large-batch regimes where $B > B_{\text{crit}}$ (as local optimization is inherently more stable), `DES-LOC` remains highly attractive due to a combination of other robust properties:

- **Provable Convergence:** It maintains strong theoretical convergence guarantees under local updates.

- **Graceful Quality-Communication Trade-off:** The synchronization frequencies $(K_x, K_u, K_v)$ provide an explicit and effective mechanism to navigate the trade-off between communication cost and model performance.

- **Inherent Elasticity:** The method is fundamentally robust to dynamic changes in the number of workers. The periodic state averaging provides a principled, low-variance mechanism for initializing new workers, a scenario where naive checkpointing and state redistribution underperform significantly.

## K  EXTENDED RELATED WORK

**Federated Optimization.**   The `DES-LOC` framework, as mentioned in Section C.1, belongs to the broader field of federated optimization. A foundational algorithm in this field is `FedAvg` (McMahan et al., 2017), which established that a central model can be trained from decentralized data by averaging the model weights from clients that have performed local training steps. These findings were later generalized by Reddi et al. (2021) through the `FedOpt` framework, which reframes the training loop as a bi-level optimization, allowing the server to employ an optimization strategy more complex than simple averaging. Consequently, Reddi et al. (2021) demonstrated the instantiation of algorithms like `FedAdam`, `FedYogi`, and `FedAdagrad`, which achieve strong empirical performance and provide nonconvex guarantees (Kingma & Ba, 2015) by substituting the server's averaging step with a corresponding optimizer. In a related approach, Hsu et al. (2019) incorporate server-side momentum to improve the stability of aggregation, particularly when data is skewed. A primary challenge in federated learning involves heterogeneous data distributions, where clients hold non-IID data partitions. To address the problems arising from this heterogeneity, algorithms such as `FedProx`, which applies a proximal regularizer for stability (Li et al., 2020b), and `SCAFFOLD`, which uses control variates for robust convergence (Karimireddy et al., 2020b), have been developed. Likewise, `FedNova` addresses objective function inconsistencies by normalizing local steps (Wang et al., 2020). The Mime algorithm aims to reduce the gap between federated and centralized convergence through the use of control variates and server statistics (Karimireddy et al., 2020a). Lastly, methods such as Per-FedAvg (Fallah et al., 2020) and Ditto (Li et al., 2021) concentrate on personalization to enhance fairness and utility with reduced communication. Previous works have also investigated alternative inner optimizers for communication-efficient methods (Thérien et al., 2025).

**Compression of payload.**   The `DES-LOC` framework lessens the communication overhead in parallel training by reducing the communication frequency of parameter and momentum states compared to standard data parallel approaches. It is important to note, however, that the communicated payloads—the states themselves—can also be compressed, which would further enhance distributed training efficiency. Specifically, quantization methods can represent (pseudo)gradients in lower precision without a loss of model performance (Douillard et al., 2025; Kale et al., 2025). As an alternative, structured compression can express an update in a lower-rank form, either through SVD-like algorithms (Robert et al., 2025) or by only communicating the fast-moving momentum components (Peng et al., 2024). Sparsification techniques can introduce sparse update structures, which allows for better compression via information redundancy (Lin et al., 2018; Alistarh et al., 2018). Because update periodicity and update compression are orthogonal operations, they are frequently applied together to create highly efficient compression schemes without performance degradation (Douillard et al., 2025; Kale et al., 2025; Wang et al., 2023). Therefore, we anticipate

that this would be a fully composable enhancement to the `DES-LOC` framework, which we leave as a direction for future work.

## L  LLM USAGE DECLARATION

As noted in our submission, large language models (LLMs) were used throughout to assist with various aspects of this work. Specifically, we used `GPT-5` and `Gemini 2.5 Pro` to:

- Improve the clarity and flow of our writing.
- Find relevant related work that would be useful for our extended literature review.
- Assist with plotting code and simple code generations.

Beyond the stated uses above, all work, including but not exclusive to the interpretation of related work and results, is our own.

## M  LIMITATIONS

**Limitations.** First, while our main non-convex convergence result holds for `SGDM`, for `Adam` our analysis uses additional assumptions like bounded gradients and homogeneous data distribution. These assumptions are common in non-convex adaptive optimization. Second, our hyperparameter search was extensive yet constrained to smaller models. Lastly, while our analysis uses `Adam`/`AMSGrad`, many experiments use modified `Adam` (`ADOPT`) (Taniguchi et al., 2024).

## REPRODUCIBILITY STATEMENT

We are committed to the reproducibility of our work and provide the code, data-processing scripts, and configurations necessary to replicate the results in this paper.

**Code and Environment.** Our complete source code is available in the supplementary material. All dependencies are open-source and can be installed using the provided scripts (`system_setup.sh`, `install_env.sh`), which automate the full environment setup.

**Datasets.**  The experiments use publicly available, open-source datasets.  We provide the script `convert_hf_dataset_to_mds_smollm_corpus.sh` to replicate our entire data pre-processing pipeline, from downloading raw corpora to converting them into the required format.

**Experimental Protocol.** Reproducing our large-scale experiments requires access to significant computational infrastructure (e.g., multi-GPU servers), as specified in our documentation. All experiments are controlled via a well-defined configuration system using YAML files. Key hyperparameters and algorithmic settings, such as the synchronization frequencies for our method (`fl.n_local_steps`, `fl.parameter_scheduler_kwargs`) and the data distribution across workers, are explicitly defined. We include example scripts that execute the main experiments reported in the pape when using the approriate hyperparameters reported in Sections A and 4, providing a clear path to reproduce our findings.

