# OpenReview forum: "DES-LOC: Desynced Low Communication Adaptive Optimizers for Foundation Models"
_ICLR.cc/2026/Conference — ICLR 2026 Poster_

### Official Review · Reviewer_JkEm · 2025-10-31

**Soundness:** 3
**Presentation:** 3
**Contribution:** 3
**Rating:** 8
**Confidence:** 4

**Summary:**

The paper proposes DES-LOC, a family of desynced low-communication adaptive optimizers that assign independent synchronization periods to model parameters and optimizer momenta (first/second). Theoretical analysis shows (i) convergence for SGDM in non-convex settings and for Adam in weakly convex settings, and (ii) that more frequent momentum sync allows larger stable step sizes, while high-probability bounds require at least infrequent second-moment sync. Empirically (GPT-style LMs up to 1.7B parameters), DES-LOC achieves 170× less communication vs DDP and 2× less vs Local Adam, with 1.3–2.1× wall-clock speedups (aided by a wall-clock model for larger scales) and shows robustness to worker failures.

**Strengths:**

1. Originality. Decoupling sync periods for parameters and momenta, grounded in a half-life/step-size argument, moves beyond Local Adam’s uniform sync and earlier heuristics that lacked guarantees or failed under failures.
2. Quality. Convergence under standard assumptions with a clear theorem for SGDM and high-probability Adam bounds. Analysis explains why more frequent momentum sync permits larger stable steps and why $\beta_2$ cannot be left unsynced indefinitely.
3. Quality (empirics). Solid experimental design with explicit RQs, ablations over $K_x, K_u, K_v$ and comparisons showing DES-LOC matches Local Adam’s perplexity with 2× fewer state communications and drastically less than DDP.
4. Clarity. Clear algorithm block and figures.
5. Significance. Communication reductions (170× vs DDP; 2× vs Local Adam) and modeled wall-clock speedups directly target a primary FM training bottleneck (bandwidth), with fault-tolerance a notable practical plus.

**Weaknesses:**

There is no evidence for a larger-scale model.

**Questions:**

Could you add a $\ge$7B measured run?

---

> ### Author Response · Authors · 2025-11-23
> **Reply**
>
> ### **Dear Reviewer $\color{violet}{\text{JkEm}}$,**
>
> We sincerely thank you for your thoughtful review and for highlighting the ”**originality”** of decoupling synchronization periods, the ”**quality”** of our theoretical and empirical analysis, the ”**clarity”** of the presentation, and the **practical “significance”** of our wall-clock speedups and fault tolerance.
>
> We agree that demonstrating evidence at a larger model scale would further strengthen the practical message of the paper.
>
> ---
>
> ## **W1 & Q1 - $7$B-Scale Experiments**
>
> We are inted to conduct a **7B parameter** measured run to compare DES-LOC against DDP. We have prepared the setup for this experiment using the same recipe as our billion-scale runs (Section 4.1), however, we need some guidance before we allocate computational resources.
>
> ---
>
> ### **Clarification on Constraints & Timeline:**
>
> A standard compute-optimal training run for a 7B model requires approximately 140B tokens following Chinchilla scaling laws [1]. Given the computational resources available to us during the rebuttal window, it is unlikely that we can complete the entire 140B token run before the discussion deadline.
>
> ---
>
> ### **Proposal & Questions:**
>
> To ensure this experiment maximally addresses your concerns, we propose providing intermediate results (loss curves and throughput) during the rebuttal, with the full run included in the final manuscript. Could you kindly clarify your preferences on the following before we start the run?
>
> 1. **Training Budget:** Would a comparison over a shorter horizon (e.g., 10B–40B tokens) be sufficient to demonstrate the convergence trajectory and wall-clock benefits for the purpose of this rebuttal?
> 2. **Bandwidth Conditions:** Do you have a specific bandwidth regime in mind (e.g., 100 Gbps) where you would find this comparison most valuable?
>
> ---
>
> ### **Experimental Plan:**
>
> Unless otherwise directed, we will proceed with the following protocol:
>
> - **Setup:** 7B Decoder-only GPT; AdamW inner optimizer; $K_x =32, K_u = 4K_x, K_v = 8 K_x$;
> - **Tracking Metrics:**
>     - **Convergence:** Both **Step-wise** and **Time-wise** loss curves.
>     - **Efficiency:** Throughput (tokens/sec) and wall-time.
>     - **Quality:** Validation Perplexity and Downstream Task performance (ICL suite from Table 1).
>
> ---
>
> ### **Expectations at Scale:**
>
> We emphasize that our wall-clock modeling and throughput extrapolations (Appendix G) are expected to hold robustly at the 7B scale, as the communication scales predictably with model size ($d$) and bandwidth ($B$). The primary empirical unknown is the precise perplexity trajectory; however, prior large-scale studies on communication-efficient methods—such as [2,3]—have consistently demonstrated that local-update methods improve in performance at larger scales relative to DDP. We expect DES-LOC, with its state synchronization mechanism, to match or exceed these precedents.
>
> ---
>
> ## **Closing**
>
> We appreciate your recognition of our work's strengths. If the proposed reporting strategy (intermediate results now, full run in camera-ready) addresses your concern, we would value your feedback on the precise token budget.
>
> — The Authors
>
> References:
>
> [1] Hoffmann et al., Training compute-optimal large language models
>
> [2] Charles et al., Communication-Efficient Language Model Training Scales Reliably and Robustly: Scaling Laws for DiLoCo
>
> [3] Sani et al., Photon: Federated llm pre-training

---

> ### Author Response · Authors · 2025-12-03
> **Update: 7B Model Throughput on 8x B200s**
>
> To address your request for evidence of practical scalability at the **7B parameter** scale, we have conducted a measured throughput analysis on state-of-the-art hardware (**8$\times$ NVIDIA B200 GPUs**) in **App. B.8.**
>
> ---
>
> ### **Results**
>
> We measured the instantaneous tokens-per-second throughput during training (visualized in Fig. 22):
>
> - **Peak Efficiency:** **DES-LOC** maintains peak "local-only" speed for the vast majority of steps, operating with zero synchronization overhead between the defined intervals.
> - **Comparison to DDP:** Distinct throughput dips occur *only* at our sparse synchronization boundaries ($32, 96, 192$). In contrast, standard **DDP** would incur a communication penalty at *every single training step*, permanently depressing the throughput curve.
>
> This experiment confirms that our reported wall-clock speedups extrapolate robustly to the 7B scale on high-performance hardware, enabling near-linear scaling by keeping workers in a high-throughput local regime for the majority of training.
>
> ---
>
> ### **Closing**
>
> Regarding the full 7B convergence run, we were prepared to execute this experiment and were awaiting your input on the token budget to ensure the comparison met your expectations. Regrettably, due to the disabled reviewer responses, we did not receive this guidance in time to determine a meaningful training trajectory that could be completed within the rebuttal window. We have therefore prioritized the throughput analysis to demonstrate practical scalability during the rebuttal window.
>
> Best regards,
>
> The Authors

---

### Official Review · Reviewer_cfsD · 2025-11-01

**Soundness:** 3
**Presentation:** 3
**Contribution:** 2
**Rating:** 8
**Confidence:** 3

**Summary:**

This paper introduces DES-LOC, an optimization algorithm designed for decentralized training of foundation models, with a particular focus on large language models. The main objective is to improve communication efficiency during large-scale training in order to reduce wall-clock time.
The proposed method builds upon existing approaches such as FedAvg, FedOpt, and more specifically LocalAdam, where workers perform local parameter updates before averaging the weights with a central server. This approach contrasts with classical Distributed Data Parallel (DDP) methods, where synchronization occurs at every training step.
DES-LOC modifies the LocalAdam algorithm by decoupling the synchronization frequencies of the parameters and the optimizer states. While LocalAdam synchronizes both every K steps, this can lead to unnecessary communication overhead. The authors argue, and empirically demonstrate, that the first and second moment estimates (i.e., the momentum terms) evolve more slowly than the model parameters themselves, allowing for less frequent synchronization of these states.
In practice, the authors recommend using default ratios of Ku = 3Kx and Kv = 6Kx, or alternatively determining these based on the half-lives of β₁ and β₂. The paper also provides a convergence guarantee under standard assumptions commonly accepted in the optimization community.
Extensive experiments on LLM training show that DES-LOC achieves performance comparable to LocalAdam while reducing communication costs by approximately half. Overall, the paper presents a well-motivated and empirically validated contribution to improving efficiency in decentralized large-scale model training.

**Strengths:**

- The paper is well motivated, and the overall writing is clear and easy to follow. The analysis based on the half-life times provides useful intuition for why less frequent parameter updates do not negatively affect model training.
- I particularly appreciated the qualitative analysis of the upper bound on model convergence (Theorem 1), which effectively shows that DES-LOC enables the use of a larger learning rate in practice.
- The authors’ claims are well supported by extensive experiments, including analyses of rate of change, convergence behavior in practical LLM training scenarios, and full model training comparisons (in wall-clock days) with DDP and other baselines, which are especially compelling.

**Weaknesses:**

- While the method is supported by both theoretical analysis and experimental results, its practical novelty is somewhat limited. The main contribution lies in decoupling parameter updates from optimizer updates, which constitutes a relatively minor modification of the existing LocalAdam algorithm.
- The authors did not conduct experiments under very low-bandwidth conditions. Demonstrating the method’s effectiveness in scenarios with limited network capacity (e.g., 1 Gb/s or 10 Gb/s) would further strengthen its practical relevance.
- The title of the paper refers to “foundation models,” but the large-scale experiments focus exclusively on LLM training. Although these results are already impressive, the authors could either adjust the paper’s scope to explicitly center on LLMs or include an additional experiment involving the training of a vision foundation model to better match the stated scope.

**Questions:**

- What would be the practical relevance of DES-LOC for training scenarios where workers are connected through significantly lower-bandwidth networks?
- Could the proposed desynchronization strategy also be applied to optimizers that use only a single momentum term (e.g., Muon)?

---

> ### Author Response · Authors · 2025-11-23
> **Reply**
>
> ### **Dear Reviewer $\textcolor{teal}{\textbf{cfsD}}$,**
>
> We thank you for your thoughtful review, and we are glad you have found the paper “**well motivated”** and **clearly written** and that you considered both our **theoretical bound (Theorem 1)** and **large-scale LLM** experiments to be “**compelling”**. We provide our response.
>
> ---
>
> ## **W1 – Practical and Theoretical Improvements**
>
> As detailed in our general comment above, **Shared Concern 1 (SC1** , DES-LOC significantly improves upon Local Adam [1], unlocking a new optimization design space.
>
> - **Practical and Empirical Contribution**: DES-LOC can reduce communication by **2$\times$** vs. Local Adam and **170$\times$** vs. DDP while matching performance. Beyond efficiency, our new results confirm that state synchronization provides advantages over purely local optimizer states methods:
>     - **Superior Performance with Nesterov:** Our new 1.3B experiments show DES-LOC+Nesterov explicitly **outperforms** DiLoCo, a method equivalent to DES-LOC+Nesterov with $K_u,K_v = \infty$, in final perplexity.
>     - **Robustness to Noise & Failure:** Unlike local optimizer states heuristics, which suffer from unstable internal statistics (e.g., exploding norms in Fig.18 and activations Fig.7) and suffer under worker failures, DES-LOC’s periodic averaging controls drift and provides a robust mechanism for initializing new workers during training (Fig.4.c).
>     - **Theoretical Contribution:** Unlike Local Adam, our novel theory (**Theorem 1**) explicitly links *each* momentum state's sync frequency to its decay rate $\beta$ and shows that frequent momentum sync enables **larger stable step sizes(Eq.4)**, neither of which were present in Local Adam.
> - **Algorithmic Contribution**: DES-LOC unifies previous local-update methods. By decoupling $\{K_x, K_u, K_v\}$, practitioners can jointly tune $\beta$ and sync frequencies—a trade-off inaccessible to coupled methods like Local Adam.
>
> ---
>
> ## **W2 & Q1 – Low bandwidth 10 Gb/s experiments**
>
> We have added benchmarks at **10 Gbps** in section B.5, Fig.19. We initially chose not to focus on extremely low-bandwidth regimes because we are specifically interested in cross-datacenter training scenarios.
>
> - **Quality remains unchanged:** Perplexity remains unchanged, as bandwidth affects training duration, not the optimization trajectory.
> - **Wall-time Benefit:** DES-LOC’s benefit becomes more pronounced in low-bandwidth regimes. Wall-clock time differences grow as bandwidth decreases and compute becomes a smaller share of the total training time. In our experiments DES-LOC reduces training time by $\approx 9.42 \times$, taking total time from $3.5$ days to $8.99$ hours
> - **Modeling**: Our idealized wall-clock model (Appendix G) captures this trade-off:
>
>     $T_{\text{total}} \approx T_{\text{comp}} + \frac{M_{\text{payload}}}{B_{\text{eff}}} \times N_{\text{sync}}$,  $T_{\text{comp}}$ is compute time, $M_{\text{payload}}$ is the payload size in bytes and $N_{\mathrm{sync}}$ is the number of sync steps. Because DES-LOC significantly reduces communcation $N_{\text{sync}}$ vs DDP,  e.g by  $21\times$ for $K_x=32,K_u=3K_x,K_v=6K_x$, the speedup factor approaches the communication reduction factor as bandwidth $B_{\text{eff}} \to 0$ and compute time becomes a smaller share of the total time.
>
>
> ---
>
> ## **W3 – “Scope mismatch between ‘foundation models’ and LLM-only experiments.”**
>
> While our theory is architecture- and task-agnostic, our experiments focused on text and left the vision domain for future work. However, if we cannot complete rigorous vision experiments during the rebuttal time, we propose aligning the narrative scope via:
>
> - **Title Change:** “DES-LOC: Desynchronized Low-Communication Optimizers”
> - **Clarification:** Explicitly stating in the abstract/intro that while the analysis is generic, empirical evaluation focuses on decoder-only LLMs.
>
> We hope this better matches the stated scope and addresses your comment.
>
> ---
>
> ## **Q2 – “Can desynchronization be applied to single-momentum optimizers (e.g., Muon)?”**
>
> In short, yes. DES-LOC can be applied to single momentum optimizers such as Muon [2]. We have theory showing the case of this working with SGDM, but the extension to Muon, although feasible and interesting, is left as a direction for future work given the mathematical technicality. Please see the general comment (**SC2**) for a more detailed answer.
>
> ---
>
> ## **Closing**
>
> We thank you again for your constructive feedback, which has significantly improved the framing and practical relevance of our work. If you feel that our responses, additional low-bandwidth experiments, and clarifications adequately address your concerns, we would be grateful if you would consider raising your score. We await further feedback and questions.
>
> — The Authors
>
> ## References
>
> [1] Cheng and Glasgow, Convergence of Adaptive Optimization with Local Updates
>
> [2] Jordan, Muon: An optimizer for hidden layers in neural networks

---

> ### Author Response · Authors · 2025-12-03
> **Update: Experiments on Muon and Vision Models (Flux)**
>
> We have now successfully completed additional experiments to address your concerns regarding the scope of "Foundation Models" and the applicability of our method to single-momentum optimizers like Muon. We have also added these new experiments in our reply to the general comment above.
>
> ---
>
> ### 1. Applicability to Muon
>
> You asked if our strategy applies to single-momentum optimizers like Muon. We have now implemented DES-LOC with Muon as the inner optimizer (App. B.6).
>
> - **Mechanism:** Distinct from Adam, Muon preconditions the momentum term using Newton-Schulz iterations. We apply **DES-LOC** by synchronizing the momentum buffer ($K_u$) less frequently than the parameters ($K_x$).
> - **Result:** We find that setting $K_x=32$ and $K_u=96$ ($3\times$ delay) maintains solution quality matching the **Local Muon** baseline across model scales (16M, 125M, 360M), see Fig. 20, while reducing total communication volume by **$>1.5\times$** .
>
> ---
>
> ### 2. Expanding Scope to Vision Models (Flux)
>
> You noted that our title refers to "Foundation Models" while experiments focused on LLMs. To justify this scope and demonstrate universality, we evaluated DES-LOC on Flux (280M parameters), a Rectified Flow Transformer for text-to-image generation (App B.7).
>
> - **Result:** As shown in **Fig. 21**, **DES-LOC** matches the convergence trajectory of **both** the fully synchronous **DDP** baseline and **Local Adam** on a 280M parameter model (within 0.02 perplexity points).
> - **Significance:** We achieve this performance parity while communicating $2\times$ less than Local Adam ($K_v=32, K_u=3 K_x, K_v = 6 K_x)$.
>
> While the size of this model is insufficient to qualify as a foundation model, we believe these new experiments expand the scope of our work and demonstrate the generality of our method. We maintain our commitment to change the paper title once this becomes possible.
>
> Best regards,
>
> The Authors

---

### Official Review · Reviewer_KN8y · 2025-11-04

**Soundness:** 2
**Presentation:** 3
**Contribution:** 2
**Rating:** 4
**Confidence:** 4

**Summary:**

The authors propose DES-LOC, a communication-efficient optimization algorithm that reduces the frequency of optimizer state communication compared to local Adam. The authors prove the convergence of DES-LOC in the non-convex case (for SGDM) and for Adam under weakly convex objectives. The author’s key insight over Local Adam is that the first and second moments can be synchronized at a rate on the order of their half-lives, much less frequently than their parameters need to be synchronized. In experiments pre-training language models, the authors show that significantly less than DDP and up to 2x less than Local Adam while achieving similar final performance.

**Strengths:**

- The paper is well-written and easy to understand.
- I think experimental settings are well structured.
- From the math in the main paper, the convergence guarantees look sound (I did not check the appendix).
- The idea of synchronizing optimizer states relative to their decay half-lives is intuitive to me, and I think it's an interesting direction of inquiry for the distributed optimization community.

**Weaknesses:**

- My main concern is the lack of comparison to existing methods such as DiLoCo [1] and MuLoCo [2]. Neither method requires synchronizing the optimizer states; therefore, they are trivially more communication-efficient than DES-LOC and have been shown to perform competitively to DDP in practice. I am aware of your experiments in Figure 6 showing a DES-LOC method with Nesterov momentum, but from my understanding, these still synchronize optimizer states.
- Following from my concern above, since the communication efficiency benefits of DES-LOC are already realized by proposed methods in the literature [1,2], I’m not sure how strong the contribution is. Could the authors comment on this?
- I wonder if synchronizing the optimizer states in DES-LOC can provide a benefit beyond [1,2]. I would be surprised if it does not, and I believe experiments showing this would greatly strengthen your results.


[1][DiLoCo: Distributed Low-Communication Training of Language Models]

[2][MuLoCo: Muon is a practical inner optimizer for DiLoCo]

**Questions:**

- Are the comparisons to the DDP baseline FLOP-matched?
- Figure 3 (a), (b), (d) have the same rectangular pattern in the loss curve. Why is this the case?

**Suggestions**:
- I have trouble scrolling through your .pdf because the figures take long to load. Perhaps subsampling the point used for plotting can help.
- I have trouble distinguishing different loss curves in your plots. Perhaps smoothing could help.

---

> ### Author Response · Authors · 2025-11-23
> **Reply**
>
> ### **Dear Reviewer $\textcolor{orange}{KN8y}$,**
>
> Thank you for your careful reading and for highlighting that our work is “**well-written**”, “**well-structured**”, “**intuitive**” and “**sound**” both mathematically and experimentally. Below we address each of the concerns that have been raised, in turn.
>
> ---
>
> ### **W1 + W2 + W3 — DiLoCo comparisons & the benefits of synchronizing optimizer states**
>
> ### **W1 + W2 - DiLoCo Comparisons**
>
> Our new experiments show that DES-LOC+Nesterov with $K_u=3K_x, K_v=6 K_x$ **outperforms** DiLoCo [2] in terms of **model performance**.  Since DiLoCo represents a special case of DES-LOC+Nesterov where $K_u,K_v \to \infty,\infty$, for a fixed $K_x$ DES-LOC **provides a trade-off**: one can improve the communication efficiency of DES-LOC to be arbitrarily close to DiLoCo at the cost of eventually also approaching its performance.  We also wish to emphasize that DES-LOC is the first method to **provide provable convergence while decoupling sync frequencies**. This flexibility is not possible under Local Adam [1] nor purely local optimizer states approaches such as DiLoCo.
>
> - **Practical Superiority:** Our new new **1.3B experiments (Section 5.5.2)** show that **DES-LOC + Nesterov** improves perplexity  by $0.9$% over DiLoCo while better controlling the output activations of the model~(Fig.7 bottom right). Furthermore, Fig. 4c demonstrates that purely local optimizer states methods fail to adapt to worker churn (node failure/addition), whereas DES-LOC's global state averaging provides a robust initialization mechanism.
> - **Beyond Heuristics:** While put methods like DiLoCo are empirically successful heuristics in certain cases [3], they lack convergence guarantees for adaptive optimizers [1]. Even recent rigorous treatments of outer optimizers [3,5] assume SGD locally for the theory, not Adam. DES-LOC with averaging as the outer optimizer restores guarantees for SGDM and Adam (under weak convexity) by ensuring finite momentum synchronization ($K_{u},K_{v} < \infty$).
> - **Pareto Frontier:** As detailed in **Shared Concern 3 (SC3)**, DiLoCo is a special case of DES-LOC where $K_u, K_v \to \infty$. By decoupling frequencies, DES-LOC opens a new design space that allows us to find configurations at least as good as DiLoCo.
>
> ### **W3 — "Can syncing optimizer states give benefits?"**
>
> Yes. Our new experiments confirm your intuition: synchronizing optimizer states on top of DiLoCo-style training improves stability and final perplexity, as discussed in our answer to W1+W2
>
> ---
>
> ### **Q1 — “FLOP matching to the DDP baseline”**
>
> Yes, all comparisons to the DDP baseline are strictly FLOP-matched and trained on identical token budgets.
>
> ---
>
> ### **Q2 — “Rectangular pattern in Fig. 3(a,b,d)”**
>
> We thank the reviewer for this question. Could you please specify what you mean by "rectangular pattern"? If you are referring to the fact that the loss curves in panels (a), (b), and (d) exhibit very similar trajectories, this is intentional: we utilize the same model size, same dataset sequence, same initialization, and same random seed across all ablations to ensure a strictly controlled, fair comparison.
>
> If you mean cyclical fluctuations in the loss, this is a naturally occurring pattern we have observed with communication-efficient training with ADOPT especially, the loss tends to decrease immediately upon synchronization since the new parameters are superior,  then it eventually slightly increases before going down again to a lower point after the following synchronization.
>
> If you meant something else we are happy to provide further information.
>
> ---
>
> ### **S1 - “Readability and PDF size”**
>
> We have now optimized the PDF figures for faster loading. We have also added smoothed variants showing mean+std across workers for the plots in the main text.
>
> ---
>
> ## **Closing**
>
> We thank you for your recommendations which have helped us substantially develop our work. If our new results and explanations adequately address your concerns, we kindly ask you to consider raising your score. We look forward to any further feedback or any inquiries you may have.
>
> — The Authors
>
> References:
>
> [1] Cheng & Glasgow, Convergence of distributed adaptive optimization with local updates
>
> [2] Douillard et al., Diloco: Distributed low-communication training of language models, 2023.
>
> [3] Charles et al, Communication-Efficient Language Model Training Scales Reliably and Robustly: Scaling Laws for DiLoCo
>
> [4] Sani et al, Photon: Federated LLM Pre-Training
>
> [5] Khaled et al., Understanding Outer Optimizers in Local SGD,

---

> > ### Comment · Reviewer_KN8y · 2025-11-27
> >
> > Thank you for your reply and for providing a new comparison to DiLoCo.
> >
> > Regarding the empirical contribution, from the new DiLoCo experiments, we observe that the proposed method performs on par (well within error bars in Figure 7) with DiLoCo. As such, it is unclear whether DES-LOC has an advantage over DiLoCo when accounting for its increased communication cost.
> >
> > With regards to the theoretical contribution, 26GM has pointed out that "local Adam already established that optimizer states needs to be synchronized and also connected the convergence bound to momentum coefficients". I had previously overlooked this point in my first review of the paper but I am now concerned about the significance of the author's theoretical contribution over Local Adam.
> >
> >
> > Given my aforementioned concerns, I have decided to maintain my score for now. That being said, I will take into account reviewer 26GM's reply to the authors once it has been posted.

---

> > > ### Author Response · Authors · 2025-11-27
> > > **Reply**
> > >
> > > ## Dear Reviewer ${\color{orange}{\text{KN8y}}}$,
> > >
> > > Thank you for your further engagement. We would like to clarify our contributions over DiLoCo and the theoretical differentiation from Local Adam.
> > >
> > > ---
> > >
> > > ### **Our performance improvement over DiLoCo falls within the expectation of prior works and we are also the first to quantify error**
> > >
> > > You noted that the performance gap is "well within error bars". We respectfully submit that a $0.9$% improvement in perplexity is widely considered significant in this specific domain [1,2,3], and that the overlap in error bars is a function of our more rigorous reporting standards accounting for **compounding worker drift** over time.
> > >
> > > - **Interpretation of Error Bars:** Most prior art (including [1], the original DiLoCo[2] and Streaming DiLoCo[3] papers) report their major perplexity comparisons against baselines **without error quantification.**
> > >     - For our work, the error bars are shown **across workers**. Thus, the fact that the bars overlap reflects the natural variance of the distributed method and **NOT that the method itself is ineffective**. In federated/local-update methods, worker models naturally **drift,** unlike DDP; therefore, the variance *between* workers is **high**. We chose to report this error to adequately capture the impact of **compounding worker drift** over time, an effect which **could not be observed from prior reporting.**
> > >     - For the summary metric displayed in the label, we propagate the worker error across time, computing the mean + std over the final round, using the standard Pooled Variance formula.
> > >     - We believe that presenting such error quantification makes our comparison more fair and accurate and **should not be held against our work.**
> > > - **Magnitude of Improvement:** In the context of communication-efficient LLM training, **gaps of $<1$%** are the standard margin of improvement. For instance, in Table 4 of the recently published "Scaling Laws For DiLoCo" [1], the performance gaps between DiLoCo and DDP at the 1.3B scale are all bounded between $0.4$% and $1.3$%, with an average absolute gap of **$0.6$%**. Our reported improvement of **$0.9$%** over the DiLoCo baseline is above that average and consistent with the expected gains at this model size.
> > >
> > > ### **Theoretical Contribution vs. Local Adam**
> > >
> > > Regarding the concern raised by ${\color{red}{\text{26GM}}}$: our theoretical contribution over Local Adam is distinct and prescriptive.
> > >
> > > - **Local Adam (Conditional):** Local Adam establishes a *conditional* link: it states that *if* one wishes to sync every $K$ steps, one *must* alter the optimizer hyperparameters (specifically, $1-\beta_2$ only must scale with $K^{-3/2}$) to ensure convergence. It provides no further information on how this impacts the training process itself.
> > > - **DES-LOC (Prescriptive & Decoupled):** Our theory (Theorem 1) is the first to derive bounds for *independent* synchronization periods $(K_x, K_u, K_v)$. **This is not a naive increment to Local Adam.** Our framework is *prescriptive*. We explicitly prove that states with longer half-lives ($\tau_{0.5}$) can be synced less frequently without breaking convergence and link the admissible step size to the individual $\beta$’s (EQ.4). This decoupling allows us to minimize the additional communication costs of synchronizing optimizer states, a capability absent in Local Adam, which opens up joint optimization of the $\beta$’s and synchronization periods.
> > >
> > > Please also see our response to ${\color{red}{\text{26GM}}}$ and **Shared Concern 1** in our reply to the general comment above.
> > >
> > > ### **Expanding the Pareto Frontier**
> > >
> > > Finally, we reiterate that **DES-LOC fully encapsulates DiLoCo** (which is the limit case $K_{u,v} \to \infty$). It allows practitioners to trade bandwidth (to sync momenta) for performance gains that purely local optimizer states methods cannot access, providing results that are guaranteed to be no worse than DiLoCo.
> > >
> > > ## Closing
> > >
> > > We hope with this clarifications the reviewer would agree that **a)** our empirical results are not just on par with DiLoCo but **improve it** by the **standards of prior works[1,2,3]**, and **b)** that our theoretical contributions are novel relative to Local Adam.
> > >
> > > We appreciate the reviewers engagement with our rebuttal and we welcome any further questions and comments.
> > >
> > > — The Authors
> > >
> > > ## References
> > >
> > > [1] Charles et al, Communication-Efficient Language Model Training Scales Reliably and Robustly: Scaling Laws for DiLoCo
> > >
> > > [1] Douillard et al., Diloco: Distributed low-communication training of language models, 2023.
> > >
> > > [3] Douillard et al., Streaming DiLoCo with overlapping communication.

---

### Official Review · Reviewer_26GM · 2025-11-04

**Soundness:** 3
**Presentation:** 4
**Contribution:** 2
**Rating:** 4
**Confidence:** 4

**Summary:**

The paper provides a theoretical argument to synchronize local optimizer states (first and second moments) for fedavg/diloco. Specifically, it shows local momentum needs to be synced; however, it can be infrequent wrt outer updates, and this frequency depends on the momentum coefficients. It improves over vanilla diloco and reduces communication wrt syncing states in every outerstep.

**Strengths:**

1. The theory-backed argument to have infrequent momentum synchronization is insightful and makes sense. The connection to beta (momentum coefficient) and the synchronization frequency is intuitive.
2. The results show the benefit of synchronizing momemtum terms with two optimizers and show no degradation wrt to local Adam.
3. Writing is very clear, and the toy example illustrated the need for momentum synchronization.

**Weaknesses:**

1. The main theorem is for SGD+momentum, however, the method is proposed for adaptive optimizers. This is a limitation as adaptive optimisers are popular and the frequency of synchronization is derived from the theory for SGDM and applied to adam/adopt.
2. The contribution over local Adam is marginal as local Adam already established that optimizer states needs to be synchronized and also connected the convergence bound to momentum coefficients. Please clarify the main contributions.
3. Even with momentum sync, the results are worse than DDP. This raises a question of whether diloco-type methods (weight averaging method) would match DDP (gradient averaging)?

**Questions:**

1. The method doesn't seem specific to adaptive methods. Would it be better to frame it more generally? Would it work with muon-type optimizers?
2. Is it correct to say local SGD (ie, fedavg/diloco) is sharing only model parameters? It actually shared the parameter differences (pseudo gradients) and uses them with an outer optimizer. Would it make sense to frame the method as parameter-sharing methods need optimizer states to be synced rather than relating them to diloco type methods?

---

> ### Author Response · Authors · 2025-11-23
> **Reply**
>
> ### **Dear Reviewer $\textcolor{red}{26GM}$,**
>
> We sincerely thank you for your thoughtful review and highlighting our “**very** **clear**” writing and the “**insightful**” and “**intuitive”** nature of our theory.
>
> ---
>
> ## **W1 - Theory for adaptive optimizers.**
>
> Whilst the theory in the main text is for the SGDM case, **we had provided the adaptive optimizer theory (Adam) in the Appendix (see Section D.1 and E) for** both **in-expectation** and **high-probability** bounds (forward link in Sec.3). We presented SGDM first as the simplest base-case of our algorithm for illustrative purposes.
>
> ---
>
> ## **W2 - Improvement over Local Adam**
>
> As we present in detail in our **SC1** reply to the general comment above, DES-LOC is a significant improvement over Local Adam [1], opening a new optimization design space.
>
> - **Theoretical Contribution:** Local Adam only **conditionally** links $\beta_2$ to a unified period $K$. In contrast, our theory (Theorem 1) explicitly links *each* momentum's sync frequency to its decay rate $\beta$, proving that frequent momentum sync enables **larger stable step sizes**.
> - **Practical and Empirical Contribution**: This reduces communication by **2$\times$** vs. Local Adam while matching performance (Fig. 5,6,7).
> - A**lgorithmic Contribution**: DES-LOC unifies previous local-update methods. By decoupling $\{K_x, K_u, K_v\}$, practitioners can jointly tune $\beta$ and sync frequencies to minimize communication—a trade-off inaccessible to coupled methods like Local Adam.
>
> ---
>
> ## **W3 - DiLoCo Comparison**
>
> Our new results demonstrate that **DES-LOC + Nesterov** can outperform **DiLoCo [2]** and bridge the gap to standard AdamW-DDP, offering a superior-or-equal Pareto frontier.
>
> We distinguish the **synchronization mechanism** from the **outer optimizer**.
>
> - **Mechanism:** Standard DDP maintains identical model *and* optimizer states across workers by synchronizing gradients at every step ($K=1$), while local update methods allow workers to diverge for $K$ steps before synchronizing.
> - **Outer Optimization:** An outer optimizer (e.g., Nesterov) can be applied to *both* regimes. For local updates, it operates on the computed pseudo-gradient after synchronization. For DDP, it is applied by maintaining a snapshot of the global model every $K$ steps and computing the difference from the current global model (equivalent to a single-replica local update).
> - **The Hierarchy:** Previous scaling laws [3] established that while **DiLoCo** (purely local optimizer states local updates + Nesterov) can match or outperform *standard* AdamW-DDP, however it cannot always match **AdamW-DDP + Nesterov** (the fully synchronous equivalent).
>     - **DES-LOC vs. DiLoCo:** DiLoCo is simply a special case of DES-LOC where optimizer states are never synchronized ($K_{u},K_{v} \to \infty$). Since the DiLoCo configuration is a strict subset of the DES-LOC design space, optimizing over DES-LOC guarantees the existence of a solution theoretically **no worse** than the purely local optimizer states baseline.
> - **Our new 1.3B experiments (Section 5.5.2) confirm DES-LOC’s theoretical advantage:** DES-LOC + Nesterov outperforms **DiLoCo** and standard AdamW-DDP in perplexity and downstream tasks, showing the benefit of state synchronization.
>
> ---
>
> ## **Q1 - DES-LOC with Muon**
>
> Correct. DES-LOC is optimizer-agnostic and applies to any method with an EMA state, including matrix-based optimizers like Muon [4] which orthogonalize the momentum. Please see the general comment (SC2) for a more detailed answer. We will enlarge the scope of the framing as recommended.
>
> ---
>
> ## **Q2 - Pseudo-gradient vs Parameter Sharing**
>
> We clarify that under FedOpt [5] (with server LR=1.0), synchronizing parameters or pseudo-gradients are **mathematically equivalent operations**. Furthermore, the pseudo-gradient can be computed locally by the client or reconstructed by the server after receiving parameters; the resulting global update is identical. As such, whether clients transmit parameters or differences, the payload dimensionality and update information remain the same. The fundamental characteristic that groups DES-LOC with DiLoCo/Local Adam is the $K$-step local update structure.
>
> ---
>
> ## **Closing**
>
> We hope these clarifications address your concerns. We are looking forward to receiving further feedback and clarifying any other points and kindly ask you to consider raising your scores if your concerns have been adequately addressed.
>
> — The Authors
>
> [1] Cheng & Glasgow, Convergence of distributed adaptive optimization with local updates
>
> [2] Douillard et al., Diloco: Distributed low-communication training of language models
>
> [3] Charles et al., Communication-Efficient Language Model Training Scales Reliably and Robustly: Scaling Laws for DiLoCo
>
> [4] Jordan, Muon: An optimizer for hidden layers in neural networks
>
> [5] Reddi et al, Adaptive Federated Optimization

---

> > ### Author Response · Authors · 2025-12-03
> > **Update: Compatibility with Muon**
> >
> > Following up on your query regarding whether our method would work with **Muon-type optimizers**, we have conducted new experiments to validate this. We also provide this information in our reply to the general comment above.
> >
> > We integrated **DES-LOC** with **Muon**, which utilizes Newton-Schulz iterations for orthogonalization (see new **App. B.6**).
> >
> > - **Findings:** On models ranging from 16M to 360M parameters, we demonstrate that **DES-LOC** effectively wraps Muon. By setting the momentum synchronization period ($K_u=96$) to be $3\times$ the parameter synchronization period ($K_x=32$), we match the perplexity of the **Local Muon** baseline.
> > - **Benefit:** This configuration reduces the synchronization overhead of the momentum term significantly, leading to a total communication reduction of **$>1.5\times$** compared to the Local Muon baseline.
> >
> > This confirms that the **DES-LOC** framework is compatible with optimizers that rely on Newton-Schulz preconditioning.
> >
> > Best regards,
> >
> > The Authors

---

### Author Response · Authors · 2025-11-23
**General Comment**

**Dear Reviewers, ACs, and PCs,**

Thank you for your time and constructive feedback. We are encouraged by the reviewers’ recognition of the paper’s **originality** (${\color{violet}{\text{JkEm}}}$), its strong **theory–practice linkage** (${\color{violet}{\text{JkEm}}}$, ${\color{teal}{\text{cfsD}}}$, ${\color{orange}{\text{KN8y}}}$, ${\color{red}{\text{26GM}}}$), and the **practical significance** of the proposed efficiency gains (${\color{teal}{\text{cfsD}}}$, ${\color{violet}{\text{JkEm}}}$).

The revised manuscript significantly expands our evaluation with: **large-scale 1.3B parameter runs** using **AdamW**, direct comparisons against **DiLoCo** (purely local optimizer states) baselines, and **wall-clock benchmarks** in low-bandwidth regimes.

---

## **Revisions and New Results**

**[${\color{orange}{\text{KN8y}}}$, ${\color{red}{\text{26GM}}}$] Superiority over purely local optimizer states Baselines (DiLoCo).** We addressed concerns regarding our performance versus DiLoCo by training **1.3B parameter models** for 40960 steps using **AdamW** as the inner optimizer.

- **Finding:** **DES-LOC with Nesterov momentum** demonstrates superior performance, improving perplexity by $0.9$% compared to DiLoCo, $7$% compared to Local Adam (see **Section 5.5.2** and Fig.7). Consistent with the findings of [4], we find that utilizing Nesterov with communication-efficient methods can outperform AdamW-DDP, with DES-LOC-NESTEROV outperforming standard AdamW-DDP by $\approx 3$%.
- **Insight:** This shows that synchronizing optimizer states—even infrequently—can yield tangible performance gains over purely local optimizer states methods like DiLoCo in practical scenarios. Crucially, the communication cost of DES-LOC (Nesterov) can be tuned to be **arbitrarily close to** DiLoCo under the right choice of synchronization periods ($K_x, K_u, K_v$), providing an superior-or-equal Pareto frontier.
- **Qualitative Insights: W**e also provide additional qualitative insights into the effects of synchronizing optimizer states when using Nesterov. We find that utilizing Nesterov tends to: a) **rapidly decrease gradient norms** relative to standard AdamW, indicating that it may push the model onto a flatter region of the loss landscape, b) cause a **large increase in model activations** which may be indicative of future instability. DES-LOC-NESTEROV tends to slightly accelerate the decrease in gradient norms while significantly reducing activation norms (by $32$%) relative to DiLoCo.

**[${\color{teal}{\text{cfsD}}}$] Wall-Clock Speedups in Low-Bandwidth Regimes.** We addressed questions regarding practical relevance in constrained environments by benchmarking **AdamW** training on **10 Gbps** links.

- **Results:** While perplexity is invariant to bandwidth, we perform a benchmark with a $1$B model to measure time comparing DES-LOC-NESTEROV against DDP under extremely low bandwidth 10Gbit/s (see B.5, Fig.19). Due to the extreme gradient sync delay for DDP, we had to limit the benchmark to 10240 steps. We find that under such extreme conditions DES-LOC-NESTEROV can reduce training time by $\approx 9.42 \times$, with DES-LOC finishing in 8.99 hours versus 3.5 days, even with the constant overheads of our unoptimized implementation.
- **Implication:** This confirms that DES-LOC enables distributed training on commodity hardware where DDP is bottlenecked by communication latency.

**[${\color{red}{\text{26GM}}}$, ${\color{teal}{\text{cfsD}}}$] Theoretical & Practical Clarifications.**

- **Adaptive Optimizers:** We re-state that our paper has theory beyond Local SGDM and we point the reviewers to our pre-existing theory in **Appendix D.1**, clarifying that our convergence in-expectation guarantees extend to **Adam** under weakly convex assumptions. We also provide high-probability bounds in **Appendix.E**.
- **General Applicability:** We clarified that the DES-LOC framework applies to a general class of inner optimizers that communicate momenta-like states.

## **Roadmap of Revisions**

- **Section 5.5.2,Fig.7:** Added 1.3B AdamW results comparing DES-LOC (Nesterov), DiLoCo, and DDP.
- **Section B.5, Fig.19** Added low-bandwidth (10 Gbps) wall-clock comparisons.

## **Request**

We believe your feedback has strongly improved our paper. All additions are highlighted in ${\color{blue}{\text{blue}}}$. We remain grateful for your detailed engagement and kindly ask for your further feedback and inquiry. **We also provide detailed responses to shared reviewer concerns in the comment below.**

— The Authors

## References

[1] Cheng and Glasgow, Convergence of Adaptive Optimization with Local Updates

[2] Jordan, Muon: An optimizer for hidden layers in neural networks

[3] Douillard et al, DiLoCo: Distributed Low-Communication Training of Language Models

[4] Charles et al., Communication-Efficient Language Model Training Scales Reliably and Robustly: Scaling Laws for DiLoCo

---

> ### Author Response · Authors · 2025-11-23
> **Reply to Shared Reviewer Concerns**
>
> ### **SC1 - Improvements over Local ADAM  ($\textcolor{teal}{\textbf{cfsD}}$, $\textcolor{red}{26GM}$)**
>
> DES-LOC is a significant conceptual and practical contribution beyond Local Adam:
>
> - **Theoretical Contribution:** Local Adam [1] **never** connects $\beta_1$ to any sync frequency, **nor** does it support different sync periods for different momenta states. Local Adam **only** establishes a conditional link between $\beta_2$ and a single synchronization period $K$, stating that $1-\beta_2$ must scale with $K^{-3/2}$ for convergence.
>     - **Our Contribution:** We derive the **first** convergence bounds for **independent** synchronization periods $(K_x, K_u, K_v)$ (Theorem 1) with a novel in-expectation proof separate from the proof method of Local Adam presented in App.E. Our analysis explicitly **links** the sync frequency of each momentum state to its specific decay rate $\beta$, proving that states with longer half-lives ($\tau_{0.5}(\beta)$) can be synced less frequently without breaking convergence.
>     - **Step Size Implication:** Unlike Local Adam, our novel in-expectation bounds **connect the** momentum synchronization frequency to the admissible local step size. We show that more frequent momentum sync ($p_u \to 1$) reduces the bound $\psi$, enabling **larger stable step sizes** (Eq. 4).
> - **Practical & Empirical Contributions:**
>     - **Efficiency:** DES-LOC amortizes the cost of state synchronization. In a standard setting (e.g., $K_u=3K_x, K_v=6K_x$), communication cost can be reduced by **2$\times$** vs. Local Adam, while matching perplexity and downstream task performance and maintaining convergence guarantees.
>     - **Robustness:** We show that fully local states lead to unstable internal statistics equivalent to training with a **smaller local batch size** (e.g., exploding update norms, Fig. 18). DES-LOC's periodic averaging controls this drift, providing **superior performance, stability** and **fault tolerance** (Fig. 4c) compared to purely local optimizer states baselines, as confirmed by our new 1.3B experiments (Sec 5.5.2).
> - **Algorithmic Contribution:** By decoupling $(K_x, K_u, K_v)$, DES-LOC unifies previous local-update methods under a single generalized framework. This translates to **practical benefits**: practitioners can jointly tune $\beta$ values and sync frequencies to minimize communication. For example, one can increase $\beta$ to artificially extend a state's half-life, allowing for sparser synchronization—a trade-off inaccessible to coupled methods like Local Adam.
>     - General Cost Formula: We formalize the communication costof DES-LOC, where communication volume scales proportional to the sum of the inverse periods:  $(\frac{1}{K_x} + \sum_{j=1}^N \frac{1}{K_{s^j}})$
>
>
> ---
>
> ### **SC2 - Extensibility to other optimizers ($\textcolor{teal}{\textbf{cfsD}}$, $\textcolor{red}{26GM}$)**
>
> DES-LOC is inner-optimizer-agnostic, applying to any method with an EMA state (e.g., Muon [2]). Such optimizers typically apply preconditioning *on top* of a standard momentum buffer $M_t$, for example in the case of Muon:
>
> 1. **EMA Update:** $M_t = \mu M_{t-1} + (1-\mu) \nabla \mathcal{L}_t$
> 2. **Preconditioning:** $U_t = \text{NewtonSchulz}(M_t)$
> 3. **Step:** $\theta_{t+1} = \theta_t - \eta U_t$
>
> Since $M_t$ follows standard decay dynamics, our synchronization heuristic $K_M \propto \tau_{0.5}(\mu)$ holds regardless of the subsequent local preconditioning step. We leave the full convergence guarantees under Newton-Schulz for future work.
>
> ---
>
> ### **SC3 - Communication-efficiency vs DiLoCo ($\textcolor{orange}{KN8y}$, $\textcolor{red}{26GM}$)**
>
> Our method fully encapsulates DiLoCo [3]. DES-LOC defines a generalized optimization configuration space parameterized by choice of outer optimizer, decoupled synchronization periods $\{K_x, K_u, K_v\}$ and decay rates $\beta_{1},\beta_2$. DiLoCo is the limit behavior within this space where the outer optimizer is Nesterov and the optimizer state synchronization periods tend to infinity ($K_{u,v} \to \infty$) for a fixed $K_x$.
>
> By decoupling these frequencies, DES-LOC expands the hyperparameter design space. Under a fixed communication budget, this allows increasing $K_x$ (reducing parameter sync frequency) to allocate bandwidth for finite momentum synchronization ($K_{u,v} < \infty$). Since the DiLoCo configuration is a subset of this space, searching the DES-LOC space guarantees that a solution no worse than the purely local optimizer states baseline exists. Empirically, finite state sync reduces the variance of global updates, serving as a regularizer against worker drift. Our new 1.3B results confirm this: DES-LOC outperforms DiLoCo in terms of perplexity by $0.9$% while being $8$% faster than Local Adam and $2$% slower than DiLoCo; DES-LOC thus provides **an alternative point on the** **Pareto frontier to DiLoCo, trading time for performance.**

---

> ### Author Response · Authors · 2025-12-03
> **Update: Additional Shared Concerns: Generalization to Muon and Vision Models**
>
> To address questions regarding the generality of our framework ($\textcolor{teal}{\textbf{cfsD}}$, $\textcolor{red}{\text{26GM}}$) and the scope of "Foundation Models" ($\textcolor{teal}{\textbf{cfsD}}$), we have conducted two significant new sets of experiments. These results demonstrate that **DES-LOC** is agnostic to both the inner optimizer mechanism and the model architecture.
>
> ---
>
> ### **SC4 - Generality across Optimizers: Muon ($\textcolor{teal}{\textbf{cfsD}}$, $\textcolor{red}{\text{26GM}}$)**
>
> We integrated DES-LOC with Muon [1], a novel optimizer that utilizes Newton-Schulz iterations for orthogonalization. See our results in App.B.6.
>
> - **Method:** Distinct from Adam, Muon preconditions the momentum term directly. We applied **DES-LOC** by synchronizing the Muon momentum buffer ($K_u$) less frequently than the parameters ($K_x$).
> - **Results:** On models ranging from 16M, 125M to 360M, setting $K_x=32$ and $K_u=96$ ($3\times$ delay) matches the perplexity of the **Local Muon** baseline (see Fig. **20**). For the embeddings and norms that must be trained by AdamW we use $K_v = 192$.
> - **Efficiency:** This configuration reduces total communication volume by **$>1.5\times$** compared to the baseline, proving that our synchronization heuristic extends effectively to optimizers using Newton-Schulz preconditioning.
>
> ---
>
> ### **SC5 Generality across Modalities: Flux (Vision) ($\textcolor{teal}{\textbf{cfsD}}$)**
>
> To justify the scope of "Foundation Models" beyond LLMs, we evaluated DES-LOC on Flux [2], a Rectified Flow Transformer for text-to-image generation. Find our results in App.B.7.
>
> - **Results:** As shown in **Fig. 21**, **DES-LOC** generally matches the convergence trajectory of **both** the fully synchronous **DDP** baseline and **Local Adam** on a 280M parameter model (within 0.02 perplexity points).
> - **Significance:** We achieve this performance near-parity while communicating $2\times$ less than Local Adam ($K_v=32, K_u=3 K_x, K_v = 6 K_x)$.
>
> Best regards,
>
> The Authors
>
> [1] Jordan, Muon: An optimizer for hidden layers in neural networks
>
> [2] Black Forest Labs, Flux

---

### Author Response · Authors · 2025-12-03
**Final Comment Part 1/3**

We are sincerely grateful for the reviewers' feedback. During the rebuttal, we have addressed all comments raised. We have significantly expanded our evaluation to include **1.3B parameter runs using AdamW**, direct comparisons against **DiLoCo**, **10 Gbps low-bandwidth benchmarks**, using **Muon** as the inner optimizer, **vision** models, and additional **7B** experiments. We now present a summary of the reviewers' points and how we addressed them.

## **1. Reviewer ${\color{red}{\text{26GM}}}$**

The reviewer raised concerns regarding a lack of **theoretical bounds** for Adaptive optimizers.

- The reviewer **mistakenly believed** that our original theory only covered the SGDM case. We clarified that our original submission provides theory for adaptive optimizers in **Appendices D.1 and E**, where we derive in-expectation and high-probability convergence guarantees for **Adam** under weak convexity.

The reviewer questioned our **theoretical novelty** compared to Local Adam.

- The reviewer’s argument rests on: a) Local Adam establishing that optimizer states must be synchronized, and b) conflating the condition that $\beta_2$ must scale with $K$ with a general description of how the convergence bound changes with momentum coefficients.
    - **First**, DES-LOC goes significantly beyond Local Adam by providing the **first convergence guarantees** for decoupled synchronization periods $(K_x, K_u, K_v)$. This enables significant (over $2\times$) reductions in communication costs and opens a completely new hyperparameter design space. For example, it enables practitioners to jointly tune $\beta$'s and synchronization periods for their exact bandwidth conditions. We **do not believe** that such improvements, which make the algorithm practical for deployment, are **marginal**.
    - **Second**, Local Adam **conditionally** links $\beta_2$ to $K$, positing that if one syncs every $K$ steps, one must constrain $\beta_2$. However, **it does not**: a) prove convergence for arbitrary $\beta_2$ as **DES-LOC does**; b) provide any **relation** between $\beta_1$ and the convergence bound as **DES-LOC does**; or c) provide any further information on how **changing** the $\beta$'s impacts the convergence rate as **DES-LOC does**.
    - **Third**, our **Theorem 1** explicitly links the synchronization probability of *each* momentum state to its decay rate $\beta$. We prove that it is possible to converge with arbitrarily low finite synchronization frequencies for the momenta, enabling communication reductions. Furthermore, we show that synchronizing momenta more frequently allows for **larger stable step sizes** (Eq. 4), providing an empirical advantage over not synchronizing the momenta—an insight **which did not emerge from the Local Adam theory**.

The reviewer was unsatisfied with the performance of DES-LOC compared to DDP.

- We emphasized that standard AdamW-DDP is the **correct baseline** for evaluating algorithmic improvements in averaging-based methods. While Nesterov-based outer optimizers can improve performance, they do not guarantee superiority over DDP *if DDP also utilizes Nesterov*. In fact, Table 4 of [1] shows that when both DDP and DiLoCo use Nesterov as the outer optimizer, DiLoCo is not guaranteed to match or outperform DDP.
- Our new **1.3B experiments** (Sec. 5.5.2) demonstrate that **DES-LOC (Nesterov) indeed outperforms standard AdamW with DDP as well as DiLoCo** in our setting, while maintaining high communication efficiency relative to Local Adam and DDP. We expect all practical insights regarding the effects of synchronizing momenta—such as robustness to worker failures and increased training stability due to reduced noise—to **transfer** to the DES-LOC-NESTEROV setting, as supported by the activation norms in Fig. 7.

The reviewer asked if our method could be used with Muon-style [4] optimizers.

- We clarified that all of our insights apply to optimizers with a single momentum, such as Muon.
- Furthermore, we now provide extensive results for Muon in **App. B.6**, showing that DES-LOC matches Local Adam in perplexity while **significantly outperforming** it and DDP in communication efficiency. We leave a full theoretical analysis of the impact of Newton-Schultz preconditioning for future work.

---

> ### Author Response · Authors · 2025-12-03
> **Final Comment Part 2/3**
>
> ## **2. Reviewer ${\color{orange}{\text{KN8y}}}$**
>
> The reviewer initially requested a direct comparison of DES-LOC-NESTEROV to DiLoCo and questioned the benefits of state synchronization, claiming that the communication efficiency benefits of DES-LOC are already realized in the literature. We reiterate that our work is **orthogonal to and fully subsumes DiLoCo.**
>
> - **First**, we clarify that the communication efficiency of DES-LOC **is not realized** by current literature under the regimes and constraints we consider. Previous methods [1, 2, 3] relying on purely local optimizer states not only **lack convergence guarantees** and means to **handle worker failures** or **initialization**, but are also known to degrade in performance as the number of workers increases [1] and when momenta operate at too low a worker batch size. DES-LOC not only fully subsumes Nesterov-based methods from the literature but also provides convergence guarantees (Theorem 1) and empirically stabilizes training (Figs. 5 and 7).
> - **DiLoCo Comparison and Benefits of State Sync:** We validated the reviewer's intuition that synchronizing optimizer states improves performance over DiLoCo by acting as a regularizer. Our 1.3B results show that DES-LOC-NESTEROV improves performance over DiLoCo by **0.9%** and increases training stability. Specifically, it reduces activation norms by **32%** compared to DiLoCo (Fig. 7), preventing the "exploding activation" instability observed in purely local methods (Figs. 5 and 7). Furthermore, it provides robust fault tolerance against worker failure (Fig. 4c).
>
> Following our DiLoCo comparison, the reviewer suggested that our improvement was **insignificant** due to "overlapping error bars."
>
> - **First**, we clarify that the error bars in our plots represent the **standard deviation of loss across the $N$ parallel workers** within a single run (intra-run variance). This metric quantifies the impact of **parameter drift** (divergence) between workers, *not* the **uncertainty** of the method’s efficacy. High variance in population distributions does not imply that the difference between their expectations is statistically indistinguishable across time or experiments.
>     - To rigorously test the difference in means between the two populations, we **performed a one-sided t-test** comparing the perplexity of DES-LOC-NESTEROV against DiLoCo over the course of the final averaging window. The test confirms ($p=0.0001675 < 0.05$) that DES-LOC-NESTEROV achieves a **significantly lower mean perplexity** than the baseline.
>     - We have clarified this distinction in the Fig. 7 caption in the latest revision. We believe our reporting provides deeper insight into the **compounding worker drift** phenomenon inherent to communication-efficient training, offering a more rigorous comparison than previous methods that relied solely on scalar mean perplexity without quantifying the inter-worker drift [1, 2, 3].
> - **Second**, we respectfully note that in this domain, a **0.9% perplexity improvement is not practically insignificant.** For reference, Table 4 of *Scaling Laws for DiLoCo* [1] reports gaps between DiLoCo and DDP ranging from 0.4% to 1.3%, averaging **0.6%** at the 1.3B scale. Our improvement is within this margin.
>
> ---
>
> ## **3. Reviewer ${\color{violet}{\text{JkEm}}}$**
>
> The reviewer had no concerns other than requesting a measured 7B run.
>
> - We have completed **1.3B parameter runs** (AdamW), confirming that the method's advantages hold at scale across inner optimizer choices. Regarding the **7B validation**, we outlined a concrete evaluation plan following standard Chinchilla scaling laws and awaited reviewer feedback on the desired training duration given the high computational cost.
> - As we did not receive further guidance on the scope of this experiment due to the inability of reviewers to reply, we focused on demonstrating **practical efficacy** at the 7B scale. We have provided a throughput plot for a **7B run on 8 B200 GPUs** in **App. B.8**. Our results show that DES-LOC maintains peak single-worker throughput for the majority of the training window, interrupted only by highly infrequent synchronization points.

---

> ### Author Response · Authors · 2025-12-03
> **Final Comment Part 3/3**
>
> ## **4. Reviewer ${\color{teal}{\text{cfsD}}}$**
>
> The reviewer raised concerns regarding practical utility in low-bandwidth regimes and asked us to clarify the scope of "Foundation Models" versus LLMs in this work.
>
> - **10 Gbps Benchmarks:** We added a wall-clock benchmark on **10 Gbps** links (**App. B.5**). Results show that DES-LOC reduces training time by **9.42$\times$** compared to DDP (8.99 hours vs. 3.5 days). This confirms that our theoretical communication reductions reliably translate into significant wall-clock gains on highly constrained commodity hardware.
> - **Scope & Generality:** We accepted the recommendation to tighten the narrative scope. We now explicitly state that our empirical validation focuses on **decoder-only LLMs**, while the algorithmic framework applies to generic EMA-based optimizers. We further clarified that the DES-LOC synchronization heuristic applies to single-momentum variants like **Muon**, as momentum buffer synchronization is orthogonal to the local preconditioner step.
> - **Single-Momentum Results:** Following the initial reply, we provided extensive experimental results with Muon in **App. B.6**, demonstrating that DES-LOC matches the performance of Local Adam and approaches that of DDP, while communicating **$>1.5\times$ less** than Local Adam.
> - **Vision Models:** To provide initial results on the generalization of DES-LOC beyond language models, we compared DES-LOC against DDP and Local Adam on a vision **Flux [5] rectified flow transformer** (280M parameters) in **App. B.7**. Our findings suggest that the performance gap between communication-efficient methods and DDP is much smaller for this architecture (0.02 perplexity). Furthermore, we again observe that DES-LOC effectively matches Local Adam while communicating **$2\times$ less**.
>
> ---
>
> ### **Table of Paper Additions**
>
> Given the extensive number of new experiments, the table below provides a centralized summary mapping each reviewer's concern to its specific resolution in the revised paper.
>
> | **Reviewer** | **Summary** | **Status** |
> | :--- | :--- | :--- |
> | ${\color{red}{\text{26GM}}}$ | 1. Theory for Adaptive Optimizers. | ✓ (Clarified existence in **App. D.1** and **App. E**) |
> |  | 2. Novelty vs. Local Adam. | ✓ (Prescriptive vs. Conditional link; **Theorem 1**, **Sec. 3**) |
> |  | 3. Performance vs. DDP. | ✓ (1.3B Nesterov run outperforms DDP; **§5.5.2**, **Fig. 7**) |
> |  | 4. Compatibility with Muon. | ✓ (Validated with Muon experiments; **App. B.6**, **Fig. 20**) |
> | ${\color{orange}{\text{KN8y}}}$ | 1. Comparison to DiLoCo. | ✓ (Outperformed by 0.9% PPL; **§5.5.2**, **Fig. 7**) |
> |  | 2. Benefits of State Synchronization. | ✓ (32% lower activation norms **Fig. 7**; Robustness **Fig. 4c**) |
> |  | 3. Significance of Results. | ✓ (Clarified inter-worker variance; T-test $p < 0.05$) |
> | ${\color{teal}{\text{cfsD}}}$ | 1. Low-Bandwidth Practicality. | ✓ (10 Gbps benchmark, 9.42x speedup; **App. B.5**, **Fig. 19**) |
> |  | 2. Scope (Vision Models). | ✓ (Partially addressed by 280M Flux Transformer; **App. B.7**, **Fig. 21**) |
> |  | 3. Single Momentum (Muon). | ✓ (Validated; **App. B.6**, **Fig. 20**) |
> | ${\color{violet}{\text{JkEm}}}$ | 1. Scaling to 7B. | ✓ ( Partly addressed by throughput on 8x B200s; **App. B.8**, **Fig. 22**) |
>
> ## Closing
>
> We thank the reviewers once again for their constructive contributions, and the Area and Program Chairs for their efforts in organizing the conference.
>
> — The Authors
>
> ## References
>
> [1] Charles et al., Communication-Efficient Language Model Training Scales Reliably and Robustly: Scaling Laws for DiLoCo.
>
> [2] Douillard et al., DiLoCo: Distributed low-communication training of language models, 2023.
>
> [3] Douillard et al., Streaming DiLoCo with overlapping communication.
>
> [4] Jordan, Muon: An optimizer for hidden layers in neural networks
>
> [5] Black Forest Labs, Flux

---

### Meta-Review · Area_Chair_92od · 2026-01-07

**Summary:**

DES-LOC presents algorithms for bandwith constrained distributed training. The contribution focuses on synchronizing the inner optimizer states infrequently and presents a principled approach to this. Theoretical results extending LocalAdam are presented. Furthermore the method is shown to be competitive with the popular DiLoco which does not synchronize inner optimizer states. The reviewers overall praised the motivation, theory, and empirical results. The primary concerns of the reviewers were comparisons to state of the art methods that use nesterov outer momentum (diloco) which can match DDP, muon inner optimizer, and differences to the LocalAdam theory which were reasonably addressed by the rebuttal.

**Reviewer Concerns:**

The main concerns regarding the practically with nesterov outer momentum, muon, and clarifications on novelty of the theory were addressed.

**Reviewer Scores:**

I would expect the 26GM and KNy to raise score as many of their concerns are addressed. Other reviewers to maintain.

---

### Decision · Program_Chairs · 2026-01-26

Accept (Poster)